# Poised PABP–RNA hubs implement signal-dependent mRNA decay in development

**Miha Modic** [1,2,3] ✉, **Klara Kuret** [1,3,4], **Sebastian Steinhauser**[1], **Rupert Faraway** [1,2], **Emiel van Genderen**[5], **Igor Ruiz de Los Mozos** [1,6], **Jona Novljan**[3], **Žiga Vičič** [3], **Flora C. Y. Lee** [1,2,7], **Derk ten Berge** [5], **Nicholas M. Luscombe**[1,8] & **Jernej Ule** [1,2,3] ✉

Signaling pathways drive cell fate transitions largely by changing gene expression. However, the mechanisms for rapid and selective transcriptome rewiring in response to signaling cues remain elusive. Here we use deep learning to deconvolve both the sequence determinants and the *trans*-acting regulators that trigger extracellular signal-regulated kinase (ERK)–mitogen-activated protein kinase kinase (MEK)-induced decay of the naive pluripotency mRNAs. Timing of decay is coupled to embryo implantation through ERK–MEK phosphorylation of LIN28A, which repositions pLIN28A to the highly A+U-rich 3′ untranslated region (3′UTR) termini of naive pluripotency mRNAs. Interestingly, these A+U-rich 3′UTR termini serve as poly(A)-binding protein (PABP)-binding hubs, poised for signal-induced convergence with LIN28A. The multivalency of AUU motifs determines the efficacy of pLIN28A–PABP convergence, which enhances PABP 3′UTR binding, decreases the protection of poly(A) tails and activates mRNA decay to enable progression toward primed pluripotency. Thus, the signal-induced convergence of LIN28A with PABP–RNA hubs drives the rapid selection of naive mRNAs for decay, enabling the transcriptome remodeling that ensures swift developmental progression.

Mammalian embryos transition their developmental fate in response to signaling dynamics. During implantation of the embryos, they undergo a pivot from Wnt to extracellular signal-regulated kinase (ERK)–mitogen-activated protein kinase kinase (MEK) signaling. This switch occurs at the embryonic rosette stage[1], where ERK pulsation is accompanied by rapid depletion of key transcription factors (TFs) such as *Nanog*, Krüppel-like factors (KLFs) 2, 4 and 5 and estrogen-related receptor-β (ESRRB) that constitute the naive regulon, which drives the naive pluripotency expression program[2,3]. The mRNAs encoding the naive regulon are highly expressed up to the early rosette stage in the embryo, as well as in the naive and rosette pluripotent stem cells (nPS and RS cells), but are rapidly depleted in the narrow time window of lumenogenesis (E5.1–E5.5)[1,4], when the primed expression program is switched on. The naive TFs form densely interconnected regulatory loops[5] such that even a single naive TF can be sufficient to reprogram cells back to the naive state[6]; thus, it is imperative to simultaneously

[1]The Francis Crick Institute, London, UK. [2]UK Dementia Research Institute at King's College London, London, UK. [3]National Institute of Chemistry, Ljubljana, Slovenia. [4]Jozef Stefan International Postgraduate School, Ljubljana, Slovenia. [5]Department of Cell Biology, Erasmus MC, University Medical Center, Rotterdam, The Netherlands. [6]Department of Gene Therapy and Regulation of Gene Expression, Center for Applied Medical Research, University of Navarra, Pamplona, Spain. [7]Centre for Developmental Neurobiology, Institute of Psychiatry, Psychology and Neuroscience, King's College London, London, UK. [8]Okinawa Institute of Science and Technology, Okinawa, Japan. ✉e-mail: miha.modic@kcl.ac.uk; jernej.ule@crick.ac.uk

clear the whole regulon. Failure to clear the naive mRNA regulon interferes with lumenogenesis in mouse and human embryos[4,7] but how such clearance is achieved and aligned with the dynamics of signaling remains unclear.

## Results

### *trans*-Acting regulators of mRNA clearance

We first set out to define the precise period of naive mRNA regulon clearance during the naive-to-primed pluripotency. Accordingly, 24 h after transferring nPS cells from naive medium containing Wnt agonist, MEK inhibitor and leukemia inhibitory factor (LIF) (2iLIF medium) to a medium activating MEK and nodal signaling ('priming' medium)[2], we observed a 4.5-fold median decrease in naive mRNA regulon (Fig. 1a), followed by a greater than twofold decrease in protein abundance of naive TFs (Extended Data Fig. 1a,b). To directly compare the mRNA decay rates at both time points, we used metabolic RNA sequencing (SLAMseq)[8] to directly measure the half-life of transcripts in nPS cells and after 12 h in the priming medium (Fig. 1b and Supplementary Table 1). To uncover the features that predict the changes in mRNA half-life between naive and priming PS cells, we trained machine learning classifiers based on gradient-boosted trees using various feature sets: mRNA dinucleotide frequencies, published and new datasets on RNA-binding sites of RNA-binding proteins (RBPs) determined with cross-linking and immunoprecipitation (CLIP), microRNA (miRNA) target sites, m6A sites, TAIL-seq[9], POSTAR2 features[10] and DeepBind[11] predictions of RBP binding within the 3′ untranslated region (3′UTR). The classifiers were trained to distinguish decreased and stable or increased half-lives using the top 1,000 most changed transcripts. Surprisingly, we identified RBP binding as the most predictive feature of the decreased mRNA stability (Fig. 1c,d and Supplementary Table 2), whereas no association was found with any specific miRNA families, A+U-rich elements, poly(A) tail length or other features (Extended Data Fig. 1c). Classifiers trained only by the enrichment of RBP binding to 3′UTRs were able to efficiently predict the change in half-life of mRNAs previously unseen by the model (area under the receiver operating characteristic curve (AUROC) = 0.8, accuracy = 0.72 and Matthews correlation coefficient (MCC) = 0.43) (Fig. 1e). Feature importance analysis, based on the relative influence, indicated LIN28A as the top-ranking RBP to predict the transcripts with decreased stability, along with poly(A)-binding protein C1 (PABPC1), both additionally verified by permutation-based feature importance (Extended Data Fig. 1d). In addition to LIN28A, top predicted RBPs included PABPC1, which was found capable of inducing mRNA decay under specific scenarios through its interaction with the Ccr4–Not complex[12], and a helicase SKIV2L, which can initiate 3′–5′ decay by binding to the 3′UTRs of a subset of transcripts[13].

To assess the role of LIN28A in regulating mRNA stability, we generated *LIN28A*-knockout (KO) PS cells and compared SLAMseq in wild-type (WT) and KO priming PS cells (Extended Data Fig. 1e,f). Notably, the decay rate of naive mRNA regulon mRNAs and mRNAs of the extended early pluripotency program displayed median increases of 28% and 35% in the priming and WT nPS cells, respectively, but this increase did not occur in the *LIN28A*-KO priming PS cells (Fig. 1f and Supplementary Table 1). We then repeated the machine learning to uncover features that predicted the changes in mRNA half-life between WT and *LIN28A*-KO priming PS cells, which identified the CLIP-based 3′UTR binding of LIN28A as the most important feature, further verified by permutation-based feature importance (Extended Data Fig. 1g,h). This postulates a central role of LIN28A-mediated mRNA decay in naive mRNA clearance during the transition to primed pluripotency.

Next, we investigated whether the LIN28A-mediated mRNA decay is necessary and sufficient to deplete the pluripotency regulon. To exclude the possibility that changes in mRNA stability are a secondary result of LIN28A-caused cell fate changes, we generated a doxycycline (Dox)-inducible iLIN28A–GFP (green fluorescent protein) PS

cell line on *LIN28A*-KO background (Extended Data Fig. 1i). Upon 6 h of LIN28A–GFP induction and 2iLIF withdrawal, we observed selective downregulation of the naive mRNA regulon with a concomitant decrease in the protein levels (Fig. 1g, Extended Data Fig. 1j and Supplementary Table 3). Most transcripts that were downregulated during the naive-to-primed transition were also downregulated upon acute LIN28A–GFP induction (Fig. 1h). Induction of LIN28A–GFP reduced the naive marker stage-specific embryonic antigen 1 (SSEA1) and increased the primed marker SSEA4 (Fig. 1i). To exclude the indirect contribution of miRNA-dependent stability effects, we observed that mRNA destabilization preceded expression of the *let-7* miRNA family (Extended Data Fig. 1k); thus, it could not be mediated by the repressive effect of LIN28 on *let-7* biogenesis[14,15]. Combined, these findings argue that the induction of selective mRNA decay by LIN28A is sufficient to drive destabilization of the naive mRNA regulon in a *let-7*-independent manner.

### Phosphorylated LIN28A induces mRNA decay

The MEK–ERK signaling cascade is crucial for commitment to primed pluripotency[1]. Levels of the naive mRNA regulon are rapidly decreased upon MEK–ERK activation[16], which is mainly ascribed to enhancer-mediated regulation[16–18], but such a transcriptional mechanism struggles to reconcile with the extremely rapid and selective mRNA clearance (Fig. 1a). Therefore, we examined whether LIN28A-induced mRNA decay might be the primary mechanism of naive regulon clearance. We inducibly expressed the LIN28A–FLAG transgene at physiological levels in *LIN28A*-KO nPS cells for 6 h in Wnt–LIF medium either with MEK inhibitor (2iLIF) or with fibroblast growth factor 2 (FGF2) stimulation of MEK–ERK signaling. While the medium changes on their own had a limited effect on mRNA levels, LIN28A–FLAG induction led to major downregulation of naive regulon mRNAs in FGF2 conditions, which were ~2.2-fold more downregulated as compared with induction in 2iLIF conditions (Fig. 2a and Supplementary Table 4) as exemplified by *Klf4* and *Nanog* mRNAs, which were 11-fold and 2.8-fold more downregulated, respectively (Fig. 2b,c and Extended Data Fig. 2a,b). This demonstrates that MEK–ERK signaling is required for LIN28A to be capable of inducing the decay of naive regulon mRNAs.

MEK–ERK activation induces LIN28A phosphorylation at position S200, which was found to increase LIN28A protein abundance by approximately 30% (ref. [19]). However, we show that this increase in LIN28A abundance cannot account for the dramatic and rapid mRNA destabilization observed in our study and it remains unknown whether and how phosphorylation impacts LIN28A function. Therefore, using publicly available phosphoproteomics data[2], we examined the magnitude of changes to dynamically regulated phosphosites and observed a greater than threefold increase in LIN28A S200 phosphorylation within minutes of MEK–ERK stimulation, achieved by switching PS cells from 2iLIF to the priming medium (Fig. 2d). We, therefore, generated two clonal cell lines with comparable expression of Dox-inducible WT and phospho-null LIN28A S200A PS cells, in which serine phosphorylation is abrogated by a substitution with alanine (Extended Data Fig. 2c,d). First, we examined the 3′-end RNA sequencing data to define the transcripts that were significantly downregulated upon 6 h of pLIN28A induced expression (Fig. 2e and Methods). Markedly, over 1,100 transcripts were downregulated upon the induction of LIN28A WT in the presence of FGF but only 66 transcripts were downregulated upon the induction of LIN28A S200A (Extended Data Fig. 2e–g).

Next, we investigated the extent of the functional impact of the S200A substitution, finding that it completely abrogates the capacity of LIN28A to activate selective mRNA decay (Fig. 2f). By calculating the interaction effect of LIN28A and MEK–ERK induction, we observed a 1.67-fold greater than additive destabilizing effect of MEK–ERK activation and LIN28A induction on both naive mRNA regulon mRNAs and transcripts undergoing LIN28A-mediated mRNA decay ($P$ value = $4.171 \times 10^{-8}$) but no interaction effect when using the phospho-null S200A mutant ($P$ = 0.161, determined using a two-sided

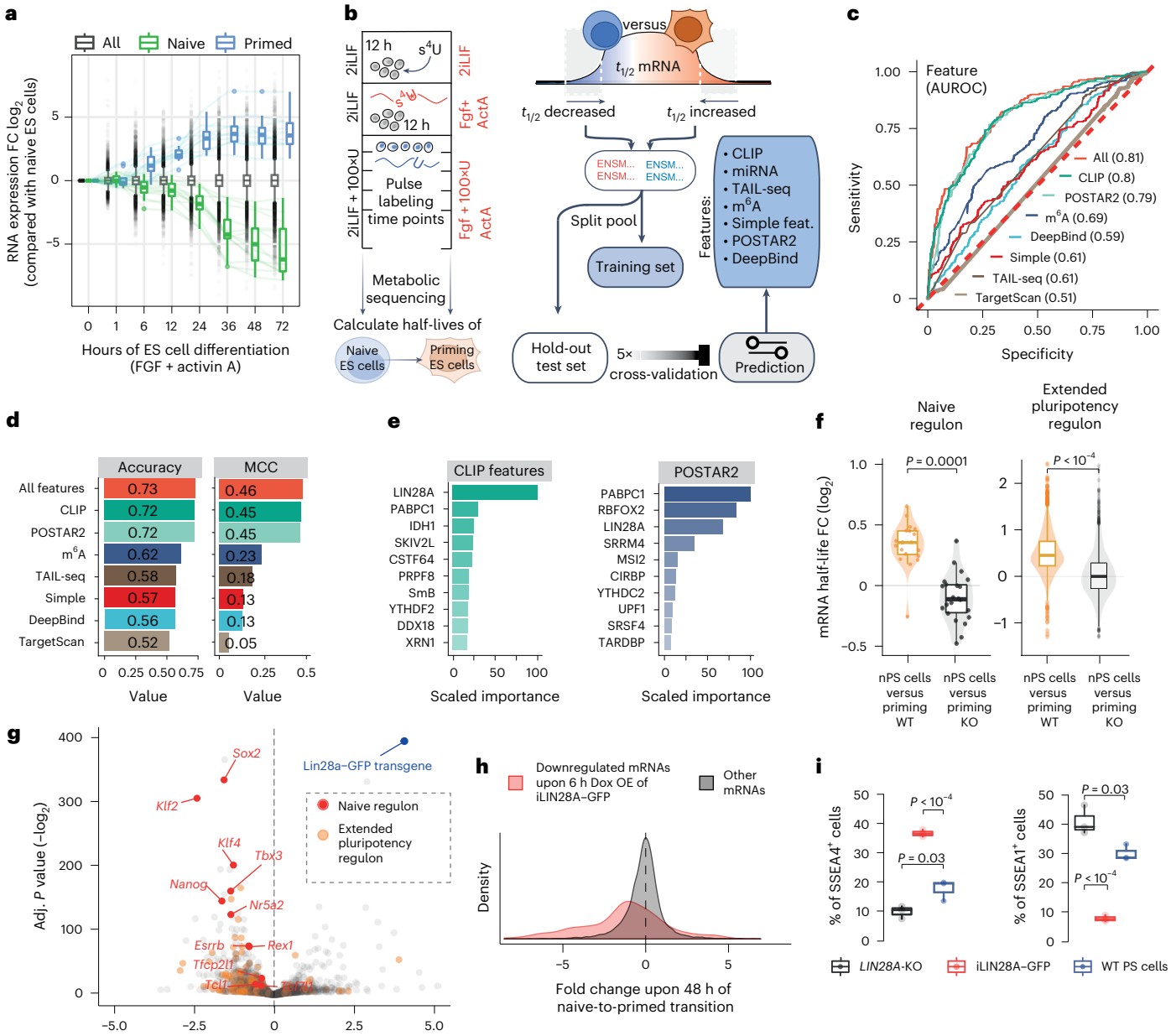

**Fig. 1 | Machine learning predicts LIN28A-dependent control of developmental switch in mRNA stability. a,** Temporal dynamics of relative mRNA levels (compared with 2iLIF at 0 h) of naive factors (green) and factors associated with primed cell fate (blue) downregulated and upregulated during PS cell naive-to-primed differentiation[2]. FC, fold change. **b,** Protocol for SLAMseq measurements of mRNA half-life during naive-to-primed pluripotency transition and machine learning framework for classification of genes based on their mRNA stability (Methods). $t_{1/2}$, half-life. **c,** ROC curves and AUROC values of all resulting binary classifiers trained on different feature sets. **d,** Accuracy and MCC of individual and combined features in predicting the direction of change in mRNA half-life during naive-to-primed conversion. **e,** Feature importance ranking for the best-performing classifiers: mouse PS cell CLIP datasets ($n = 62$) and POSTAR2 features[10] ($n = 54$) (see Methods for details). **f,** Violin plots depicting mRNA half-life fold change in WT and *LIN28A*-KO priming PS cells compared with nPS cells of naive mRNA regulon genes (left, $n = 23$), and genes encoding an extended early pluripotency program (outlined in Extended Data Fig. 1a) (right, $n = 931$). A two-sided two-sample $t$-test was performed to assess

the significance of differences between the annotated groups. **g,** Volcano plot displaying gene expression fold changes and their respective statistical score (Benjamini–Hochberg-adjusted $P$ value determined using Fisher's exact test with a false discovery rate (FDR) control level of $\alpha = 0.05$), comparing Dox-induced and untreated iLIN28A–GFP PS cells at 6 h ($n = 4$ biological replicates per condition, two clones). Naive pluripotency markers are labeled in red (marked factors fulfilled $P < 2^{-10}$) and extended pluripotency genes are labeled in orange. **h,** Density of fold change in gene expression changes of mRNAs downregulated upon 48 h of transferring nPS cells into primed medium ($n = 931$). Genes that are downregulated upon 6 h of Dox-induced iLIN28A–GFP overexpression are labeled in red, while the remaining genes are labeled in black. OE, overexpression. **i,** Flow cytometry quantification of primed (2 days of naive-to-primed transition) PS cells that were treated with Dox (iLIN28A–GFP) or left untreated (*LIN28A* KO), compared with WT PS cells and immunostained for SSEA1 and SSEA4. The mean of three independent primed experiments is depicted below (error bars, s.d.) and a two-sided two-sample $t$-test was performed to assess the significance of differences between the annotated groups.

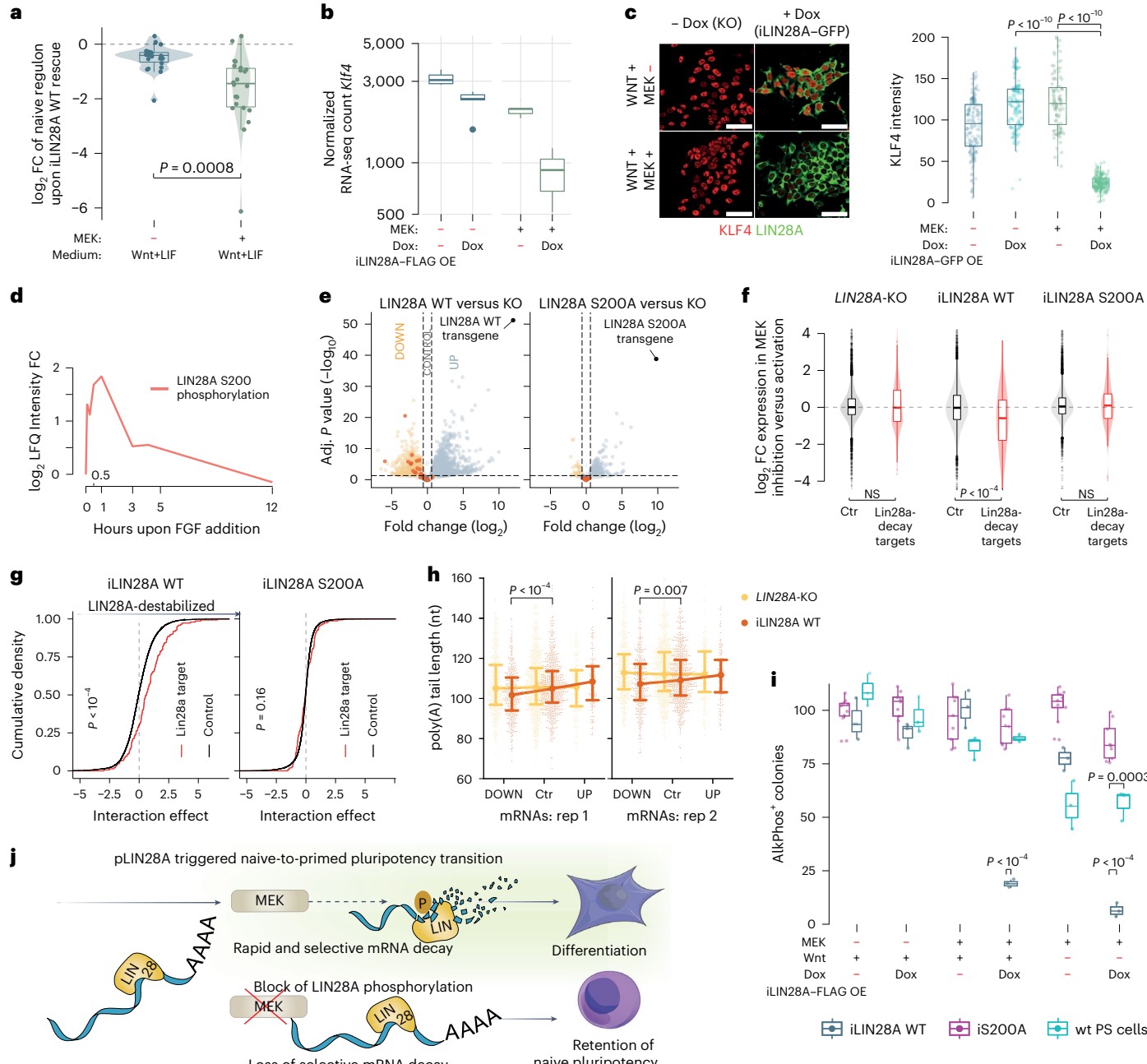

**Fig. 2 | MEK–ERK-driven LIN28A S200 phosphorylation is required for clearance of the naive regulon. a**, Fold change of DESeq-normalized counts for naive mRNA regulon upon 6 h of LIN28A WT induction in WL and 2iLIF conditions. A two-sided Welch's *t*-test was used to compute the *P* value. **b**, DESeq-normalized counts of naive marker *Klf4*, comparing Dox-induced and untreated iLIN28A in 2iL and MEK–Wnt conditions at 6 h. **c**, Representative immunofluorescence photomicrographs and fluorescence arbitrary units of immunostained naive marker KLF, comparing Dox-induced and untreated iLIN28A in 2iL and MEK–Wnt conditions at 6 h (*n* > 200). A two-sided Welch's *t*-test was used to compute *P* values. **d**, Temporal dynamics of LIN28A S200 phosphoproteome measurements as quantified using liquid chromatography–tandem mass spectrometry (LC–MS/MS) during the naive-to-primed pluripotency transition[2]. **e**, Differential expression of protein-coding genes obtained by comparing 3′-end RNA sequencing data for iLIN28A WT (left) or iLIN28A S200A overexpression (right) cells to KO cells in the FGF2-treated conditions. DESeq2 was used to calculate adjusted *P* values (FDR control level of α = 0.1) and fold changes. Naive regulon genes are marked in red. **f**, Violin plot showing the expression change of LIN28A-target (*n* = 438, twofold downregulated upon 6 h of LIN28A–GFP overexpression as in Fig. 1h; Benjamini–Hochberg-adjusted *P* value < 0.005)

and remaining mRNAs upon induction of LIN28A WT or LIN28A S200A together with MEK–ERK activation. NS, not significant. **g**, Density plot outlining the interaction effect for LIN28A-target and remaining mRNAs between the additive destabilizing effect of MEK activation and induction of WT or S200A mutant LIN28A in 2iLIF. **h**, The direct RNA sequencing of total RNA and quantification of poly(A) length using Nanopolish (see Methods for details). The mean poly(A) length for each gene is plotted for three groups of genes: downregulated, control and upregulated (670, 141 and 371 genes, respectively). The circle marks the mean poly(A) tail length in the gene group and the error bars represent the interquartile range of the distribution. A two-sided Mann–Whitney–Wilcoxon test was used to compute *P* values within each sample, comparing downregulated and control groups. **i**, Quantifications of alkaline phosphatase-positive WT, iLIN28A WT and iLIN28A S200A PS cell colonies after 24 h in 2iLIF, LFW (LIF, Fgf2, and Wnt), and LFI (LIF, FGF2 and IWP2) with and without Dox induction. Dots represent replicates from three independent experiments. A two-sided Welch's *t*-test was used to assess whether LIN28A rescues had different effects on the number of alkaline phosphatase colonies in LFW and LFI. **j**, Graphical model of MEK-dependent LIN28A phosphorylation and its binding to 3′UTRs of naive pluripotency mRNAs to induce their selective decay.

Mann–Whitney–Wilcoxon test) (Fig. 2g). We further confirmed that the naive mRNA regulon is destabilized by induction of the WT but not S200A LIN28A at both mRNA and protein levels (Extended Data Fig. 2h,i). To assess whether deadenylation contributes as a mechanism of selected mRNA decay, we performed long-read direct sequencing of mRNAs isolated from *LIN28A*-KO cells, with or without LIN28A WT induction in the presence of FGF2. The expression of LIN28A was induced for 4.5 h instead of 6 h, aiming to detect initial deadenylation that drives the later mRNA decay. Indeed, LIN28A WT induction selectively resulted in a significant trend of poly(A) tail shortening in the downregulated transcripts but not in the control and upregulated categories (Fig. 2h and Supplementary Table 5). This demonstrates that deadenylation has a role in pLIN28A-mediated selective mRNA decay, which is key to naive mRNA regulon clearance.

Because PS cells cannot dismantle naive pluripotency in the absence of LIN28A (ref. 20), we next asked whether LIN28A alone is sufficient to induce an exit from naive pluripotency and transition toward primed pluripotency. It was previously demonstrated that Wnt signals protect nPS cells from the differentiation-inducing effect of MEK activation[21,22]. Therefore, we examined whether LIN28A overexpression can overcome the total RNA protection from Wnt such that, upon MEK stimulation, it can induce the exit of naive pluripotency on its own. Indeed, within 24 h of simultaneous LIN28A induction and MEK activation in nPS cells, naive pluripotency was dismantled in spite of the opposing effects of Wnt signaling, as evidenced by the decreased ability to form alkaline phosphatase-positive colonies and the depletion of naive pluripotency hallmark CD117 (Fig. 2i and Extended Data Fig. 3a–c). MEK–ERK phosphorylation of LIN28A was also sufficient for commitment toward the primed state as indicated by the threefold increase in the number of SSEA4-positive cells (Extended Data Fig. 3c), whereas LIN28A phosphomutant or *LIN28A*-KO cells failed to differentiate toward the primed state and, hence, retained naive pluripotency features. Conversely, even in the absence of Wnt signaling, the capacity of MEK activation to dismantle naive pluripotency still depended on the phosphorylation of LIN28A, as *LIN28A*-KO cells, even upon LIN28A S200A overexpression, failed to exit naive pluripotency as evident by the retained colony-forming ability (Fig. 2i,j). These combined findings demonstrate that MEK activation cannot initiate priming without LIN28A, while, on the other hand, MEK activation can overrule the Wnt-mediated block of differentiation when acting through LIN28A to induce selective mRNA decay.

### *cis*-Acting determinants of mRNA decay

Next, we investigated the position-dependent sequence features that predict the regulation of developmental transcripts by LIN28A-mediated decay. We trained a hybrid convolutional neural network (CNN) and recurrent neural network (RNN) classifier based on the 3'UTR nucleotide sequence of upregulated and downregulated transcripts as the sole training feature[23]. This deep learning model's high efficacy and robustness (accuracy = 81.2% and AUROC = 0.89) in classifying the transcripts on the basis of sequence input alone implies that sequence features are essential determinants of developmental decay (Extended Data Fig. 4a,b). We further investigated which sequence motifs and 3'UTR regions are of high predictive importance by combining Shapley additive explanations (SHAP)[24] and TF-MoDISco (TensorFlow motif discovery and scoring)[25] (see Methods for details). This combined approach identified five motif clusters that were predictive of pLIN28A-dependent destabilization that were all A+U-rich; however, in contrast to the A+U-rich hexamers that were previously studied in relation to mRNA decay[26,27], they were longer and involved a multivalent and heterogeneous combination of U-rich interlinked with A-rich motifs that were highly positionally constrained at the ends of 3'UTRs (Fig.

We asked how these A+U-rich sequences enable pLIN28A to induce selective mRNA decay. Because A+U-rich sequences differ

from the documented WGG consensus bound by the zinc finger (ZnF) or GAU consensus bound by the cold-shock domain (CSD) of LIN28 (refs. 28,29), we performed further iCLIP experiments to identify potential alternative binding modes induced upon LIN28A phosphorylation. We produced iCLIP data with LIN28A WT and LIN28A S200A mutant upon 6 h of MEK activation with Wnt–LIF medium and with LIN28A WT in 2iLIF medium. Notably, a 60% increased efficiency of RNA cross-linking was observed upon MEK activation by normalizing the amount of cross-linked RNA, as seen on the iCLIP membrane, for the amount of IPed LIN28A WT protein, as seen by immunoblot (Fig. 3c and Extended Data Fig. 4f). To study the sequences that may contribute to such increased LIN28A–RNA binding upon MEK–ERK stimulation, we investigated the motifs enriched at the cross-link sites of both phosphorylated and unphosphorylated LIN28A WT and LIN28A S200A. By clustering the enriched motifs (Extended Data Fig. 5a,b and Methods), we obtained three distinct groups of *k*-mers enriched across experimental conditions. We identified groups of expected *k*-mers corresponding to the canonical WGG consensus (*n* = 24 *k*-mers) (Fig. 3d) or GAU consensus (*n* = 14 *k*-mers) (Fig. 3e), as well as a group of auxiliary AUU-rich motifs (Fig. 3f, *n* = 55). Interestingly, we noticed that, in the crystal structure of LIN28A bound to the *let-7* pre-miRNA, CSD forms π-stacking interactions and hydrogen bonds with an AUU stretch (Fig. 3f)[30]. Contacts of CSD with AUU or UUU motifs were also observed in three additional structures of LIN28A from three organisms[31], indicating that CSD might contribute to the interactions with these auxiliary AUU-rich motifs. We visualized the coverage of each motif group around the cross-link sites within 3'UTRs and observed that LIN28A phosphorylation strongly increases its binding to AUU-rich motifs while concomitantly decreasing its binding to the canonical ZnF-bound WGG motifs (Fig. 3d–f).

### PABP–RNA hubs mediate selective mRNA decay

To decipher how the increased pLIN28A binding to AUU-rich motifs could explain selective mRNA decay, we examined its cross-linking profiles in 3'UTRs. Notably, pLIN28A strikingly increased its binding toward mRNA termini in the region just upstream of the poly(A) signal (PAS), particularly in the downregulated mRNAs and the naive regulon (Fig. 4a,b, Extended Data Fig. 5c,d and Supplementary Table 6). To ascertain how the 3'UTR sequence of these mRNAs confers such selective binding of pLIN28a, we examined the incidence of representative trimers for the WGG, GAU and AUU motif group trimers, which were enriched in LIN28A iCLIP, in the region around the PAS. Notably, the valency of AUU trimers in this region was higher in the downregulated transcripts by a factor of 1.6 or 2.3 when compared with the control or upregulated transcripts, respectively, with a median of eight AUU motif trimers within the 100 nt upstream of the PAS (Extended Data Fig. 5e,f). This suggests an important role of AUU-multivalent regions at the termini of 3'UTRs in selecting mRNAs for developmental decay by recruiting the phosphorylated LIN28A.

Because LIN28A is not known to directly catalyze mRNA decay, we aimed to identify other factors that might contribute to this process in a selective fashion. Our machine learning model identified PABPC1 as a predictor of developmental mRNA decay (Fig. 1e). Moreover, previous studies showed that PABP can directly promote deadenylation and mRNA decay[12,32,33] and was found to interact with the CSD of LIN28A (ref. 34). We, therefore, produced PABPC1 iCLIP and observed enriched PABP binding at 3'UTR termini already in nPS cells, especially on downregulated transcripts (Extended Data Fig. 6). Surprisingly, LIN28A phosphorylation strikingly increased its binding to sites overlapping with PABP peaks but reduced its binding elsewhere (Extended Data Fig. 7a,b). Moreover, such increased pLIN28A binding was specific to the PABP peaks within the terminal 200 nt of the 3'UTR as compared with the remaining PABP peaks in the other 3'UTRs (Extended Data Fig. 7c and Supplementary Table 6). This indicates that the PABP–RNA hubs at the 3'UTR termini specify the mRNA repositioning of

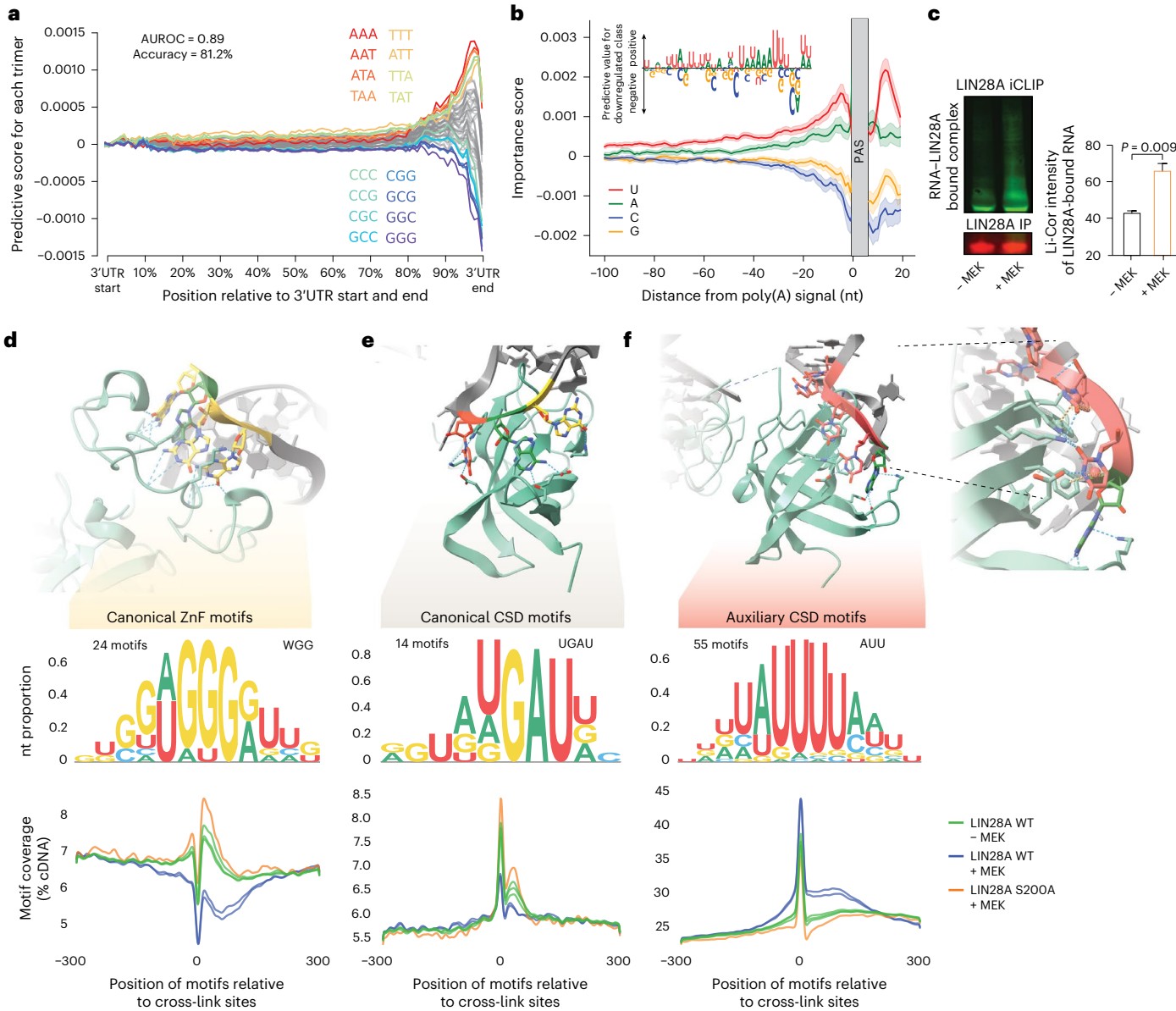

**Fig. 3 | Multivalent AUU-rich regions at 3′UTR termini enable selective mRNA decay and pLIN28A binding. a**, Line plots represent summarized trimer importance scores for the classification of 3′UTR sequences into downregulated and upregulated transcripts. Nucleotide-level importance scores were obtained with SHAP and summarized for each trimer along the evaluated groups of 3′UTRs (see Methods for details on analysis and sample size). Each line summarizes the predictive impact of trimers on the selection of mRNAs for decay and trimers with the highest predictive impact are highlighted. **b**, Line plots represent the mean nucleotide importance scores for the classification of 3′UTR sequences into downregulated and upregulated transcripts, obtained with SHAP. The scores are visualized in a region 100 nt upstream and 20 nt downstream of the terminal PAS in the 3′UTRs. The window of 5 nt was used for smoothing and the shaded areas represent the 95% confidence interval for the mean (computed with bootstrapping; $n = 1,000$). Inset: the metamotif represents an example

sequence motif, identified with TF-MoDISco[25], by clustering 3′UTR regions of high importance for the downregulated prediction class (see Methods for details). **c**, Representative Li-Cor imaging of the iCLIP nitrocellulose membrane, representing the protein–RNA complexes (top) and the amount of IPed LIN28A protein in the experiment (middle). The bar plot shows the quantification of RNA intensity comparing the LIN28A iCLIP capture of protein–RNA complexes with or without MEK–ERK inhibition ($n = 3$). A two-sided two-sample $t$-test was performed to assess the significance of differences between the annotated groups. **d–f**, Top: crystal structure of LIN28A (PDB 3TRZ)[30] in complex with GAGG (**d**), GAU (**e**) and AUU (**f**) RNA motifs through its ZnF (**d**) and CSD domains (**e**,**f**). Middle: corresponding metamotif representations of three motif groups that were identified by PEKA[48] as enriched in 3′UTRs in LIN28A CLIP data (Methods). Bottom: line plots showing the distribution of motif group coverage around cross-link sites located in the 3′UTRs (Methods).

pLIN28A, thus becoming poised for pLIN28A–PABP convergence upon MEK activation.

We reasoned that the convergence with pLIN28A might affect the quantity of PABP binding to the terminal mRNA hubs. We, therefore, proceeded with producing iCLIP data for PABPC4 and PABPC1 in *LIN28A*-KO cells with or without induction of LIN28A WT in the presence of FGF2. We observed the strongest cross-linking of both PABP

proteins to A-rich and AUU-rich motifs close to 3′UTR termini in all conditions, with PABPC1 directly overlapping pLIN28A and PABPC4 being slightly closer to 3′UTR termini (Fig. 4c,d and Extended Data Fig. 8a–d). In *LIN28A*-KO cells, the terminal 300 nt in the 3′UTR of downregulated mRNAs had only 50% more PABP binding compared with control mRNAs but ~4.5-fold more binding upon phosphorylation of LIN28A (Extended Data Fig. 8e,f). This increase in PABP binding

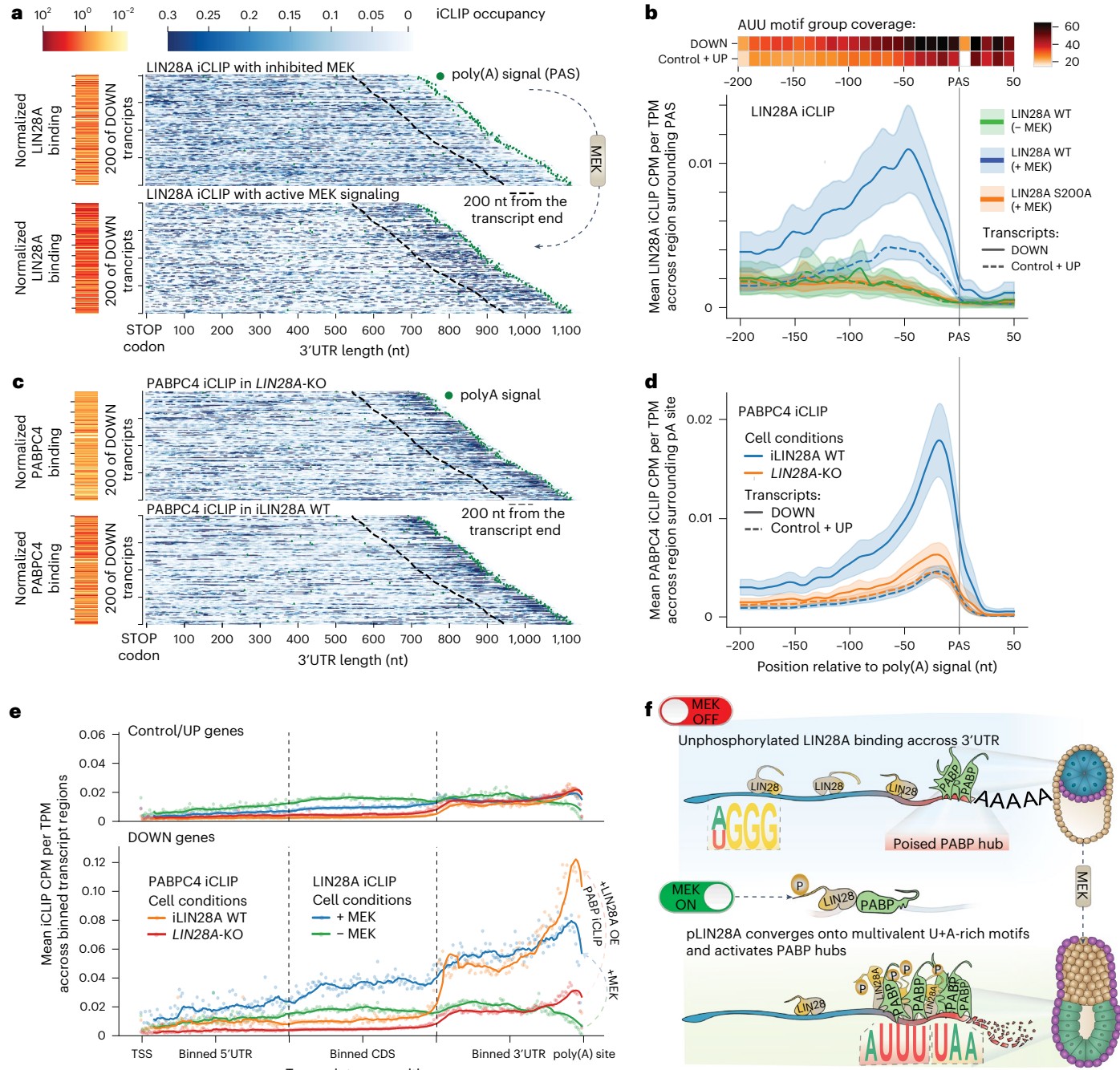

**Fig. 4 | pLIN28A converges with PABP–RNA hubs at 3′UTR termini to trigger selective mRNA decay. a,c** Heat maps in blue showing the distribution of iCLIP signal across 200 downregulated 3′UTRs for LIN28A (**a**) and PABPC4 (**c**) (Methods). For LIN28A, iCLIPs in 2iLIF-treated (top) and FGF2-treated (bottom) cells are shown. For PABPC4, iCLIPs in FGF2-treated *LIN28A*-KO cells without (top) and with (bottom) LIN28A overexpression are shown. Heat maps in orange depict the total level of CLIP signal in each 3′UTR, normalized by 3′UTR length, expression and library size, thus reflecting the overall RBP abundance in these regions. **b,d,** Line plots indicating the mean of expression-normalized cross-link coverage in the region 200 nt upstream and 50 nt downstream of the PAS for iCLIPs of LIN28A WT and LIN28A S200A in the FGF2-treated condition (+MEK) and LIN28A WT in the 2iLIF-treated condition (−MEK) (**b**) and PABPC4 in *LIN28A*-KO cells with or without the induction of LIN28A WT in the presence of FGF2 (**d**). Included 3′UTRs were filtered for minimum length and expression

level as described in Methods. Top: heat map showing the mean percentage of nucleotides covered by AUU motif groups (each square is a 10-nt bin) in downregulated (n = 831) and control or upregulated (n = 2,372) gene. Shaded areas indicate the 95% confidence interval. **e,** Line plot indicating the metaprofile of mean expression-normalized cross-link coverage for LIN28A and PABPC4 across the 5′UTR, CDS and 3′UTR regions of transcripts in downregulated (n = 831) and control or upregulated (n = 2,372) genes. Overlapping cross-links for each transcript region were normalized and divided into 100 bins. The mean normalized cross-link coverage over all transcripts in a group was obtained for each bin (dot) across the transcript region (see Methods for details). The line shows a rolling mean across ten bins. TSS, transcription start site. **f,** Graphical overview outlining how the convergent binding of pLIN28A together with PABP at AUU-rich terminal regions triggers selective mRNA decay essential for epiblast morphogenesis.

was observed only at the 3′UTR termini of downregulated mRNAs and only upon 3′UTR terminal relocation in pLIN28A binding (Fig. 4e and Extended Data Fig. 8g,h). This demonstrates that the convergence of pLIN28A into the poised PABP–RNA hubs selectively enhances the PABP binding to these hubs in downregulated mRNAs and this rapid pLIN28A–PABP coassembly at mRNA termini determines the selectivity of mRNA decay (Fig. 4f).

## Discussion

We found that a central effector of mRNA metabolism, PABP, binds in a poised state to the mRNA termini of mRNAs in naive pluripotent cells, which enables a rapid response to MEK–ERK signaling. MEK phosphorylates the intrinsically disordered region (IDR) of LIN28A, inducing its repositioning from the canonical G-rich motifs to multivalent AUU-rich PABP hubs, which in turn activates selective mRNA decay to drive cell fate progression toward primed pluripotency. The PABP hubs at the multivalent 3′UTR termini are, thus, poised for the rapid, signal-induced convergence between the phosphorylated regulatory RBP (LIN28A) and its effector (PABP). This demonstrates that multivalent RNA interactions complement the well-known roles of protein–protein interactions[33] in the convergent assembly of regulator–effector hubs to selectively remodel the transcriptome.

Thus, we suggest that the multivalent mRNA termini act as the long-sought nexus[33] connecting effector proteins (that is, PABP) with activated regulatory RBPs (that is, pLIN28A). We demonstrate the biological potency of pLIN28A in directly regulating mRNA decay, which complements the well-studied capacity of LIN28 paralogs to indirectly affect mRNA stability through *let-7* repression[14]. The strong mRNA stability defects upon LIN28A loss indicates a lack of redundancy with LIN28B, which is predominantly a nucleolar protein[35], with undetectable expression before priming in our cells. Beyond tumorigenesis[15,36], pLIN28A-mediated direct regulation of mRNA decay might also account for the developmental timing of *Caenorhabditis elegans* larvae[37] and the functions of LIN28A in glucose metabolism[20] and could be relevant for a range of diseases in which LIN28A is implicated, including mouse tissue repair[38], human β cell differentiation[39] and Parkinson disease pathogenesis[40].

PABP was found capable of inducing mRNA decay by recruiting various deadenylases, including the Ccr4–Not complex[12,32,33], whereas it is also known to repress decay by binding to the non-templated poly(A) tail and protecting it from deadenylation[33]. It was not known, however, whether PABP could switch between these opposing roles on specific transcripts. Here we found that the PABP–RNA hubs at the termini of 3′UTRs might control such a switch, as they are key selectors of developmental mRNA decay that mediate the signal-induced, IDR-mediated convergence with LIN28A. While the mechanism of PABP–RNA hubs remains to be directly evaluated in the context of the functions of PABP at the poly(A) tails[41], we posit that the multivalent 3′UTR regions compete with the poly(A) tail for PABP binding, thus shifting PABP away from its protecting role on the poly(A) tails.

The observation that PABP can bind to A+U-rich mRNA sequences in 3′UTRs is in agreement with previous protein–RNA cross-link studies in yeast and mammalian cells[42–44] and in vitro studies that identified a broad RNA specificity of the ribosomal DNA recombination mutation 3 (RRM3) and RRM4 domains of PABP[45]. The PABP bound to A+U sequences appears to be poised for convergence with regulatory RBPs such as pLIN28A that bind at nearby 3′UTR regions, thus enabling the rapid switch in PABP functionality toward deadenylation and mRNA decay. The genomically templated nature of multivalent motifs ensures specificity for subsets of 3′UTRs, thus explaining how PABP can be primed for a functional switch on selected mRNAs upon convergence with a regulatory RBP[46]. Interestingly, structural studies of transposon mRNAs showed that AUU-rich 3′UTR sequences can interact with the poly(A) tail; thus, the convergent pLIN28A–PABP binding could also involve changes in RNA structures at 3′UTR termini[47]. In sum, we show

how the multivalent 3′UTR termini serve as hubs for regulator–effector convergence[47] to determine the selectivity and timing of mRNA decay that drives the exit from naive pluripotency, thus maximizing the speed of signal-induced transcriptome remodeling.

## Online content

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

## Methods

### Embryonic stem cell (ES cell) culture

All mouse ES cell lines, such as IDG3.2 (Institute of Stem Cell Research, Helmholtz Zentrum München) and V6.5 (NBP1-41162, Novus Biologicals), were maintained in 1:1 neurobasal (21103049) DMEM (11320033) containing N2 (17502048) and B27 (17504044) supplements, 1% Glutamax (35050), 1% non-essential amino acids (1140050) and 0.1 mM 2-mercaptoethanol (31350-010) (all Thermo Fisher Scientific), as well as 1,000 U per ml LIF (ESGRO ESG1107, Merck), with the additional use of small-molecule inhibitors. For the 2iLIF condition, 1 mM MEK inhibitor PD0325901 (1408, Axon Medchem) and 3 mM glycogen synthase kinase 3 (GSK3) inhibitor CHIR99021 (4953/50 Tocris) were added for >2 days. Cells were passaged using Stempro-Accutase (A1110501, Thermo Fisher Scientific). Cells were grown on gelatin-coated plates that were prepared by first incubating the wells with 0.1% gelatin (EmbryoMax ES-006-B, Merck) and 1% FBS (EmbryoMax ES Cell Qualified FBS, ES-009-B, Merck) for 1 h at 37 °C followed by a rinse with PBS.

To transit from the ES cell to MEK-activated and Wnt-inhibited (LFI) or MEK–Wnt-activated state, Accutased ES cells were seeded onto gelatin-coated and FBS-coated plates in N2B27 medium supplemented with 1,000 U per ml LIF (ESGRO ESG1107, Merck), 12 ng ml⁻¹ bFGF (100-18B, Peprotech) and either 2 μM inhibitor of Wnt production 2 (IWP2; 3533, Tocris) for LFI or 3 mM GSK3 inhibitor CHIR99021 (4953/50 Tocris) for LFW cells. To transit from naive ES to priming ES cells, Accutased naive ES cells were seeded onto gelatin-coated and FBS-coated plates in N2B27 medium supplemented with recombinant 20 ng ml⁻¹ activin A (338-AC-050, R&D Systems) and 12 ng ml⁻¹ bFGF (100-18B, Peprotech) for 12 h or as indicated in the figure legends (12, 24, 36 or 48 h). To determine developmental genes that were downregulated during naive-to-primed transition, publicly available data were used[2]. Epiblast-derived stem cells (EpiSCs) were cultured on gelatin-coated and FCS-coated plates in N2B27 medium supplemented with recombinant 12 ng ml⁻¹ bFGF (100-18B, Peprotech), 20 ng ml⁻¹ activin A (338-AC-050, R&D Systems) and 2 μM IWP2 (3533, Tocris) to suppress spontaneous differentiation. EpiSCs were passaged 1:4–1:10 every 3 days by triturating the colonies into small clumps using 0.5 mg ml⁻¹ collagenase IV (Sigma). MEK–Wnt-inhibited RS cells were cultivated with 1,000 U per ml LIF, 2 μM IWP2 and 1 μM PD0325901 (manufacturers as mentioned above) and were passaged as ES cells. All cell lines in culture were tested for *Mycoplasma* contamination every 2–3 months.

### Generation of CRISPR–Cas9 genome engineering mouse ES cell lines

To generate *Lin28a*-KO cells, we used two independent targeting strategies allowing us to verify the effect of *LIN28A* KO independent of the use of individual guide RNAs (gRNAs). The first strategy entailed the use of two gRNAs with target sites flanking the second *Lin28a* exon to excise it on both alleles. gRNA oligos were cloned into the SpCas9-T2A-PuroR/gRNA vector (px459) by cut ligation (gRNA sequences ACCCGTCTGGGAGATCCGG and AGGCTGGGAGTTCCGAGGTA). ES cells were transfected with an equimolar amount of each gRNA vector. Then, 2 days after transfection, cells were plated at clonal density and subjected to a transient puromycin selection (1 μg ml⁻¹) (P8833, Sigma-Aldrich) for 48 h. Clones were picked 5–6 days later, expanded in 96-well plates and PCR-screened to identify clones with alleles harboring deletions. PCR primers (CAGCCAGGACAGGTTTCTTC and GCAGTTACTACAAACTCAAAACTG) were used to identify clones in which the *Lin28a* second exon was removed. Removal of the *Lin28a* second exon was confirmed with Sanger sequencing and loss of LIN28A expression was assessed by western blot. For CRISPR–Cas gene editing, all transfections were performed using Lipofectamine 3000 (Thermo Fisher Scientific) according to the manufacturer's instructions.

### Generation of a stable inducible LIN28A-expressing mouse ES cell line

To generate the inducible PiggyBac system, the LIN28A–eGFP, FLAG–LIN28A WT and FLAG–LIN28A S200A constructs were synthesized as gBlocks (Integrated DNA Technologies) and inserted into the enhanced PiggyBac (ePB) vector for stable integration[49]. This plasmid contains a TET-on system for inducible transgene expression. Mouse ES cells were cotransfected with 4 μg of transposable vector and 1 μg of the piggyBac transposase using Lipofectamine 3000 (Thermo Fisher Scientific) according to the manufacturer's instructions. Selection in 5 μg ml⁻¹ blasticidin was initiated 3 days after transfection and maintained until resistant colonies became visible. For LIN28A–eGFP PiggyBac lines, individual resistant inducible clones were selected, expanded and checked for LIN28A–GFP expression, whereas, for expression of the FLAG–LIN28A WT and FLAG–LIN28A S200A transgenes evaluated with western blot, two verified clones were used for studies including QuantSeq (Fig. 2e). The LIN28A transgene was induced by adding 1 μg ml⁻¹ Dox (Thermo Fisher Scientific).

### Cell culture immunofluorescence

Cells were grown in Geltrex-coated (Thermo Fisher Scientific, A1569601) eight-well chamber slides (80826, Ibidi) and fixed with 4% PFA–DPBS solution (Thermo Fisher Scientific; 16% formaldehyde (w/v), methanol-free, 28906) for 15 min at room temperature (RT) and permeabilized using 0.3% Triton X-100–DPBS solution for 15 min at RT. Primary and secondary antibodies were diluted as per the manufacturer's recommended concentrations in 10% FBS and 0.3% Triton X-100–DPBS and incubated for 1 h at RT using the following antibodies: anti-Tbx3 (sc-17871, Santa Cruz), anti-LIN28A (A177, Cell Signaling and ab63740, Abcam), anti-*Klf4* (AF3158, R&D Systems), anti-*Nanog* (8822, Cell Signaling), anti-Sox2 (2748S, Cell Signaling), anti-KLF2 (Mab5466, R&D Systems), donkey anti-rabbit immunoglobulin G (IgG) 555 (A31572, Invitrogen), donkey anti-goat IgG 488 (A11055, Invitrogen) and donkey anti-goat IgG 633 (Invitrogen, A21082). The samples were washed with DAPI (50 μg ml⁻¹) solution and imaged using a Nikon Ti2 spinning disk confocal microscope with the parameters described in the next section.

### Super-resolution cell imaging

Steady-state super-resolution imaging of live ES cells and immunofluorescence images were acquired using an immunoOlympus IX83 microscope equipped with a VT-iSIM super-resolution imaging system (Visitech International) and Prime BSI Express scientific complementary metal–oxide–semiconductor camera (Teledyne Photometrics) with an Olympus ×150 (1.45 numerical aperture) TIRF Apochromat oil immersion objective. The imaging system was equipped with a motorized stage with piezo Z (ASI) and environmental chamber maintained cells at 37 °C with 5% CO₂ (Okolab). Images were acquired with the 405-nm, 488-nm, 561-nm and 640-nm laser lines, generating an image size of 1,541 × 1,066 with 1 × 1 binning and a pixel size of 43 nm. The microscope was controlled with Micro-Manager version 2.0 gamma software[50]. ER-Tracker (E12353, Thermo Fisher Scientific) was used for staining of the cytoplasm in live cells. Deconvolution was performed using Huygens (Scientific Volume Imaging).

### Flow cytometry

For flow cytometry experiments, single-cell suspensions were made using Accutase (A6964, Sigma-Aldrich) for 5 min in 37 °C or enzyme-free cell dissociation buffer (13151014, Gibco) for 30 min at 37 °C, washed with 5% FBS (EmbryoMax ES Cell Qualified FBS, ES-009-B, Merck) in PBS and incubated with fluorophore-conjugated antibodies against anti-CD117 (c-Kit) antibody (105820, Biolegen) or SSEA4 (MC813-70, Thermo Fisher Scientific) for 30–60 min on ice. Cells were centrifuged and resuspended. Then, 10,000 cells were analyzed using an LSR Fortessa cytometer (BD Biosciences, BD FACSDiva Version 9.2.).

Cell debris were excluded by forward and side scatter gating. FlowJo was used for data analysis.

## Western blotting

Cells were trypsinized and lysed using radioimmunoprecipitation assay (RIPA) buffer containing phosphatase (Sigma-Aldrich, 4906837001) and protease (Merck, 539134) inhibitors. After the addition of 4× LDS loading buffer (Thermo Fisher Scientific, NP0007) with 2-mercaptoethanol (Sigma-Aldrich, M3148), samples were heated to 95 °C for 5 min. Samples were run on NuPAGE 4–12% with Bis-Tris (Thermo Fisher Scientific, NP0321PK2) and blotted using the Power Blotter XL System (Thermo Fisher Scientific, PB0013). Following three 5-min washes with PBS-T, membranes were blocked with 5% milk powder (T145.1, Carl Roth) in TBS-T. Membranes were then incubated overnight at 4 °C with 5% milk powder in PBS-T containing the primary antibodies at the following dilutions: 1:1,000 anti-LIN28A (A177, Cell Signaling and AF3757, R&D Systems); 1:4,000 anti-H3 (Abcam, ab1791), 1:2,000 anti-glyceraldehyde 3-phosphate dehydrogenase (anti-GAPDH; Cell Signaling, 2118S) and 1:1,000 *Klf4* (AF3158, R&D Systems). After three 5-min PBS-T washes, the membrane was incubated with horseradish peroxidase-conjugated goat anti-rabbit IgM (sc-2030, Santa Cruz) or goat anti-mouse IgG (Dianova, 115-035-003) in 5% milk powder in PBS-T. Following four 10-min washes with PBS-T, the membrane was incubated for 1 min with Clarity Western enhanced chemiluminescence substrate (170-5060, Bio-Rad Laboratories) and imaged with an Amersham Imager 600 (GE, 29-0834-61).

## Cell fractionation

A total of ~$10^7$ cells were resuspended in 500 µl of buffer S (10 mM HEPES pH 7.9, 10 mM KCl, 1.5 mM MgCl$_2$, 0.34 M sucrose, 10% glycerol, 0.1% Triton X-100, 1 mM DTT and proteinase inhibitor (04693116001, Roche)). The cells were incubated for 5 min on ice. Nuclei were collected into a pellet by low-speed centrifugation (4 min, 1,300$g$) at 4 °C. The supernatant containing the cytoplasm fraction was further clarified by high-speed centrifugation (15 min, 20,000$g$, 4 °C) to remove cell debris and insoluble aggregates. To further purify the nuclei, they were washed three times in buffer S and spun down (1,700$g$ for 5 min) at 4 °C to get a soluble nuclear fraction.

## RNA preparation, reverse transcription–qPCR assays and RNA sequencing

Total RNA was prepared from cell pellets using an miRNeasy Micro Kit (Qiagen, 217084) according to the manufacturer's instructions. The miR *let-7* levels were analyzed using the miScript protocol (Qiagen, 218193) using primers amplifying miR *let-7* (Qiagen, MS00005866; *let-7*g Qiagen, MS00010983). For QuantSeq 3′ mRNA sequencing (mRNA-seq), 0.5 µg of total RNA was used per library prepared using the Lexogen QuantSeq-FWD kit (Lexogen GmbH, 0.15) according to the manufacturer's instructions. All libraries were evaluated on an Agilent 2200 TapeStation using the High-Sensitivity D1000 ScreenTape (Agilent, 5067-5585) and DNA concentration was measured using a Promega Quantifluor double-stranded DNA system (Promega, E2670). Samples were sequenced using HiSeq4000 to generate 100-nt single-end reads.

Thiol-linked alkylation for the metabolic sequencing of RNA (SLAMseq) was generated according to the published protocol[8]. Naive or priming ES cells were treated with 100 µM 4-thiouridine (s4U; Sigma, T4509-25MG) for 24 h of pulse labeling. During the last 15 h of pulse labeling, fresh s4U-containing medium was exchanged every 3 h to enhance s4U incorporation into the RNA species. Before initiation of the chase labeling with 10 mM uridine-containing (Sigma, U3750-25G) growth medium (100× excess of uridine compared with initial s4U concentrations), the medium was removed from the cells, which were washed twice with chase-labeling medium. The chase labeling was initiated upon washing off the pulse-labeling medium and, at the indicated

time points, the medium was removed and cells were lysed directly in Qiazol (Qiagen, 79306), followed by RNA preparation.

## QuantSeq 3′ mRNA-seq processing and analysis

Reads obtained with QuantSeq 3′ mRNA-seq were quantified using Salmon[51] with a transcriptome built from GENCODE M22 transcripts. To monitor the expression level of the Lin28a–GFP transgene, the Lin28a–GFP sequence was included with the decoy sequences derived from the whole genome sequence[52]. The command used to quantify RNA expression with Salmon was 'salmon quant -i {input.reference} -l A -r {input.fastq_trimmed} --threads {threads} --validateMappings --gcBias --seqBias -o {out_dir}'.

Next, differential expression analysis was performed using DESeq2 (ref. 53), with effect size shrinkage implemented using the apeglm package[54]. To find genes that were differentially regulated upon Dox induction of LIN28A–GFP (Fig. 1g,h), we performed differential expression analysis comparing iLIN28A–GFP inductions following 6 h of Dox induction using Dox-inducible iLIN28A–GFP PS cells built in the *LIN28A*-KO background with and without 6 h of Dox induction. For iLIN28A–GFP overexpression analysis, the developmentally downregulated genes were defined as genes at least twofold downregulation at the 24-h and 48-h time points in previously published data[2] ($n$ = 931) and a list of naive pluripotency genes was derived from the same study.

To find genes that were differentially regulated upon phosphorylation of LIN28A, we performed differential expression analysis comparing *LIN28A*-KO cells, cultured in the presence of FGF2, with or without the induction of LIN28A WT (Fig. 2e, left) or LIN28A S200A (Fig. 2e, right). Reads were obtained with QuantSeq 3′ mRNA-seq, and quantified with Salmon as described above. Differentially expressed genes were determined with DESeq2 (ref. 53), with effect size shrinkage implemented using the apeglm package[54], by applying criteria for an adjusted $P$ value < 0.05 and fold change ≥ 1.5. The control group of genes was defined by an adjusted $P$ value ≥ 0.05 and |log$_2$(fold change)| < 0.5. For the subsequent analysis of 3′UTR features, we filtered the gene groups to contain only protein-coding genes with a minimum 3′UTR length of 100 nt and an expression level ≥ 5 TPM (transcripts per million reads) in *LIN28A*-KO or WT in the FGF2-treated condition. This resulted in 1,183 downregulated genes, 989 upregulated genes and 2,703 control genes. Representative 3′UTRs for each gene were annotated on the basis of the most abundant mRNA isoform in cells expressing LIN28A WT cultured in 2iL medium. Expression levels were estimated from TPM values obtained with Salmon by taking a mean TPM value for each transcript across all replicate experiments.

## RNA stability prediction methods

To calculate mRNA T>C conversion, the pipeline was enclosed in Snakemake (version 5.3.0)[55]. For analysis of mRNA 3′-end sequencing data, reads were demultiplexed with Cutadapt 1.18, prohibiting mismatches in the barcode[56]. Then, sequencing read quality was assessed (-q 20) and barcode-trimmed (--cut 12), discarding short reads (--length 16). Poly(A) stretches (>4) at the 3′-end were removed up to 20 tandems without indels or mismatches. Trimmed reads were aligned with Slamdunk map (version 0.3.3)[57] to the reference mouse genome (mm10) using local alignment and allowing up to 100 multimapping reads (-n 100). Aligned reads were subject to a filtering step, retaining only alignments with a minimum identity of 95% and a minimum of 50% of the read bases mapped. Reads ambiguously mapping to more than one 3′UTR were discarded (https://github.com/AmeresLab/UTRannotation). Single-nucleotide polymorphisms (SNPs) were called using VarScan (version 2.4.1) with default parameters and establishing a tenfold coverage cutoff and a variant fraction cutoff of 0.8 (-c 10 -f 0.8). SNPs overlapping T>C conversions needed to present a base quality Phred score of at least >26. T>C conversions were counted with rolling windows and normalized by the T content, coverage of each position and average

per possible transcript 3′UTR. Lastly, the T>C conversion quality was assessed by Slamdunk alleyoop (version 0.3.3)[57].

To calculate half-life, T>C conversions were background-subtracted and normalized to the chase S4U treatment. T>C conversions were fit to an exponential decay nonlinear function using the R minpack.lm 1.2 (ref. [58]). Only 3′UTR half-lives that correlated to a model $R^2 > 0.6$ in all experimental conditions were retrained to measure the mRNA stability of 6,276 transcripts.

## Preprocessing and feature computation for classification of developmentally destabilized transcripts

Initially, 3′UTRs were filtered for $R^2 > 0.6$ and a delta half-life was computed (naive versus primed and *LIN28*-KO versus WT). All retained 3′UTRs were sorted according to their delta half-life. The top and bottom 1,000 3′UTRs with the largest half-life changes were assigned to a specific condition (for example, naive and primed).

Next, we computed a feature matrix for these 3′UTRs. Features were distinguished into seven classes and computed by the gradient boosting machine (GBBM) as described below.

| Feature class | Feature description | Number of features |
|---|---|---|
| Simple features | G+C content, dinucleotide frequency, 3′UTR width and average PhastCon score | 19 |
| miCLIP-seq | Average m⁶A miCLIP-seq scores measured in different ES cell conditions | 6 |
| ES cell iCLIP-seq | Average CLIP-seq scores for various RBPs measured in mouse ES cells | 62 |
| TAIL-seq[9] | poly(A) length and number of A, monoC, monoG, monoT, oligoC, oligoG and oligoT from mouse ES cells | 8 |
| TargetSCAN | Binding predictions for different miRNAs | 11 |
| DeepBind[11] | Average binding prediction of 50-bp tilled 3′UTRs | 105 |
| POSTAR2 (ref. [10]) | CLIP-seq peak scores from various RBPs and conditions | 54 |

All these features were assembled in one matrix, filtered for zero variance features and split into a training set (75%) and a hold-out test set (25%). The training set was used for subsequent model training (one model per set and one model for all features together).

For Fig. 1c–e, we compared the univariate feature importance, as measured by the AUROC for each CLIP or POSTAR2 feature. The AUROCs were computed per feature to distinguish WT versus KO and naive versus primed 3′UTRs. For this analysis, we used the varImp function from the R package caret.

## GBM training and model benchmarking

Hyperparameter search and model training were performed with the R packages caret and GBM[59]. Models were trained for a binary classification task to distinguish WT versus *LIN28*-KO or naive versus primed specific 3′UTRs. Gradient-boosted trees were trained on the features of the training set using fivefold cross-validation repeated five times while performing hyperparameter optimization using grid search. Furthermore, the hyperparameter search was performed as a grid search over the parameters listed below with the aim to maximize the AUROC.

| Parameter | Value |
|---|---|
| Interaction depth | 1, 3, 6 |
| Number of trees | 1, 10, 20, 30, 40, 50, 100, 150, 200, 250, 300, 350, 400, 450, 500 |
| Shrinkage | 0.001, 0.005, 0.01, 0.05, 0.1 |
| Minimum number of observations in terminal node | 10 |

The resulting GBM models were evaluated on the hold-out test set using the AUROC and MCC as performance measures.

The feature importance for each GBM model was computed as the relative influence and the reduction in model performance by shuffling a particular feature. For this analysis, we used the associated GBM R package functions 'relative.influence' and 'permutation.test.gbm'.

## Prediction of pLIN28A-dependent decay from RNA sequence

The longest 5% of pLIN28A upregulated and downregulated 3′UTR sequences were excluded and the remaining 2,060 sequences were N-padded to ensure a uniform length of 3,425 nt across all inputs. Each sequence was then encoded using a one-hot encoding scheme, with the bases A, C, G, T and N represented as binary vectors [1, 0, 0, 0], [0, 1, 0, 0], [0, 0, 1, 0], [0, 0, 0, 1] and [0, 0, 0, 0], respectively. The 3′UTR regulation classes were further transformed into a binary class matrix, where the upregulated and downregulated classes were designated as [1, 0] and [0, 1], respectively.

The sequences were split into fixed training, validation and testing sets (66:17:17 split), preserving class distribution and using fixed sampling. A hybrid model combining a one-dimensional (1D) CNN and RNN was used[23] through TensorFlow 2.12.0. The architecture included an initial 1D convolutional layer (Conv1D) with L1 and L2 regularization and three stacked convolutional blocks each containing an L1-regularized Conv1D layer, followed by a dropout layer, with max-pooling transformation. This was followed by a gated recurrent unit (GRU) and two dense fully connected layers, with the latter using the softmax activation, thereby producing predictive probabilities for each class. The GRU was oriented to process the data in reverse order, such that the nucleotide sequence was still most recently considered by the network, as opposed to the N-padded terminus. Layer normalization and a rectified linear unit (ReLu) activation function were applied following each Conv1D layer, with batch normalization and the ReLu activation function succeeding the GRU layer, dense layer and final softmax output (Extended Data Fig. 4a). Dropout and L1 and L2 regularization were incorporated as the main tools to reduce overfitting of the limited data.

The regularization coefficients (~0.89 for L2, ~7.53 × 10⁴ for L1), dropout probability (~0.34), number of Conv1D blocks (3), Conv1D kernel number (5) and number of neurons (64) were set as per the optimization results from Optuna version 3.1.0 (ref. [60]), optimizing for best validation set accuracy. The Adam optimizer was employed with the learning rate set to $5 × 10^{-5}$, beta_2 set to 0.98 and clipnorm set to 0.5, to additionally prevent overfitting. The loss was computed using the categorical cross-entropy. Training was performed on batches of 64 examples from the training set with early stopping, monitoring validation loss with a patience of 25 epochs and restoring the best weights upon stopping. The model's final performance was evaluated on the test set on the basis of accuracy, AUROC score, precision, recall and $F_1$-score metrics. The code for model training and subsequent feature importance analyses is available from GitHub (https://github.com/ulelab/LIN28A_RNPreassembly_bioinformatics).

## SHAP importance analysis and characterization of 3′UTR features predictive of pLIN28A-dependent decay

The impact of individual features in the 3′UTRs on the predictions of the neural network was assessed using SHAP version 0.35.0 (ref. [24]) with a gradient-based explainer. This process generates the SHAP values at the nucleotide level, which quantify the contribution of each hypothetical feature (in this case, each of the four nucleotides) to each prediction class (upregulated versus downregulated). For each evaluated sequence, this results in an array in which each row represents one nucleotide within a sequence and the four columns represent importance scores for each of the four nucleotides (that is, model features). All sequences were used for the SHAP analysis of each class.

Following our analysis with SHAP, we used the tool TF-MoDISco Lite version 2.0.0 to identify important sequence motifs in all 3′UTRs for the downregulated prediction class[25] (Extended Data Fig. 4e). TF-MoDISco works by first extracting short spans of nucleotide sequence (that is, seqlets) with high absolute importance scores and then dividing them into positive (that is, the seqlet positively contributes to the prediction) and negative (that is, the seqlet negatively contributes to the prediction) metaclusters. For each metacluster, a similarity matrix is calculated and used for the identification of underlying sequence motifs by Leiden clustering. We ran TF-MoDISco Lite with the maximum number of seqlets per metacluster, the number of Leiden clusterings to perform with different random seeds and the window length surrounding the peak center for motif discovery set to 1,000, 10 and 150 nt, respectively.

To further understand the important trimer motifs in model prediction, we removed the N-padding and summarized the importance of individual trimers across 3′UTR sequences. By applying a sliding window of 3 nt using a step of 1 nt to each scored nucleotide sequence, we computed the mean importance score for each possible trimer at each sequence position. Next, each sequence was grouped into 100 bins and the scores across all nucleotide positions within one bin were summed and normalized by bin length. Finally, the mean importance score of each trimer was computed within each bin (across all evaluated sequences) and summarized in the metaprofile for the downregulated prediction class (Fig. 3a). Specific trimers with high positive or negative importance scores for the downregulated prediction class were highlighted in color against a background of all possible trimers.

To quantify the contributions of individual nucleotides at relative positions within the 3′UTRs on model predictions (Extended Data Fig. 4c), we normalized the nucleotide-level importance scores for the downregulated prediction class across all transcripts to their respective 3′UTR lengths. The normalized scores were subsequently split into 100 bins according to their relative position in the 3′UTR and scores within each bin were summed. Lastly, the mean score in each bin was calculated across all evaluated 3′UTRs and these summarized scores were visualized as a heat map (Extended Data Fig. 4c). Within this heat map, each cell represents the summarized importance score of a specific nucleotide within a particular bin.

## iCLIP

We used iCLIP to identify a repertoire of RBP binding sites in PS cells following the iiCLIP protocol[61] with some modifications. Cells were ultraviolet (UV) cross-linked on ice and then lysed in RIPA and CLIP lysis buffer. Then, 0.4 U of RNaseI and 4 U of Turbo DNase were added per 1 ml of cell lysate at 1 mg ml⁻¹ protein concentration for RNA fragmentation. Negative controls (no UV) were prepared. Antibodies (anti-GFP polyclonal antibody, Thermo Fisher Scientific, A6455; anti-LIN28A polyclonal antibody, Cell Signaling, 3978; anti-PABPC1, Abcam, ab21060 and Proteintech, 10970-1-AP; PABPC4, Proteintech, 14960-1-AP) were coupled to magnetic protein G beads used to isolate protein–RNA complexes and RNA was ligated to a preadenylated infrared-labeled IRL3 adaptor[62] with the following sequence:

*/5rApp/AG ATC GGA AGA GCG GTT CAG AAA AAA AAA AAA /iAzideN/ AA AAA AAA AAA A/3Bio/*

The complexes were then size-separated by SDS–PAGE, blotted onto nitrocellulose and visualized by Odyssey scanning. For the multiplexed sample, one replicate was run in parallel with the IRL3 adaptor to allow quality control of the RNP complex on the membrane and to help with cutting of the bands. RNA was released from the membrane by proteinase K digestion and recovered by precipitation. Complementary DNA (cDNA) was synthesized with Superscript IV reverse transcriptase (Life Technologies) and AMPure XP bead purification (Beckman-Coulter) and then circularized using Circligase II (Epicentre) followed by AMPure XP bead purification. After PCR amplification, libraries were size-selected with AMPure beads (if necessary by gel purification) and quality-controlled for sequencing. Libraries were sequenced as single-end 100-bp reads on an Illumina HiSeq 4000.

## iCLIP data analysis

Sequencing data produced by iCLIP experiments were processed on the iMaps Genialis server (https://imaps.genialis.com/). Briefly, experimental barcodes were removed with Cutadapt[56] and sequencing reads were aligned with STAR[63] to the mouse genome build (GRCm38. p5 GENCODE version 15 annotation) allowing two mismatches. Unique molecular identifiers (UMIs) were used to distinguish and remove the PCR duplicates. The nucleotide preceding each sequencing read was assigned as the cross-link event.

iCLIP data for LIN28A WT (in 2iL-treated and FGF2-treated cells), LIN28A S200A (in FGF2-treated cells) and PABPC1 and PABPC4 (in *LIN28A*-KO cells with and without LIN28A overexpression) were analyzed on the iMaps Goodwright server (https://imaps.goodwright. com/). First, reads were demultiplexed using Ultraplex[64], using default settings, and barcodes were removed using Cutadapt version 3.4 (ref. 65). Reads were then premapped to ribosomal RNA, transfer RNA sequences referred to as small RNA (smRNA) and smRNA with Bowtie[66]. Next, reads that did not map with Bowtie were aligned with STAR version 2.7.9a (ref. 63) using the mouse genome build (GRCm39 GENCODE M28 annotation). This was followed by the removal of PCR duplicates using UMI-tools[67] and identification of cross-link events, as described above. Peaks of the cross-linking signal were identified with Clippy version 1.4.1 (ref. 68) and used together with the cross-link sites to find enriched motifs with positionally enriched *k*-mer analysis (PEKA) version 1.0.0 (ref. 48) using the default settings. For Clippy and PEKA, the GENCODE primary assembly annotation M28 was filtered to retain only entries with a transcript support level of 1 or 2, in genes where such transcripts were available, and used to produce a segmentation file with the get_segments function from the iCount tool[69]. All files generated by running the analysis pipeline are available from the iMaps Goodwright webserver for analysis of CLIP data (see https:// imaps.goodwright.com/collections/882635250203/ and https://imaps. goodwright.com/collections/340215254997/ for LIN28A and PABPC1/4 iCLIPs, respectively) and on the updated Flow webserver (see https:// app.flow.bio/projects/882635250203/ and https://app.flow.bio/projects/340215254997/ for LIN28A and PABPC1/4 iCLIPs, respectively). We archived key data analyzed in this study, specifically, cross-link sites, peaks and motif enrichments from PEKA, on Zenodo (https:// doi.org/10.5281/zenodo.10054231)[70]. The code and settings used in the iCLIP analysis pipeline (release v0.30) can be viewed at https://github. com/goodwright/imaps-nf, and are also archived on Zenodo (https:// doi.org/10.5281/zenodo.10054231)[70].

## Identification of motif groups from CLIP data

For the identification of enriched motif groups, we obtained PEKA results in 3′UTRs (files ending in *5mer_distribution_UTR3.tsv, accessible from https://imaps.goodwright.com/collections/882635250203/) for all LIN28A WT (in 2iL-treated and FGF2-treated cells) and LIN28A S200A (in FGF2-treated cells) iCLIPs, except for LIN28A-S200A_ESC_ LIF-CHIR-FGF0220626_MM_2, which was excluded from subsequent analyses because of low read coverage. First, we combined enriched *k*-mers ($P < 0.05$) from all samples into one group of unique *k*-mers ($n = 93$), which we then clustered on the basis of their sequence similarity and their ranking in PEKA. To achieve this, we computed Euclidean distances between *k*-mer ranks in PEKA and Jaccard distances between *k*-mer sequences. To obtain sequence distances, each *k*-mer sequence was first converted into a list of all possible substrings with length less than *k* (for example, a trimer 'UGA' would be converted into 'U', 'G', 'A', 'UG' and 'GA'). Then, Jaccard similarity was calculated on sets of substrings for each pair of *k*-mers. The Jaccard similarity is a quotient of the number of shared substrings between two *k*-mers and the number of all

substrings in the union. Finally, Jaccard similarities were converted into distances by subtracting the similarity values from 1. Next, we applied standard scaling to each of the resulting distance matrices to account for differences in variance between the two metrics, followed by min–max scaling. Normalized matrices were combined with Pythagorean addition and used to cluster $k$-mers using 'scipy.hierarchy', with correlation to compute distances and the UPGMA (unweighted pair group method with arithmetic mean) algorithm to perform clustering. Next, the dendrogram was plotted with 'scipy.hierarchy.dendrogram' and the tree was cut into four clusters (Extended Data Fig. 5a) to obtain the following motif groups: (1) an AGGG motif group, representative of the ZnF domain (Fig. 3d); (2) the UGGG motifs; (3) the GAU motif group, bound by the coding sequence (CDS) (Fig. 3e); and (4) the AUU motif group (Fig. 3f). Finally, we combined AGGG and UGGG into one motif group (WGG), as they exhibited similar $k$-mer sequences and enrichment patterns in PEKA (Extended Data Fig. 5a). The $k$-mer logos for resulting motif groups were plotted as described in the literature[48]. The source code for $k$-mer clustering and for generation of $k$-mer logos is available on Zenodo[71].

For the quantification of significantly enriched A-rich and U-rich 5-mers at 3'UTR cross-link sites of PABPC1 and PABPC4 (Extended Data Fig. 8d), we obtained enriched 5-mers for each sample ($P < 0.05$) from files ending in *5mer_distribution_UTR3.tsv (accessible from https://imaps.goodwright.com/collections/340215254997/). Then we defined U-rich and A-rich $k$-mers as those with three or more Us or As, respectively, and computed the proportion of $k$-mers in each of these groups relative to all significantly enriched $k$-mers.

### Metaprofiles of motif coverage around cross-link sites

We used the AGGG, UGGG, GAU and AUU motif groups to plot metaprofiles of motif coverage around cross-link sites in Fig. 3d–f with a script described in the literature[48], which is available on Zenodo[72]. The script visualizes the coverage of a $k$-mer group around cross-link sites within specified transcriptomic regions. The coverage is expressed as a percentage of cDNAs that have a $k$-mer from the group aligned to a specific position relative to the cross-link site. For this study, we used cross-link sites in the 3'UTR regions as input and computed motif coverage in 601-nt-long regions, centered on the cross-link sites. The cDNA scores of input cross-links were capped at 20 to avoid any excessive contributions of a few cross-link sites with disproportionately high cDNA scores. Finally, coverage was smoothed with a rolling mean using a triangular window of 20 nt and plotted.

### Motif-based binding-site assignment

We defined LIN28A binding sites corresponding to WGG, GAU and AUU motif groups using the approach of motif-based binding-site assignment, which was adopted from the approach used previously by Hallegger et al.[73]. First, cross-link sites from all LIN28A WT (in 2iL-treated and FGF2-treated cells) and LIN28A S200A (in FGF2-treated cells) iCLIP samples, except for LIN28A-S200A_ESC_LIF-CHIR-FGF0220626_MM_2, were merged by summing up cDNA counts at overlapping positions. Then, binding sites were assigned separately for each motif group where $k$-mers from a given group were located at relevant positions in the range of ±20 nt around cross-link sites. Relevant positions for each $k$-mer were determined by combining relevant positions identified by PEKA (prtxn, available in files ending in *5mer_distribution_UTR3.tsv) from all LIN28A iCLIP samples. The script for motif-based binding-site assignment is available on GitHub[74]. Motif-based binding sites of LIN28A are shown as auxiliary tracks in Extended Data Fig. 7a,d,e.

### Visualization of nucleotide distribution around PAS in downregulated and upregulated or control genes

For the generation of the line plot of nucleotide composition 100 nt upstream to 20 nt downstream of the PAS for downregulated and upregulated or control transcripts (Extended Data Fig. 4d), we iterated over the 3'UTR sequences of each group and extracted the location of the terminal canonical PAS ('AATAAA') for each sequence; sequences without the canonical PAS or with the PAS located less than 20 nt from the 3'UTR termini were removed. The number of sequences excluded and retained for the downregulated group was 310 and 873, respectively. Meanwhile, for the upregulated or control group, there were 1,154 excluded sequences and 2,539 kept sequences. For the remaining valid sequences, we summed the instances of each nucleotide at every position within the window of 100 bp downstream and 20 bp upstream of the PAS. These nucleotide counts were subsequently normalized at each location to represent their percentage of the total number of considered sequences. The scores were smoothed by a rolling mean with a window of 5 nt and plotted as a separate plot for each transcript group.

### Visualization of iCLIP data in 3'UTRs of naive genes

The visualization of iCLIP data within the 3'UTRs of *Tfcp2l1*, *Zfp281* and *Esrrb* (Extended Data Fig. 7a,d,e) was performed with a clipplotr tool[75], using the normalization of iCLIP cDNA counts at a given cross-link site by the experimental library size. Normalized counts were smoothed with a rolling mean across a window of 50 nt and plotted across the regions of interest. Auxiliary tracks were added below the cross-linking signal to represent LIN28A binding sites corresponding to WGG, GAU and AUU motif groups, the location of the AAA trimer and the location of the canonical PAS 'AAUAAA'.

### Estimation of gene-level expression

Expression levels for each gene were obtained by summing up TPM values of transcripts with the same stable Ensembl gene identifier and then averaging these sums across replicate 3'-end sequencing experiments. Transcript TPM values were obtained from 3'-end sequencing data with Salmon, as described in 'QuantSeq 3' mRNA-seq processing and analysis'. These gene-level expression estimates are referred to in text as the gene-level TPM.

### iCLIP cross-linking profiles for 200 downregulated 3'UTRs

For the generation of iCLIP cross-linking profiles for LIN28A WT (in 2iL-treated and FGF2-treated cells) and PABPC4 (in *LIN28A*-KO cells with and without LIN28A overexpression) on 200 medium-length (754–1,168 bp) downregulated 3'UTRs, the iCLIP cross-linking counts and the location of the terminal PAS corresponding to 'AAUAAA' were extracted for each 3'UTR. Next we applied a moving average with a window size of 10, thereby smoothing out the signal. In each 3'UTR, the smoothed signal was min–max normalized, highlighting positional information shown in the heat maps (Fig. 4a,c). Additionally, the location of PAS was marked with a green dot. The same process was applied to produce iCLIP cross-linking profiles for all downregulated 3'-UTRs. After excluding the longest 5% for better interpretability, we were left with 1,123 sequences (Extended Data Fig. 6). For quantitative comparisons of RBP binding to 3'UTRs, the sum of the CLIP signal for each 3'UTR was computed and normalized by length and gene-level TPM to adjust for expression levels in respective experimental conditions. This quantitative comparison was plotted together with RBP occupancy profiles, enabling a comparative analysis of different conditions and RBPs.

### Metaprofiles of normalized cross-link coverage

To generate the metaprofiles of library-normalized and expression-normalized cross-link coverage upstream of 3'UTR termini (Extended Data Fig. 5c and Extended Data Fig. 8a,b), around the canonical PAS (Fig. 4b,d and Extended Data Fig. 8c) and around peak centers (Extended Data Fig. 5b,c), we first computed raw cross-link coverage for individual iCLIP samples in regions of interest with the bedtools coverage utility. Next, raw cDNA coverage was normalized by sequencing depth for each sample to CPM (cross-links per million) values, which was followed by normalization within each gene to its expression level in a relevant experimental condition, using gene-level TPM values,

obtained as described above. This yielded a set of quantified genomic regions with cross-link coverage at each position expressed as CPM per TPM. Then, we smoothed the expression-normalized coverage within each quantified region, using a rolling mean across 20 nt and a triangular window. A smoothed value was assigned to the position located at the center of the window. Missing values resulting from smoothing at the 5′ and 3′ ends of the region were replaced with the closest valid observation. Finally, we computed the mean of smoothed coverages across all regions of interest with a 95% confidence interval at each position within the region.

To generate metaprofiles of cross-linking coverage of LIN28A WT (in 2iL-treated and FGF2-treated cells) and PABPC4 (in *LIN28A*-KO cells with and without LIN28A overexpression) across full-length transcripts, cross-link coverage was obtained for each transcript across its 5′UTR, CDS and 3′UTR regions (Fig. 4e). Raw cross-link counts were normalized by library size and gene-level TPM to adjust for expression. Each transcript region was divided into 100 bins, summing the normalized cross-linking counts in each bin and normalizing them to bin length. Obtained values were then averaged across all genes for each transcript region to create a metaprofile representative of the average RBP binding across full-length transcripts in the downregulated and upregulated or control groups. Each bin was plotted as a dot with an overlaid smoothed line derived from a rolling mean with a window of ten bins.

For the metaprofiles upstream of 3′UTR termini (Extended Data Fig. 5c and Extended Data Fig. 8a,b), the 3′UTRs were filtered to a minimum length of 500 nt, resulting in the inclusion of 1,760 control, 768 downregulated and 623 upregulated 3′UTRs. For metaprofiles around the PAS (Fig. 4b,d and Extended Data Fig. 8c), the 3′UTRs with the canonical PAS sequence 'AATAAA' and a length of minimum 300 nt were included in the analysis. In the case of multiple canonical PASs within a 3′UTR, only the PAS closest to the 3′UTR terminus was used. This included 1,739 PASs from control, 831 from downregulated and 633 from upregulated genes.

For the metaprofiles shown in Extended Data Fig. 7c, we defined the regions as 100 nt upstream and downstream from the centers of PABPC peaks that were located either within the last 200 nt of the 3′UTRs or elsewhere in the 3′UTR. PABPC peaks were obtained by merging book-ended or overlapping Clippy peaks of PABPC1 or PABPC4 iCLIPs in *LIN28A*-KO cells with and without LIN28A overexpression, using 'bedtools merge'. For the metaprofiles shown in Extended Data Fig. 7b, we defined the regions as 100 nt upstream and downstream from the centers of LIN28A peaks located in the 3′UTRs that overlapped or did not overlap with PABPC peaks. A minimum overlap of 1 nt was used and peak overlap was identified with 'bedtools intersect'. LIN28A peaks were obtained by merging book-ended or overlapping Clippy peaks of LIN28A WT (in 2iL-treated and FGF2-treated cells) and LIN28A S200A (in FGF2-treated cells), using 'bedtools merge'; PABPC peaks were obtained by merging PABPC1 and PABPC4 peaks from *LIN28A*-KO cells with and without LIN28A overexpression.

To compare the distribution of cross-link signal of LIN28A WT in FGF2-treated cells to that in 2iL-treated cells in naive genes (Extended Data Fig. 5d), we computed the raw cross-link coverage for merged samples in 500-nt regions upstream of 3′UTR termini. Then, we quantified the raw cross-linking signal in bins of 20 nt and converted the cross-link counts into the percentage of counts within each region to enable a direct comparison between the two conditions. The value of 1 was then added to each bin to avoid division by 0. We proceeded to calculate the $log_2$ fold change between the percentage of cross-links in the FGF2-treated condition relative to the 2iL-treated condition for each bin in each naive gene. Fold changes obtained across all evaluated genes were then plotted as a bar plot for each bin. Naive genes included in this analysis were filtered to have a minimum 3′UTR length of 500 nt and a sufficient expression level (≥5 TPM in *LIN28A*-KO or WT in the FGF2-treated condition). This resulted in the inclusion of 16 of the 22 genes of the naive regulon.

## Trimer valency and motif coverage in 3′UTR regions

To assess the number of trimers in regions 100 nt upstream of the PAS for genes belonging to the control, downregulated and upregulated groups (Extended Data Fig. 5f), we first obtained genomic sequences of those regions using 'bedtools getfasta'. For each gene, only the PAS closest to the 3′UTR termini with the canonical 'AATAAA' sequence was used, which resulted in the inclusion of 2,126 PAS in the control, 1,002 in the downregulated and 778 in the upregulated group. Next, we located the motifs of interest in these regions with 'seqkit locate' (version 2.3.1) and wrote their genomic coordinates into a BED file. Then, we counted the number of relevant trimers in individual 3′UTRs and plotted the distribution of counts for control, downregulated and upregulated genes.

To compute the density of motif coverage (Fig. 4b and Extended Data Fig. 5c,e), we first obtained genomic sequences of relevant regions using 'pybedtools.sequence' (version 0.9.0). Next, we scanned these sequences with a window of length 5 and checked whether the sequence in a window corresponded to any of the 5-mers in the given motif group (namely, AUU, WGG or GAU motif group; see Methods, 'Identification of motif groups from CLIP data'; Extended Data Fig. 5a). If the sequence in a window matched with a 5-mer in a motif group, the nucleotides in the window received a score of 1. Next, we binned the regions into bins of 10 nt (Fig. 4b and Extended Data Fig. 5e) or 20 nt (Extended Data Fig. 5c) and computed the percentage of covered nucleotides in each bin. Finally, we computed the average bin coverage across all evaluated regions and plotted these values in the form of a heat map. The 3′UTRs included in each heat map correspond to those shown in the associated metaprofiles.

## Abundance of protein binding in 3′UTR regions

To compare protein abundances in the terminal regions of 3′UTRs between groups of differentially regulated genes (Extended Data Fig. 8h), we computed cDNA counts in merged iCLIP samples within regions of interest. These counts were then normalized by the sequencing depth of merged samples and expression as described in 'Metaprofiles of normalized cross-link coverage'. For the comparison of protein binding at the 3′UTR termini, we evaluated the regions spanning 300 nt upstream of the 3′UTR terminus and 300 nt downstream of the stop codon. For this analysis, 3′UTRs were filtered to a minimum length of 800 nt, the minimum average expression level of genes in all conditions was set to 1 TPM and both the starting and terminal 3′UTR regions were required to have at least five cDNAs mapped to them in all LIN28A, PABPC1 and PABPC4 iCLIP samples. These criteria excluded 3′UTRs that were too short to represent their starting and terminal regions with 300 nt, genes that were lowly expressed and genes that did not exhibit a sufficient iCLIP signal for studied RBPs. These filters yielded 150 3′UTRs in the downregulated, 284 3′UTRs in the control and 46 3′UTRs in the upregulated group.

## Modeling of protein structure

For Fig. 3d–f, we modeled a crystal structure of LIN28A in complex with *let-7*d microRNA pre-element (Protein Data Bank (PDB) 3TRZ)[30] using ChimeraX[76]. The protein is displayed in light-green color and the RNA chain is displayed in gray. Nucleotides interacting with the protein are represented with sticks and color-coded (A in green, G in yellow and U in red). Hydrogen bonds between the protein and the colored RNA motifs are shown as dashed blue lines. Planes of aromatic rings with a potential to form $\pi$–$\pi$ stacking interactions are shown with a mesh.

## Analysis of poly(A) tail length

Libraries for direct RNA sequencing were prepared following the direct RNA sequencing protocol for MinION (SQK-RNA002) with 500 ng of total RNA as input. Sequencing was performed on FLO-MIN106D flow cells. Raw reads were base-called using Guppy version 6.0.0 with the high accuracy model (rna_r9.4.1_70bps_hac.cfg). Mapping was

performed using minimap2 (ref. 77) with the parameters '-ax map-ont -k14 -secondary=no' to the GENCODE M28 transcriptome and processed with SAMtools version 1.6 to filter for mapping quality score of 60 and discard reads mapping to the reverse strand. The poly(A) tail length for each read was estimated using the Nanopolish 0.14.0 poly(A) function and only length estimates with the quality control tag reported as a pass were considered in subsequent analyses. Genes with more than 20 reads in all samples were retained and their poly(A) tail length was compared across downregulated, control and upregulated groups (Fig. 2h). In total, 1,116 transcripts were analyzed: 638 in the control, 133 in the upregulated and 345 in the downregulated group. Because of substantial inter-replicate variability in basal poly(A) length when performing Nanopore direct RNA sequencing, each sample was plotted independently. The $y$ axis in Fig. 2h is limited to show values between 60 and 160 for clarity; however, all values were included in the calculation of means, interquartile ranges and statistics.

### Statistics and reproducibility
No statistical methods were used to predetermine sample size and the experiments were not randomized. The number of replicates used in each experiment is described in the figure legends and/or this section, as are the statistical tests used including the adjustment methods and α values for adjusted $P$ values. Statistical tests were selected appropriately to the analyzed data, considering normality, variance, independence (paired or independent tests) and direction of effect (two-sided or one-sided tests). To compare two independent samples, with approximately normal distribution and approximately equal variance, we used two-sided two-sample $t$-tests. When the data were approximately normally distributed but the equal variance criterion was not met, we used two-sided Welch's $t$-tests. When the data did not meet the criteria for normality and/or equal variance, we used the two-sided Mann–Whitney–Wilcoxon rank-sum test. We occasionally used the non-parametric two-sided Mann–Whitney–Wilcoxon rank-sum test in place of the two-sample $t$-test and Welch's $t$-test because of fewer underlying assumptions. In the figures and figure legends, we indicate the comparisons of interest and report the exact $P$ values when greater than $10^{-4}$. For $P$ values lower than $10^{-4}$, we label them as $<10^{-4}$; for $P$ values lower than $10^{-10}$, we label them as $<10^{-10}$.

### Reporting summary
Further information on research design is available in the Nature Portfolio Reporting Summary linked to this article.

## Data availability
Sequencing data related to iCLIP, QuantSeq and Nanopore direct RNA sequencing experiments for LIN28A WT in 2iLIF-treated and FGF2-treated cells, LIN28A S200A in FGF2-treated cells and iCLIPs of PABPC1 and PABPC4 (in *LIN28A*-KO cells with and without LIN28A overexpression) are available from the European Nucleotide Archive under accession code PRJEB60519. SLAMseq data and QuantSeq data of LIN28A–GFP overexpression in *LIN28A*-KO cells and LIN28A–GFP iCLIP experiments can be retrieved from the Gene Expression Omnibus under accession code GSE169555. Processed iCLIP data are available for LIN28A (https://imaps.goodwright.com/collections/882635250203/ and https://app.flow.bio/projects/882635250203/) and PABP (https://imaps.goodwright.com/collections/340215254997/ and https://app.flow.bio/projects/340215254997/). Source data are provided with this paper.

## Code availability
Code for bioinformatic analyses is available on GitHub (https://github.com/ulelab/LIN28A_RNPreassembly_bioinformatics), together with YAML environment files (containing versioned packages used for the analysis). Code and dependencies are also archived on Zenodo (https://doi.org/10.5281/zenodo.10054297) (ref. 78). iCLIP-seq data

were processed on the iMaps Goodwright webserver (https://imaps.goodwright.com/) and the links to the relevant collections are specified in Methods. The code and settings used in the iCLIP analysis pipeline (release version 0.30) can be viewed on GitHub (https://github.com/goodwright/imaps-nf) and are also archived on Zenodo (https://doi.org/10.5281/zenodo.10054231) (ref. 70).

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

## Acknowledgements

We thank G. Daley and K. Tsanov (Harvard University) for advice and generous reagent gifts. We thank A. Chakrabarti, C. Capitanchik and I. Iosub for detailed feedback on the paper. For technical support, we are grateful to F. Capraro, U. Janjoš, R. Arora and Crick science technology platforms for advanced sequencing, advanced light microscopy and flow cytometry. We gratefully acknowledge the HPC RIVR consortium (www.hpc-rivr.si) and EuroHPC JU (eurohpc-ju.europa.eu) for funding this research by providing computing resources of the HPC system Vega at the Institute of Information Science (www.izum.si). This research was funded in part by the Wellcome Trust (Sir Henry Wellcome Fellowship 218672/Z/19/Z to M.M.), the European Union's Horizon 2020 research and innovation program (835300-RNPdynamics to J.U.), the joint Senior Wellcome Trust Award (215593/Z/19/Z to J.U. and N.M.L.) and the Francis Crick Institute, which receives its core funding from Cancer Research UK (FC001002), the UK Medical Research Council (FC001002) and the Wellcome Trust (FC001002). For the purpose of open access, we have applied a CC BY public copyright licence to any author-accepted paper version arising from this submission. The funders had no role in study design, data collection and analysis, decision to publish or preparation of the manuscript.

## Author contributions

M.M. and J.U. conceptualized and jointly supervised the study. M.M. coordinated data analysis and designed and performed most experiments, with contributions from Ž.V., F.C.Y.L. and E.G., who was supervised by D.t.B. K.K. led the bioinformatics of iCLIP and QuantSeq data and RNA feature extraction. S.S. and J.N. performed deep learning on CLIP datasets and mRNA decay 3′UTRs, respectively, with input from N.M.L. and K.K. I.R.d.l.M., K.K., M.M. and R.F. analyzed the QuantSeq sequencing data. M.M. and J.U. wrote the paper, with contributions from K.K. and all coauthors. All authors agreed with the content and consented to submit the paper with their contributions.

## Funding

## Competing interests

The authors declare no competing interests.

## Additional information

**Extended data** is available for this paper at https://doi.org/10.1038/s41594-024-01363-x.

**Correspondence and requests for materials** should be addressed to Miha Modic or Jernej Ule.

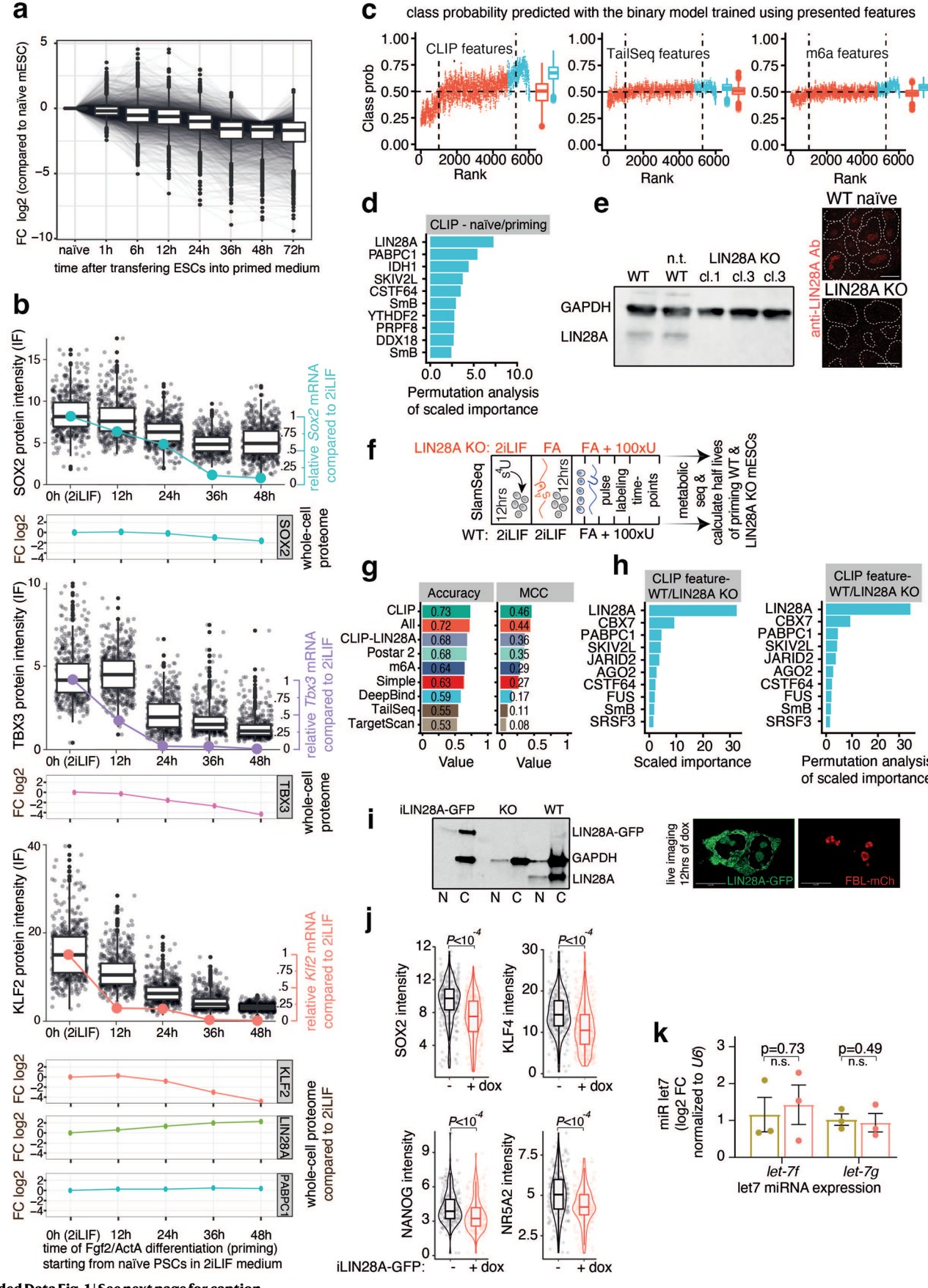

**Extended Data Fig. 1 | See next page for caption.**

**Extended Data Fig. 1 | Earliest developmental gene and protein expression changes during naïve-to-primed transition.** (**a**) Temporal dynamics of relative mRNA levels (compared to 2iLIF) of all genes downregulated during hours of PSC naive-to-primed differentiation (n = 20936). (**b**) Relative mRNA levels and proteome levels[7] (compared to 2iLIF) indicated by line plots and fluorescence units of naïve markers SOX2, KLF2, TBX3, NR5A2 (n = >200) indicated by boxplots during hours of naïve-to-primed differentiation. Below: temporal dynamics of relative protein levels during naïve-to-primed transition of top-ranking RBPs predicting transcripts with decreased stability, LIN28A and PABPC1[7]. (**c**) Ranking of all half-life transcripts from UP (half-life higher in nPSCs) towards DOWN (half-life lower in nPSCs) compared with the class probability predicted with the binary model trained using features as presented in Fig. 1d. Class probabilities were smoothed using the running mean (k = 10). (**d**) Permutation analysis of feature importance impacting mRNA stability comparing naïve and priming PSCs. (**e**) Left: Western blot analysis of LIN28A and histone H3 in LIN28A WT nPSCs, non-targeted LIN28A clone and independent verified LIN28A KO clones. Right: Representative immunofluorescence photomicrographs of WT nPSCs and LIN28A KO counterpart. (**f**) Protocol for SLAMseq measurements of mRNA half-life in LIN28A KO vs WT priming PSCs, and Machine learning framework for classifier implementation (See methods). FA stands for Fgf2 + ActA medium during naïve-to-primed conversion. (**g**) Accuracy and Matthews correlation coefficient (MCC) in predicting the half-life change of test-set mRNAs comparing LIN28A KO vs WT priming PSCs trained with indicated features. (**h**) Left: Feature importance ranking computed as relative influences, for best performing classifiers of PSC CLIP datasets comparing LIN28A KO to WT priming PSCs. Right: Permutation analysis of feature importance by shuffling a particular feature. (**i**) Left: Western blot analysis of LIN28A, LIN28A-GFP and GAPDH in wild-type EpiSCs, doxycycline-treated and -untreated iLIN28A-GFP nPSCs. Right: Live cell representative fluorescence micrographs of nPSCs expressing endogenous FBL-mCherry and dox-inducible LIN28A-GFP overexpression in 2iLIF, validating the cytoplasmic accumulation of LIN28A-GFP protein. (**j**) Fluorescence arbitrary units of immunostained naïve markers SOX2, KLF4, NANOG, NR5A2 comparing 20 hrs doxycycline-induced and untreated iLIN28AGFP PSCs. n = >200. Two-sided two-sample t-test; ***P = < 0.0001. (**k**) Bar-chart depicting relative levels of miR let-7 in iLIN28A-GFP nPSCs with and without doxycycline treatment (n = 3).

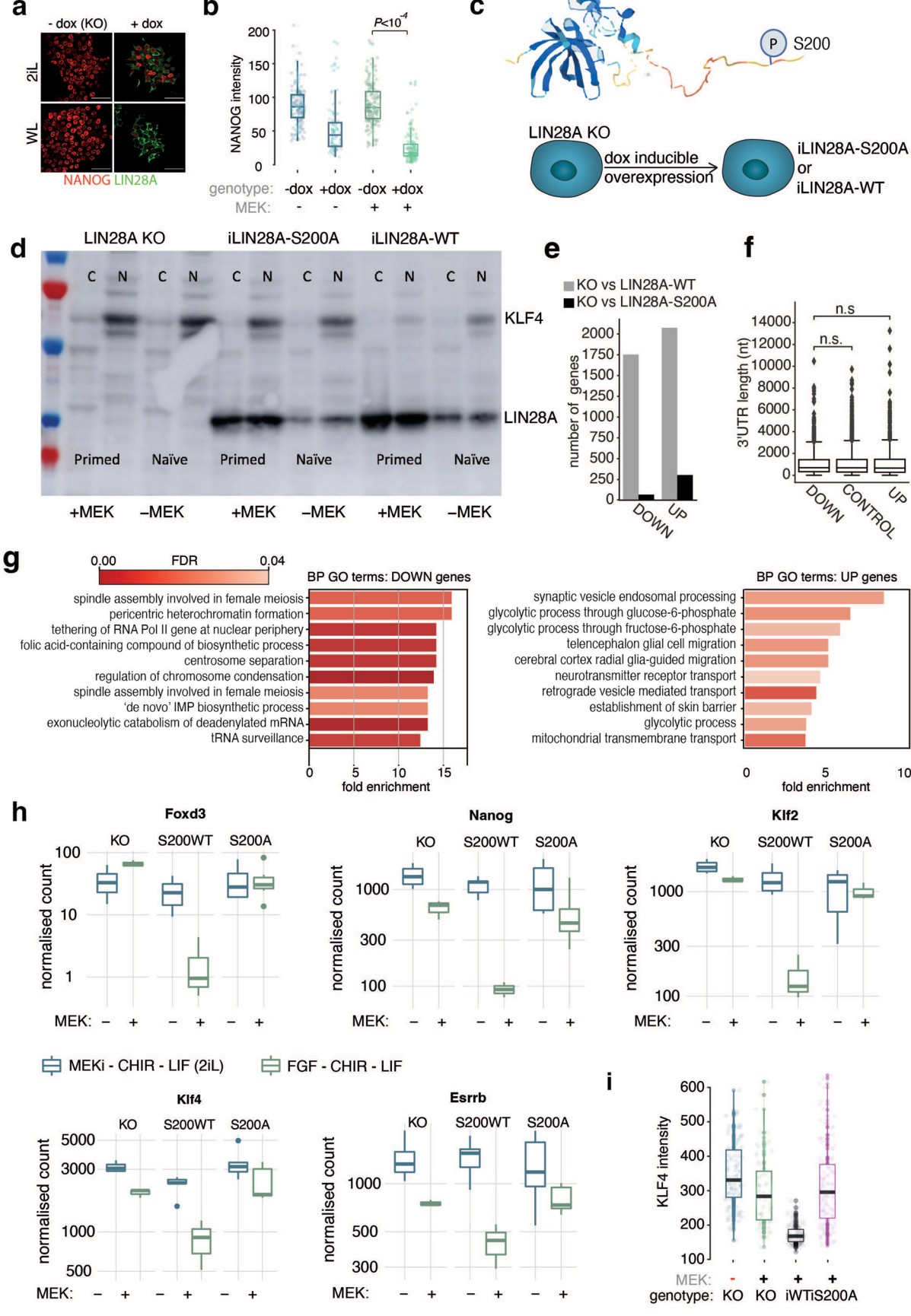

**Extended Data Fig. 2 | See next page for caption.**

**Extended Data Fig. 2 | MEK/ERK-driven LIN28A S200 phosphorylation and its role in the clearance of naïve pluripotency regulon.** (**a-b**) Representative immunofluorescence photomicrographs (**a**) and fluorescence arbitrary units of immunostained naïve marker NANOG (**b**), comparing 6 hrs doxycyclin-induced and untreated iLIN28A in naïve and MEK/WNT conditions. The size of all samples was =>200 and Welch's t-test was used to assess whether KLF4 measurements upon LIN28A dox-induction and MEK activation significantly differ in their mean. ***P = < 0.0001 (**c**) Schematic visualisation of AlphaFold predicted LIN28A structure entailing C-terminal intrinsically disorder region, where S200 is located. Below, schematic of LIN28A knockout cells with doxycycline-inducible LIN28A-WT or LIN28A-S200A. (**d**) Fractionation western blot analysis of LIN28A and naïve pluripotency factor KLF4 in LIN28A knockout cells with doxycycline-inducible LIN28A-WT or LIN28A-S200A and the same cells left untreated as LIN28A KO. N and C indicate nucleus and cytoplasm, respectively. (**e**) Number of differentially regulated genes when comparing 3'end RNA sequencing data for LIN28A-WT (grey) or LIN28A-S200A overexpression (black) to KO, in the FGF2 treated condition. (**f**) Boxplots display the length of 3'UTRs for genes in DOWN, Control, and UP group. For each gene, representative 3'UTR was annotated based on the most abundant mRNA isoform in cells expressing LIN28A-WT cultured in 2iL medium (Methods). Welch's t-test was performed to assess whether the groups of genes significantly differ in mean length of 3'UTRs. (**g**) Gene Ontology annotation of DOWN and UP genes. Bar graph representing fold change of GO annotation of biological processes. (**h**) Normalised Quantseq counts comparing 6 hrs doxycyclin-induced and untreated iLIN28A in naïve and FGF/MEK/WNT conditions for indicated transcripts, without or with MEK activation, respectively (n = >3). (**i**) Immunofluorescence quantification of KLF4 abundance in LIN28A KO, iLIN28A-WT and iLIN28A-S200A in Wnt/LIF media with or without MEK/ERK activation. The size of all samples was =>200.

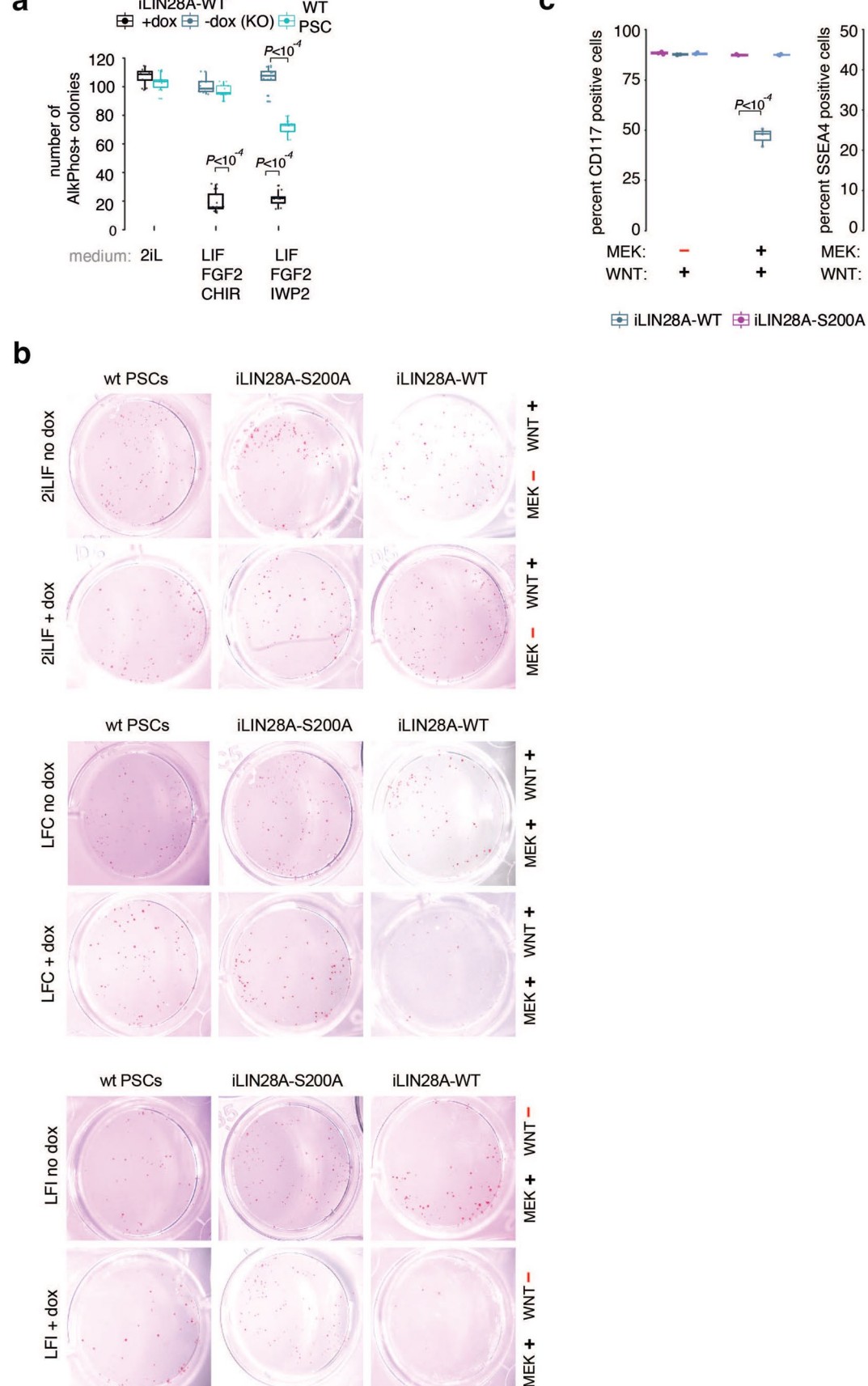

**Extended Data Fig. 3 | See next page for caption.**

**Extended Data Fig. 3 | Cellular surface markers and alkaline phosphatase stainings dependent on MEK/ERK-driven LIN28A S200 phosphorylation.** (**a**) Quantifications of alkaline phosphatase positive wild-type, iLIN28A-WT and iLIN28A-S200A PSC colonies after 24 h in 2iLIF, LFW (Lif/Fgf2/Wnt), and LFI (Lif/FGF2/IWP2) with and without dox induction. Dots represent replicates from three independent experiments, Two-sided t-test; ***P = < 0.0001.

(**b**) Representative images of alkaline phosphatase stainings as quantified in Fig. 2i. (**c**) Quantification of flow cytometry analyses of PSCs that were treated with doxycycline (iLIN28A-S200A or iLIN28A-WT), or left untreated (LIN28A KO), and immunostained for naïve pluripotency marker CD117 and primed pluripotency marker SSEA-4 grown in indicated media for 24 h. n = 3. Two-sided t test, ***p < 0.001, **<0.01.

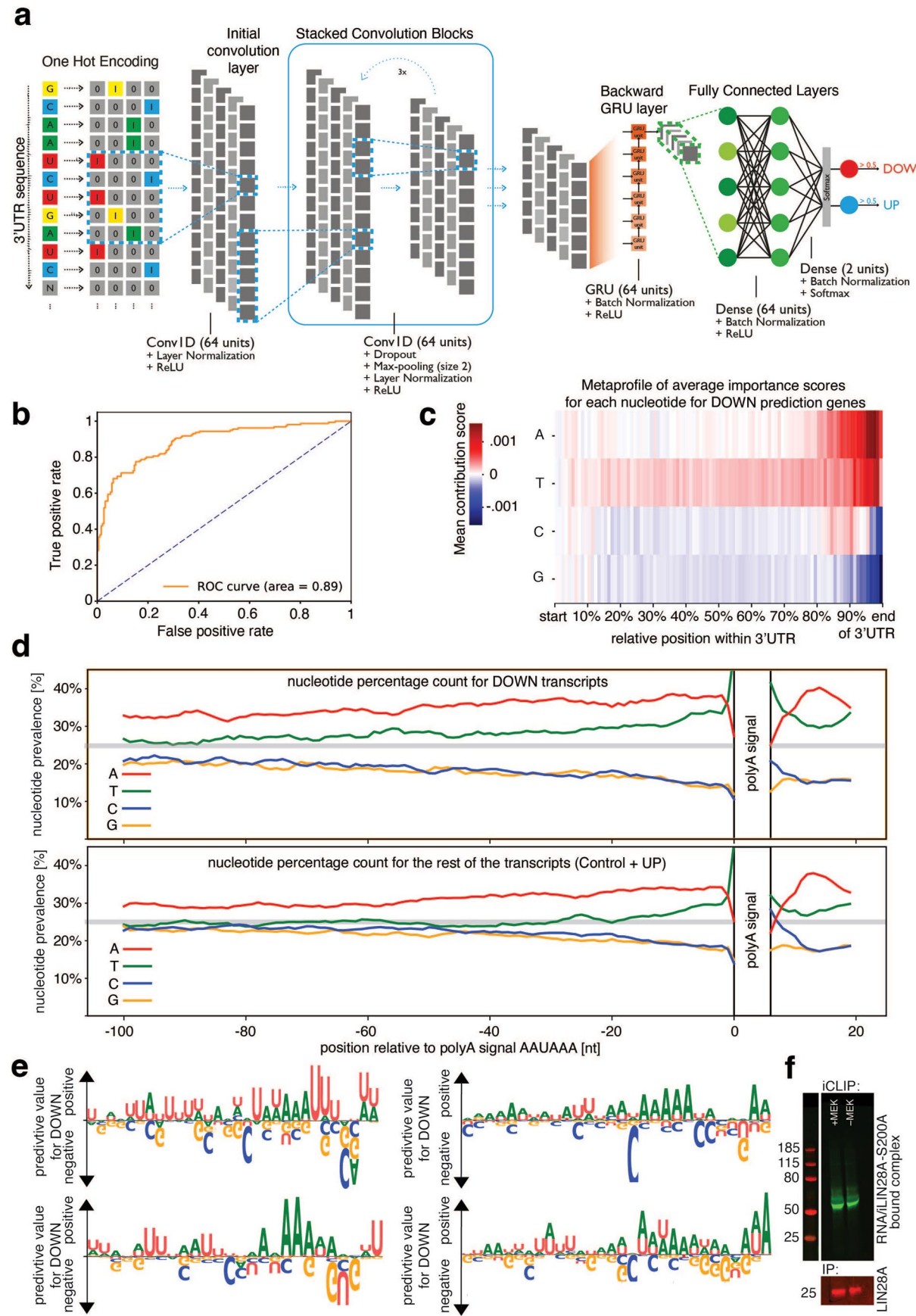

**Extended Data Fig. 4 | See next page for caption.**

**Extended Data Fig. 4 | Cis-acting determinants of mRNA decay. (a)** Schematic visualisation of the deep learning model for prediction of pLIN28A-dependent transcript destabilisation from 3'UTR nucleotide sequences (see Methods for details). **(b)** ROC curve benchmark shows high efficacy and robustness of the model in classifying the transcripts into DOWN or UP groups based on 3'UTR nucleotide sequences. The area under the ROC curve (auROC) is indicated. **(c)** Heatmap indicates summarised importance scores for DOWN prediction class for each of the four nucleotides, quantifying their contributions to model predictions across the full 3'UTR sequence. For each 3'UTR, importance scores at each nucleotide position were normalised by 3'UTR length, binned into 100 equal-sized bins, and summed within each bin. Finally, the mean score in each bin was calculated across all evaluated 3'UTRs and visualised (see Methods for details). **(d)** Metaprofile shows mean nucleotide composition 100nt upstream and 20nt downstream of PAS for DOWN (top) and UP/Control (bottom) transcripts (see Methods for details). The line plots indicate the increased A/U nucleotide content in DOWN transcripts around PAS, compared to UP/Control. **(e)** Top 4 sequence motifs identified with TF-MoDISco, by clustering 3'UTR regions of high-importance for the DOWN prediction class (see Methods for details). **(f)** Control experiment for Fig. 3c and representative Li-Cor imaging of the iCLIP nitrocellulose membrane, representing the protein-RNA complexes (above) and the amount of IPed phosphomutant LIN28A-S200A protein in the experiment (below).

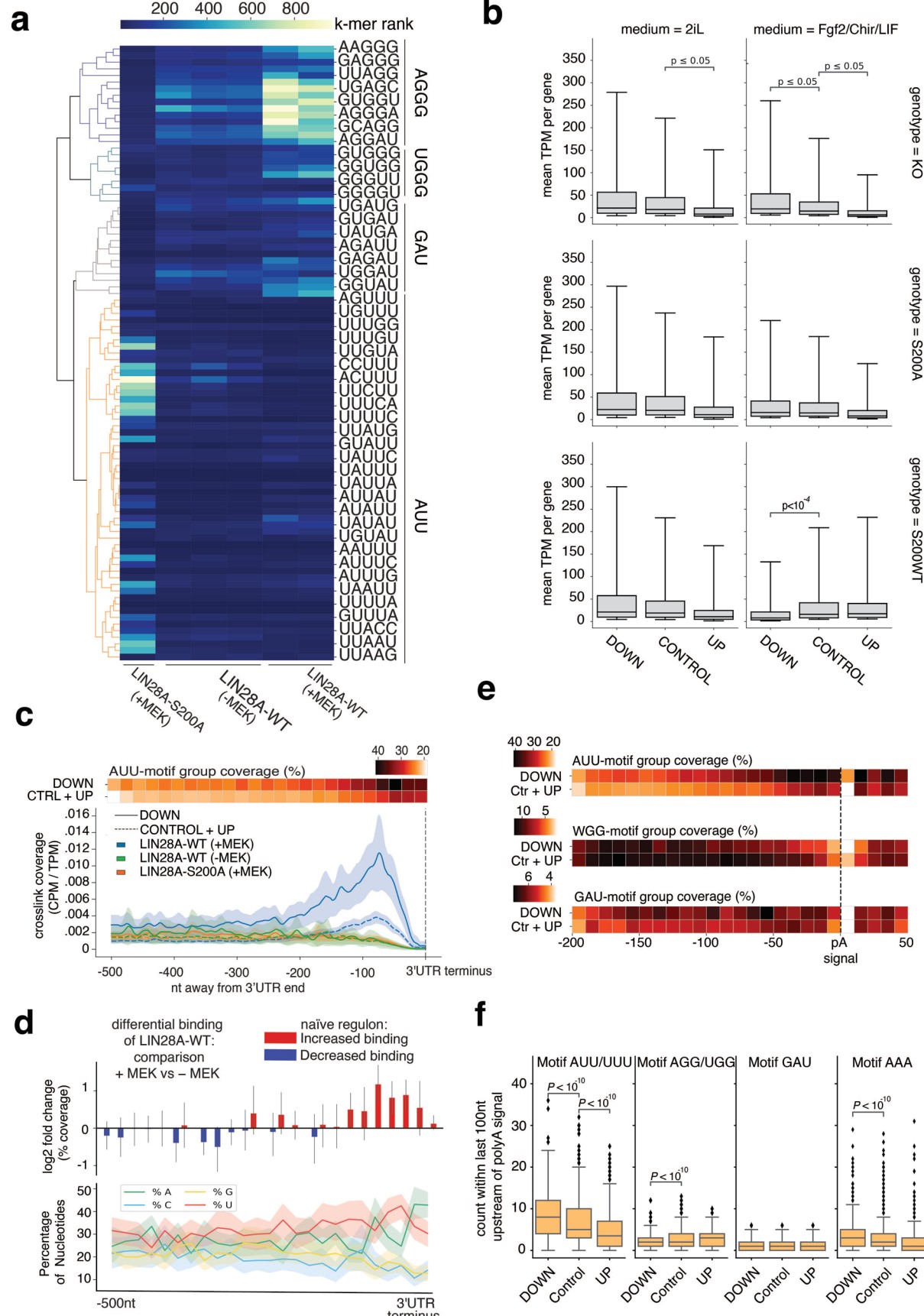

**Extended Data Fig. 5 | See next page for caption.**

**Extended Data Fig. 5 | Motifs enriched at crosslink sites of both phosphorylated and unphosphorylated LIN28A and their occurrence in 3'UTRs of naïve and DOWN mRNAs.** (**a**) Heatmap shows hierarchically clustered rankings of 5mers in the 3'UTRs significantly enriched by PEKA in any of the LIN28A iCLIP samples (Methods). (**b**) Boxplots display the average expression levels for genes in DOWN (n = 1183), Control (n = 2705) and UP (n = 988) groups. Expression for a given gene is assessed by summing up TPM values of its transcripts, and then taking a mean of this value across replicates. The plots show expression levels for LIN28A KO cells (top), with induced LIN28A-S200A (middle) and LIN28A-WT (bottom) expression, without (left; 2iLIF) and with (right, Fgf2/CHIR/LIF) MEK induction. Boxplot whiskers indicate a range of values within the 5 to 95 percentile. Two-sided Welch's t-test was performed to assess whether groups of genes significantly differ in average expression level and only significant comparisons are indicated. (**c**) Heatmap (top) shows the mean percentage of nucleotides covered by the AUU-motif group in DOWN (n = 786) and Control+UP (n = 2383) genes (Methods). The mean coverage is computed across evaluated genes in 20nt bins spanning the region of 500 nts before the 3'UTR termini. Line plot (bottom) indicates the mean of expression-normalised crosslink coverage for indicated iCLIPs. Shaded areas indicate 95% confidence interval. (**d**) Barplot (top) shows the log2 fold-change in expression-normalised crosslink coverage between iCLIPs of LIN28A-WT with and without MEK induction in genes of naïve regulon (n = 16, Methods). Fold-changes are shown in the region of 500 nts upstream of the 3'UTR termini. Error-bars represent 95% confidence interval, computed with bootstrapping (n = 1000). Line plot (bottom) shows the mean percentage of nucleotides in each bin. The shaded areas represent 95% confidence intervals, computed with bootstrapping (n = 1000). (**e**) Heatmaps show the mean percentage of nucleotides covered by AUU-, WGG- and GAU-motif groups in DOWN (n = 831) and Control/UP (n = 2372) (Methods). Mean coverage is calculated across evaluated genes in 10nt bins in a region 200nts upstream and 50nts downstream of PAS. (**f**) Boxplots show the count for WGG, GAU, AUU and AAA-motif trimers in a region 100nts upstream of the polyA signal. Each boxplot shows values for three groups of genes DOWN (n = 1002), Control (n = 2126) and UP (n = 778). Only 3'UTRs that contained the canonical PAS motif 'AAUAAA' were included in the quantification. Two-sided Mann-Whitney-Wilcoxon test was performed to assess significance.

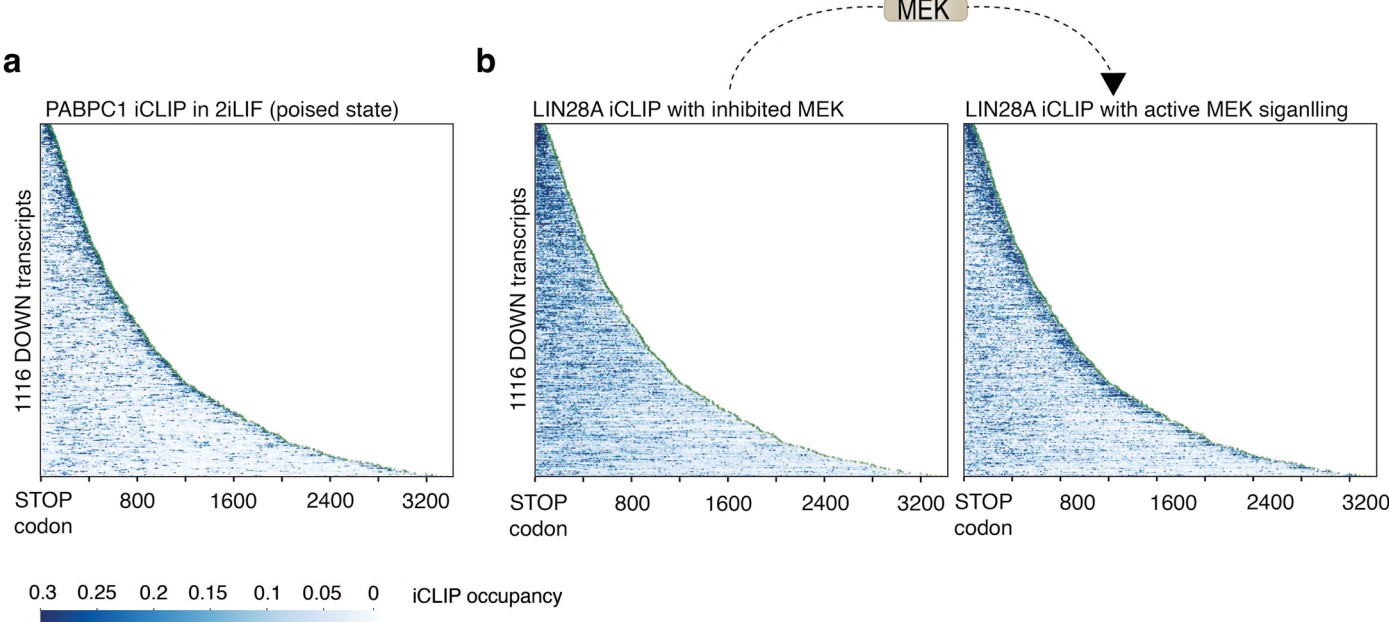

**Extended Data Fig. 6 | Heatmaps of iCLIP crosslinking profiles across all DOWN mRNAs. (a, b)** Heatmaps in blue show the iCLIP crosslinking profiles across all 1116 DOWN 3'UTRs for PABPC1 (**a**) and LIN28A (**b**). For LIN28A, iCLIPs in 2iLIF- (left) and FGF2-treated cells (right) are show and for PABPC1, iCLIPs in 2iLIF WT PSC cells are shown. For visualisation, iCLIP signal in each 3'UTR was smoothed and min-max normalised (Methods). PolyA signal is annotated with a green dot for the respective mRNA.

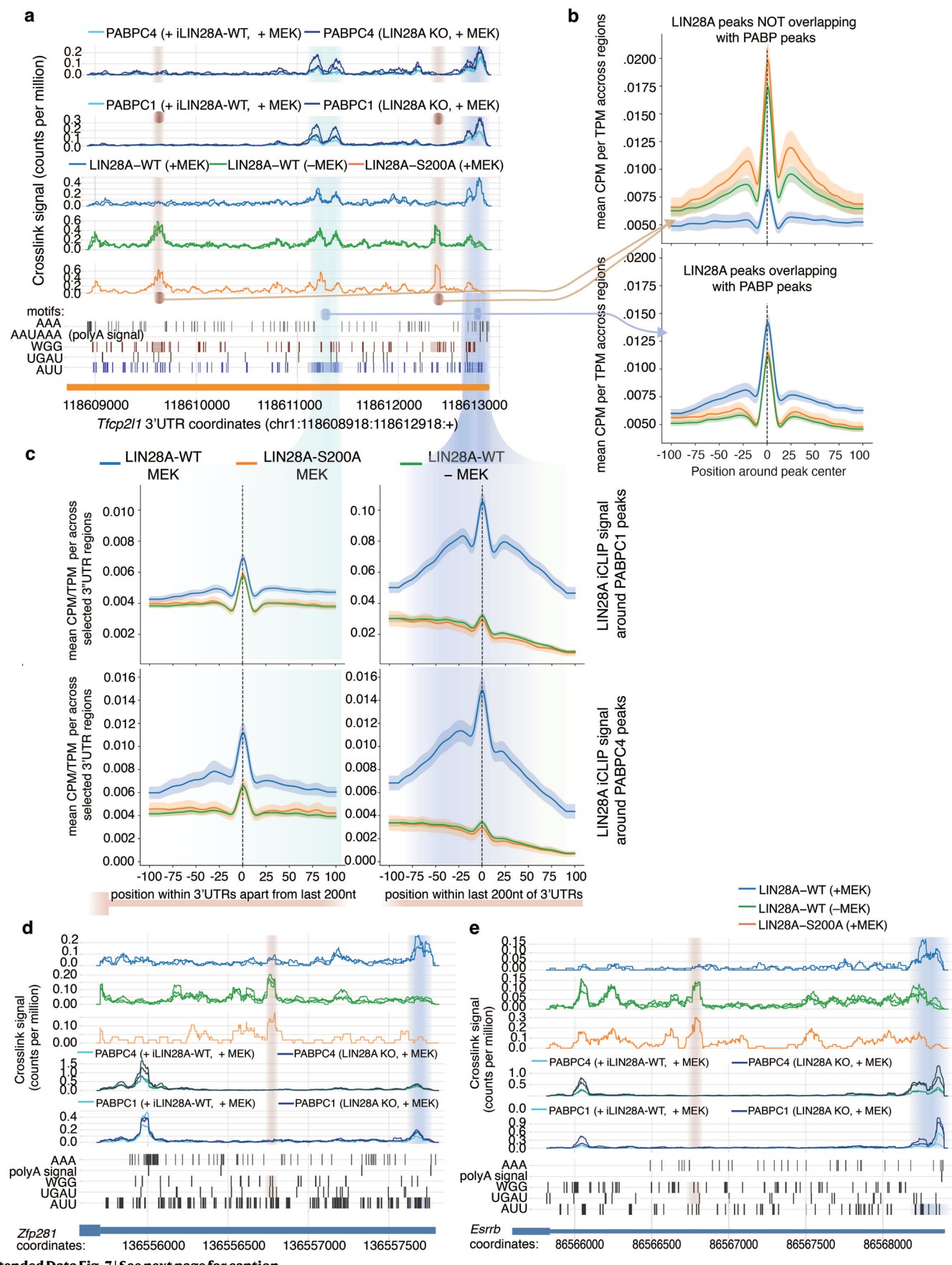

**Extended Data Fig. 7 | See next page for caption.**

**Extended Data Fig. 7 | AUU-rich mediate a convergence of pLIN28A with PABP at terminal 3'UTR regions.** (**a**) Library-normalised crosslink profiles of ~5 kb part of 3'UTR of naïve pluripotency factor Tfcp2l1 for LIN28A-WT (in 2iL and FGF2 treated cells), LIN28A-S200A (FGF2 treated cells) iCLIPs and PABPC1/4 iCLIPs (LIN28A KO with and without LIN28A overexpression). Auxilliary tracks below show motif-based binding sites of LIN28A that correspond to WGG-, GAU- and AUU-motif groups (Methods), to AAA trimer, and to canonical 'AAUAAA' PAS. (**b**) Line plot shows metaprofiles of expression-normalised crosslink coverage for LIN28A iCLIPs around the centers of LIN28A peaks in 3'UTRs, merged together from all samples (Methods). Peaks were stratified into those that overlap with peaks of PABPC1/4 (bottom) and those that do not (top). Metaprofile shows that the crosslinking signal of phosphorylated LIN28A-WT increases around peaks that overlap with PABPC1/4 binding and decreases around peaks that do not overlap with PABPC1/4 peaks. Shaded areas indicate a 95% confidence interval. (**c**) Line plot shows metaprofiles of expression-normalised crosslink coverage for LIN28A iCLIPs around the centers of PABPC1 peaks (above) and PABPC4 peaks (below) located within the final 200nts of 3'UTRs (right) or the rest of the 3'UTRs (left). Metaprofile shows a prominent increase in crosslink signal of pLIN28A around PABP peaks located in the last 200nt and thus indicates phosphorylation-induced repositioning of LIN28A to the 3'-termini. Shaded areas indicate a 95% confidence interval. More information in Methods. (**d, e**) Library-normalised crosslink profiles of naïve pluripotency factors Zfp281 (d) and Esrrb (e) 3'UTR for LIN28A-WT (in 2iL and FGF2 treated cells), LIN28A-S200A (FGF2 treated cells) iCLIPs and PABPC1/4 iCLIPs (LIN28A KO with and without LIN28A overexpression). Auxilliary tracks below show motif-based binding sites of LIN28A that correspond to WGG-, GAU- and AUU-motif groups (Methods), to AAA trimer, and to PAS.

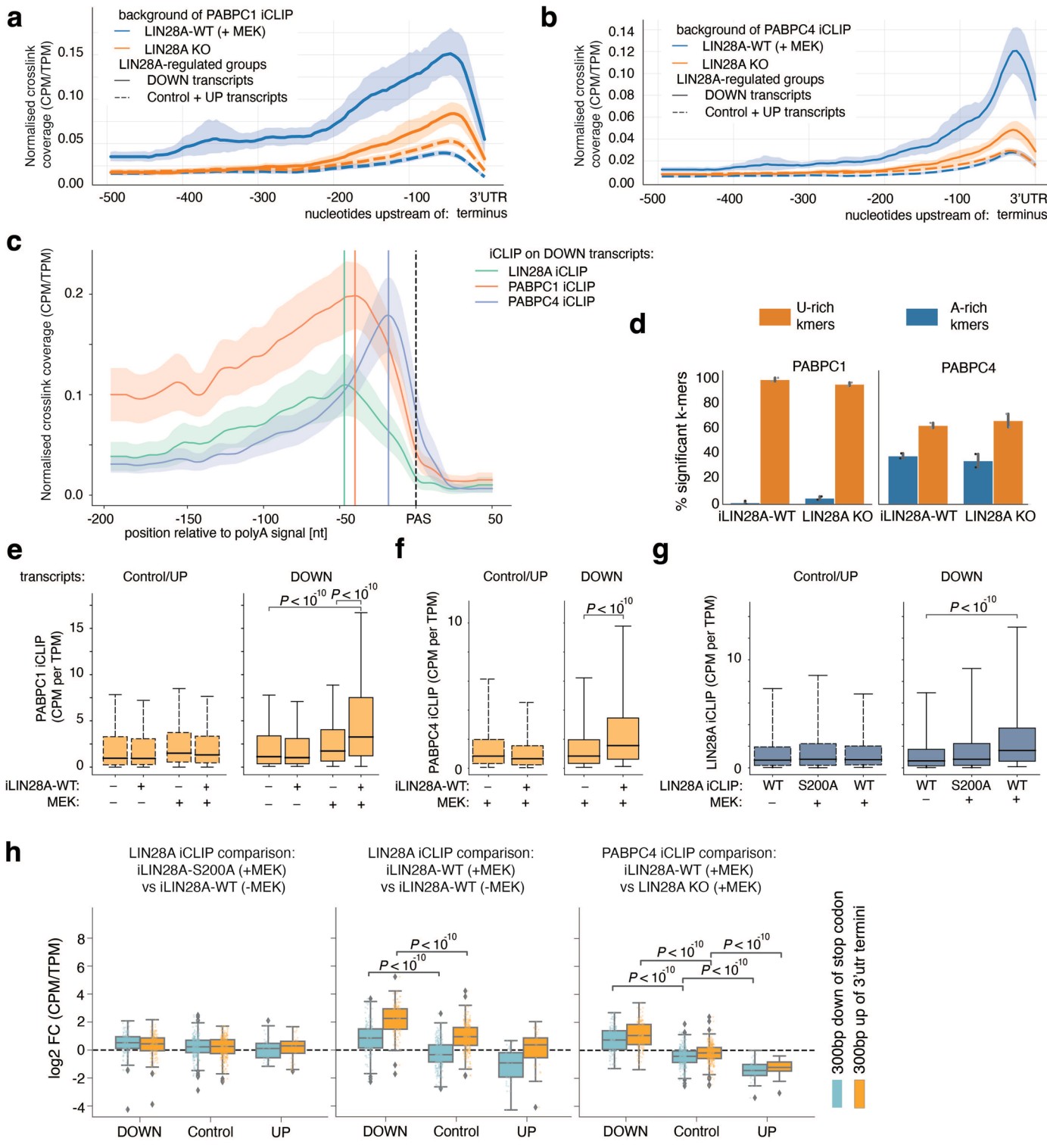

**Extended Data Fig. 8 | See next page for caption.**

**Extended Data Fig. 8 | Convergence of pLIN28A into the poised PABP-RNA hubs selectively enhances the PABP binding only in the DOWN mRNAs.**
(**a-b**) Line plots indicate the mean of expression-normalised crosslink coverage in the region 500 nts upstream of 3'UTR termini for iCLIPs of PABPC1 (a) and PABPC4 (b) in the FGF2-treated condition (+MEK) with (blue) or without (orange) LIN28A induction. Solid line shows the crosslink coverage in DOWN (n = 786), dashed line shows coverage in Control+UP (n = 2383) genes. Shaded areas indicate a 95% confidence interval. (**c**) Line plot indicates the location of enriched PABP/LIN28A crosslink coverage upstream of the polyA signal on DOWN (n = 831) transcripts. Shaded areas indicate a 95% confidence interval. (**d**) Barplots show the mean percentage of significantly enriched (p < 0.05) U-rich and A-rich 5-mers at PABPC1 (left) and PABPC4 (right) binding sites in 3'UTRs. The mean number of k-mers of two replicates indicated with dots and the errorbar. Motif enrichment was analysed with PEKA (see Methods). (**e**) Quantification of expression-normalised crosslink signal in the region 300nts upstream of 3'UTR termini for PABPC1 with and without MEK induction in the presence of absence of LIN28A–WT expression. Two-sided Mann-Whitney-Wilcoxon test was performed to assess significance. (**f**) Quantification of expression-normalised crosslink signal in the region 300nts upstream of 3'UTR termini for PABPC1 with and without MEK induction in the presence of absence of LIN28A–WT overexpression. (**g**) Quantification of expression-normalised crosslink signal in the region 300nts upstream of 3'UTR termini for LIN28A with and without MEK induction. (e-g) DOWN and Control+UP feature 586 and 2125 genes, respectively. The length of boxplot whiskers is limited to a maximum 1.5-times the interquartile range. Two-sided Mann-Whitney-Wilcoxon test was performed to assess significance. Only comparisons that pass the 10e-10 statistical threshold are annotated. (**h**) Boxplots show log2 fold-change of binding in indicated iCLIP experiments, quantified in the region of 300nts upstream of 3'UTR terminus (yellow) or 300nts downstream of stop codon (cyan). Each boxplot shows coverage values for three groups of genes (150, 284, 46 of DOWN, Control and UP, respectively), passing filters for 3'UTR length, expression and crosslink signal (Methods). Two-sided Mann-Whitney-Wilcoxon test was performed to assess significance.

# Reporting Summary

## Statistics

For all statistical analyses, confirm that the following items are present in the figure legend, table legend, main text, or Methods section.

| n/a | Confirmed | |
|---|---|---|
| ☐ | ☒ | The exact sample size (*n*) for each experimental group/condition, given as a discrete number and unit of measurement |
| ☐ | ☒ | A statement on whether measurements were taken from distinct samples or whether the same sample was measured repeatedly |
| ☐ | ☒ | The statistical test(s) used AND whether they are one- or two-sided *Only common tests should be described solely by name; describe more complex techniques in the Methods section.* |
| ☒ | ☐ | A description of all covariates tested |
| ☐ | ☒ | A description of any assumptions or corrections, such as tests of normality and adjustment for multiple comparisons |
| ☐ | ☒ | A full description of the statistical parameters including central tendency (e.g. means) or other basic estimates (e.g. regression coefficient) AND variation (e.g. standard deviation) or associated estimates of uncertainty (e.g. confidence intervals) |
| ☐ | ☒ | For null hypothesis testing, the test statistic (e.g. *F*, *t*, *r*) with confidence intervals, effect sizes, degrees of freedom and *P* value noted *Give P values as exact values whenever suitable.* |
| ☒ | ☐ | For Bayesian analysis, information on the choice of priors and Markov chain Monte Carlo settings |
| ☒ | ☐ | For hierarchical and complex designs, identification of the appropriate level for tests and full reporting of outcomes |
| ☒ | ☐ | Estimates of effect sizes (e.g. Cohen's *d*, Pearson's *r*), indicating how they were calculated |

*Our web collection on statistics for biologists contains articles on many of the points above.*

## Software and code

Policy information about availability of computer code

| Data collection | Steady state super-resolution imaging of live ESCs and immunofluorescence images were acquired using a immunoOlympus IX83 microscope equipped with a VT-iSIM super resolution imaging system using MicroManager system. FACS data was acquired using a LSR Fortessa (BD Biosciences, BD FACSDiva Version 9.2.) flow cytometer. RT-qPCR data was acquired using QuantStudio7 (Thermo) and corresponding software. Western blot data was acquired using Amersham Imager 680 blot and gel imager. NGS libraries were sequenced as single end 100bp reads on Illumina HiSeq 4000. iCLIP experiments targeting LIN28A, PABPC1, and PABPC4 were sequenced on NovaSeq platform. Direct RNA sequencing for assessment of polyA-tail length was performed with MinION. |
|---|---|
| Data analysis | Data analysis, the software, and the settings used are described in "Methods". Code for bioinformatic analyses is available on GitHub (https://github.com/ulelab/LIN28A_RNPreassembly_bioinformatics), together with YAML environment files (containing versioned packages used for the analysis). Code and dependencies are also archived on Zenodo (https://zenodo.org/doi/10.5281/zenodo.10054297). iCLIP-seq data was processed on iMaps Goodwright web-server (https://imaps.goodwright.com/), and the links to relevant collections are specified in Methods. The code and settings used in the iCLIP analysis pipeline (release v0.30) can be viewed at https://github.com/goodwright/imaps-nf, and are also archived on Zenodo (https://zenodo.org/doi/10.5281/zenodo.10054231). Package versions used: apeglm bedtools (v2.29.2) Clippy (v1.4.1) Cutadapt (v3.4) DESeq2 (v1.42) |

For manuscripts utilizing custom algorithms or software that are central to the research but not yet described in published literature, software must be made available to editors and reviewers. We strongly encourage code deposition in a community repository (e.g. GitHub). See the Nature Portfolio guidelines for submitting code & software for further information.

```
 fastp (v0.19.11)
Genome Analysis Toolkit (v3.5)
gbm (v2.1.8.1)
Guppy (v6.0.0)
iCount (v2.0.1.dev)
ImageJ (v1.52p)
MAJIQ (v2.1)
Integrative Genomics Viewer (v2.9.1)
minpack.lm (v1.2)
Nanopolish (v0.14.0)
Optuna (v3.1.0)
pybedtools (v0.9.0)
R (v4.0.3)
Salmon (v1.5.1 2)
samtools (v1.6)
seqkit (v2.3.1)
SHapley Additive exPlanations (SHAP) (v0.35.0)
STAR (v2.7.9a)
Snakemake (v5.3.0)
Slamdunk (v0.3.3)
TF-MoDISco Lite (v2.0.0)
VarScan (v2.4.1)
Pyranges (v0.0.117)
```

For manuscripts utilizing custom algorithms or software that are central to the research but not yet described in published literature, software must be made available to editors and reviewers. We strongly encourage code deposition in a community repository (e.g. GitHub). See the Nature Portfolio guidelines for submitting code & software for further information.

# Data

Policy information about availability of data

All manuscripts must include a data availability statement. This statement should provide the following information, where applicable:
- Accession codes, unique identifiers, or web links for publicly available datasets
- A description of any restrictions on data availability
- For clinical datasets or third party data, please ensure that the statement adheres to our policy

Sequencing data related to iCLIP, Quantseq and Nanopore direct RNA sequencing experiments for Flag-tagged LIN28A-WT in 2iLIF and FGF2 treated cells, Flag-tagged LIN28A-S200A in FGF2 treated cells as well as for iCLIPs of PABPC1 and PABPC4 (in LIN28A KO cells with and without LIN28A overexpression) are available from ENA, with the accession code PRJEB60519. SlamSeq sequencing data, Quantseq data of LIN28A-GFP overexpression in LIN28A KO cells and LIN28A-GFP iCLIP experiments can be retrieved from GEO accession GSE169555.In addition, full data produced by iCLIP analysis pipeline for LIN28A-WT (in 2iL and FGF2 treated cells), LIN28A-S200A (in FGF2 treated cells) as well as for PABPC1 and PABPC4 (in LIN28A KO cells with and without LIN28A overexpression), can be accessed on the iMaps and Flow web-servers:
LIN28A iCLIP experiments: https://imaps.goodwright.com/collections/882635250203/; https://app.flow.bio/projects/882635250203/
PABPC iCLIP experiments: https://imaps.goodwright.com/collections/340215254997/; https://app.flow.bio/projects/340215254997/
To facilitate reproduction of our work, we archived key data from iCLIP analysis, used in downstream bioinformatic analyses–crosslink sites, peaks, and motif enrichments from PEKA–on Zenodo (https://zenodo.org/doi/10.5281/zenodo.10054231).

# Research involving human participants, their data, or biological material

Policy information about studies with human participants or human data. See also policy information about sex, gender (identity/presentation), and sexual orientation and race, ethnicity and racism.

| | |
|---|---|
| Reporting on sex and gender | Not applicable |
| Reporting on race, ethnicity, or other socially relevant groupings | Not applicable |
| Population characteristics | Not applicable |
| Recruitment | Not applicable |
| Ethics oversight | Not applicable |

Note that full information on the approval of the study protocol must also be provided in the manuscript.

# Field-specific reporting

Please select the one below that is the best fit for your research. If you are not sure, read the appropriate sections before making your selection.

☒ Life sciences    ☐ Behavioural & social sciences    ☐ Ecological, evolutionary & environmental sciences

For a reference copy of the document with all sections, see nature.com/documents/nr-reporting-summary-flat.pdf

# Life sciences study design

All studies must disclose on these points even when the disclosure is negative.

| | |
|---|---|
| Sample size | No statistical methods were used to predetermine the sample size. For RNA-seq on cell lines new to this study was determined by prior literature (for instance PMID: 34380047 and PMID: 28945705) and thereby we used similar experimental approaches rather than by power analysis. |
| Data exclusions | One iCLIP sample (LIN28A-S200A_ESC_LIF-CHIR-FGF0220626_MM_2) was excluded from subsequent analyses due to low read coverage. |
| Replication | The number of replicates used in each experiment are described in the figure legends and/or in the Methods section, as are the statistical tests used including the adjustment methods and alpha values for adjusted P values. All replicates successfully reproduced the presented findings, giving consistent results. LIN28A rescue was replicated in multiple independent cell lines, including multiple LIN28A KO clones were used. Statistical tests are selected appropriately to the analysed data, considering normality, variance, independence (paired or independent tests), and direction of effect (two-sided or one-sided tests). To compare two independent samples, with approximately normal distribution and approximately equal variance, we used two-sided two sample t-test; When the data is approximately normally distributed, but the equal variance criteria is not met, we used two-sided Welch's t-test; When the data did not meet the criteria for normality and/or equal variance, we used the two-sided Mann-Whitney-Wilcoxon rank-sum test. We occasionally employed the non-parametric two-sided Mann-Whitney-Wilcoxon rank-sum test in place of two sample t-test and Welch's t-test, due to fewer underying assumptions. In the figures, we indicate the comparisons of interest and report the exact P values when they are in range between 10e-4 to 10e-10; for P values lower than 10e-10, we label the P values as <10e-10. |
| Randomization | Randomization is not relevant to this study as no randomization is required due to the homogeneous nature of the cell lines. Furthermore, our RNA-seq discovery assays are high-throughput and were initially done in a hypotheses-free analysis. Observations in the RNA-seq were validated and confirmed by biochemical assays to bolster initial observations from the high-throughput assays. |
| Blinding | Immunoflourescence validation experiments were analysed blinded to genotype status. |

# Reporting for specific materials, systems and methods

We require information from authors about some types of materials, experimental systems and methods used in many studies. Here, indicate whether each material, system or method listed is relevant to your study. If you are not sure if a list item applies to your research, read the appropriate section before selecting a response.

## Materials & experimental systems

| n/a | Involved in the study |
|---|---|
| ☐ | ☒ Antibodies |
| ☐ | ☒ Eukaryotic cell lines |
| ☒ | ☐ Palaeontology and archaeology |
| ☒ | ☐ Animals and other organisms |
| ☒ | ☐ Clinical data |
| ☒ | ☐ Dual use research of concern |
| ☒ | ☐ Plants |

## Methods

| n/a | Involved in the study |
|---|---|
| ☒ | ☐ ChIP-seq |
| ☐ | ☒ Flow cytometry |
| ☒ | ☐ MRI-based neuroimaging |

# Antibodies

| | |
|---|---|
| Antibodies used | Antibodies used for cell immunoflourescence<br>LIN28A (A177 Cell Signaling and ab63740 Abcam)<br>Klf4 (AF3158, R&D Systems)<br>Nanog (8822, Cell Signaling)<br><br>Antibodies used for FACS experiments<br>SSEA1 (MC480, Thermo)<br>SSEA4  (MC813-70, Thermo)<br><br>Antibodies used for Western blotting<br>LIN28A (A177 Cell Signaling and AF3757, R&D Systems)<br>H3 (Abcam, ab1791) |

GAPDH (Cell Signaling, 2118S)

Antibodies used for iCLIP
GFP polyclonal Antibody - Thermo Fisher A6455
LIN28A polyclonal Antibody - Cell signaling #3978
PABPC1 - Abcam ab21060 and Proteintech 10970-1-AP
PABPC4 - Proteintech 14960-1-AP

| | |
|---|---|
| Validation | Antibodies used for Western blotting and cell immunoflourescence:<br>LIN28A (A177 Cell Signaling). An antibody routinely used as a first choice for LIN28A immunoprecipitations, Western Blots and immunoflourescence: Product website outlines 42 relevant citations (https://www.cellsignal.co.uk/products/primary-antibodies/lin28a-a177-antibody/3978?N=0+4294956287&Nrpp=200&No=3200&fromPage=plp), including publications that validated the specificity of A177 LIN28A Ab when compared to a cell line not expressing LIN28A (PMID: 27992407).<br><br>LIN28A (ab63740 Abcam). Relevant newer monoclonal antibody that has already been cited 8times and we have thoughtfully validated it's specificity for Immunofluorescence and WB using a LIN28A KO cell line (Extended Data Figure 1). Product Website: https://www.abcam.com/lin28-antibody-ab63740.html<br><br>Klf4 (AF3158, R&D Systems). Widely used antibody that has been extensively validated (for instance PMID: 30540935) and published with 33 citations according to the product website (https://www.rndsystems.com/products/mouse-klf4-antibody_af3158#product-citations)<br><br>Nanog (8822, Cell Signaling). Widely used antibody that has been widely accepted as a marker of naive pluripoteny stem cells, is extensively validated (e.g. PMID: 32034125) and published with 32 citations according to the product website ( https://www.cellsignal.com/products/primary-antibodies/nanog-d2a3-xp-rabbit-mab-mouse-specific/8822)<br><br>Two PABPC1 antibodies were cross-validated (ab21060 and 10970-1-AP) with comparable enrichment in kmers and peak-calling. Antibodies have also been used in 120 and 4 citations respectively.<br><br>PABPC4 antibody (Proteintech) was validated using a KO cell lines and cited in 6 additional publications. https://www.ptglab.com/products/PABPC4-Antibody-14960-1-AP.htm<br><br>The following secondary antibodies were used that were previously tested in PMID: 31047794:<br>Donkey anti-rabbit IgG 555 Invitrogen Cat no. A31572; RRID: AB_162543<br>Donkey anti-goat IgG 488 Invitrogen Cat no. A11055; RRID: AB_2534102<br><br>Antibodies used for FACS experiments:<br>All listed antibodies have already been thoroughly tested and validated with isotype controls for the same application of detecting cell-Surface Markers Specific to Naive and Formative/Primed Pluripotent States (PMID: 31078527 and PMID: 31047794)<br>Pacific Blue™ anti-mouse CD117 (c-Kit) Antibody (105820, Biolegen)<br>SSEA4 Monoclonal Antibody (MC-813-70, Thermo) |

## Eukaryotic cell lines

Policy information about cell lines and Sex and Gender in Research

| | |
|---|---|
| Cell line source(s) | IDG3.2. | Helmholtz Zentrum Munich Transgenic Facility (AG Schirge)<br>V6.5. | Novus Bio and Harvard (gift of Prof. George Daley).<br>All CRISPR and other genetic modifications were perfomed on outlined cell lines by paper authors. |
| Authentication | CRISPR modifications were genotyped and assessed by Western Blot and PCR. |
| Mycoplasma contamination | All cell lines were tested free of mycoplasma contamination by Cell Services of the Francis Crick Institute, London, UK. |
| Commonly misidentified lines<br>(See ICLAC register) | No commonly misidentified lines were used in this study. |

## Flow Cytometry

### Plots

Confirm that:

☒ The axis labels state the marker and fluorochrome used (e.g. CD4-FITC).

☒ The axis scales are clearly visible. Include numbers along axes only for bottom left plot of group (a 'group' is an analysis of identical markers).

☒ All plots are contour plots with outliers or pseudocolor plots.

☒ A numerical value for number of cells or percentage (with statistics) is provided.

### Methodology

| | |
|---|---|
| Sample preparation | For flow cytometry experiments, single-cell suspensions were made using Accutase (A6964, Sigma-Aldrich) for 5 min in 37 °C; or Enzyme-Free Cell Dissociation Buffer (13151014, Gibco) for 30 min at 37 °C and washed with 5% FBS (EmbryoMax® ES Cell |

Qualified FBS, ES-009-B, Merck) in PBS, incubated with fluorophore-conjugated antibodies for 30-60 min on ice. Cells were centrifuged, resuspended and 10000 cells was analyzed using a LSR Fortessa cytometers (BD Biosciences, BD FACSDiva Version 9.2.). Cell debris were excluded by forward and side scatter gating.

Instrument                    LSR Fortessa (BD Biosciences, BD FACSDiva Version 9.2.).

Software                      Data was acquired using BD FACSDiva and analyzed using FlowJo V10.6.1.

Cell population abundance     No sorting was performed. The entire cell population was analysed. No post-gating sort was performed.

Gating strategy               Forward and side scatter were set as to gate single cells.

☒ Tick this box to confirm that a figure exemplifying the gating strategy is provided in the Supplementary Information.

