## [Peer Review File · Nature Structural & Molecular Biology]

Peer Review Information

Manuscript Title: Poised PABP-RNA hubs implement signal-dependent mRNA decay in development

Corresponding author name(s): Jernej Ule, Miha Modic

Reviewer Comments & Decisions:

Decision Letter, initial version:

Message: 18th Sep 2023

Dear Dr. Ule,

Thank you again for submitting your manuscript "Poised PABP-RNA hubs implement signal-dependent mRNA decay in development". We now have comments (below) from the 3 reviewers who evaluated your paper. In light of those reports, we remain interested in your study and would like to see your response to the comments of the referees, in the form of a revised manuscript.

You will see that all reviewers appreciate the results and find the conclusions timely and of wide interest. There are, however, several comments and suggestions that should be addressed in a revision. Specifically, reviewer #2 raises several technical issues that should be addressed, while reviewer #3 requires important clarifications and discussion in the context of the literature, and some helpful additional analysis. Please be sure to address and respond to all concerns of the referees in full in a point-by-point response and highlight all changes in the revised manuscript text file. If you have comments that are intended for editors only, please include those in a separate cover letter.

We expect to see your revised manuscript within 6 weeks. If you cannot send it within this time, please contact us to discuss an extension; we would still consider your revision, provided that no similar work has been accepted for publication at NSMB or published elsewhere.

Reporting Summary:

When submitting the revised version of your manuscript, please pay close attention to our [Digital Image Integrity Guidelines](https://www.nature.com/nature-portfolio/editorial-policies/image-integrity). and to the following points below:

Please note that all key data shown in the main figures as cropped gels or blots should be presented in uncropped form, with molecular weight markers. These data can be aggregated into a single supplementary figure item. While these data can be displayed in a relatively informal style, they must refer back to the relevant figures. These data should be submitted with the final revision, as source data, prior to acceptance, but you may want to start putting it together at this point.

Data availability: this journal strongly supports public availability of data. All data used in accepted papers should be available via a public data repository, or alternatively, as Supplementary Information. If data can only be shared on request, please explain why in your Data Availability Statement, and also in the correspondence with your editor. Please

note that for some data types, deposition in a public repository is mandatory - more information on our data deposition policies and available repositories can be found below: <https://www.nature.com/nature-research/editorial-policies/reporting-standards#availability-of-data>

Nature Structural & Molecular Biology is committed to improving transparency in authorship. As part of our efforts in this direction, we are now requesting that all authors identified as 'corresponding author' on published papers create and link their Open Researcher and Contributor Identifier (ORCID) with their account on the Manuscript Tracking System (MTS), prior to acceptance. This applies to primary research papers only. ORCID helps the scientific community achieve unambiguous attribution of all scholarly contributions. You can create and link your ORCID from the home page of the MTS by clicking on 'Modify my Springer Nature account'. For more information please visit please visit www.springernature.com/orcid.

[Redacted]

Sincerely,

Carolina Perdigoto, PhD
Chief Editor
Nature Structural & Molecular Biology
orcid.org/0000-0002-5783-7106

Referee expertise:

Referee #1:

Referee #2:

Referee #3:

Reviewers' Comments:

Reviewer #1:

Remarks to the Author:

This is a fantastic stories the bring great new knowledge onto how phosphorylation of ERK leads to decay of naive pluripotency and priming towards formative and primed pluripotency. the authors provide very clean results on mouse naive ESCs, how upon priming ERK activity leads to LIN28 phosphorylation which then targets several naive mRNA transcripts for decay via repositioning pLIN28A to the PABP-RNA hubs at 3'UTR termini. The authors do very convincing work on motif analysis and RNA-decay to prove the results and selective decay hypothesis.

I did not find any major points to further improve the manuscript and did not find any flaws. It is super solid work and clearly written manuscript and with proper background and citations to the issues being addressed. I think it is a noteworthy contribution, linking signaling, cell state change and RBA in depth mechanism analysis to show selective RNA decay and changes.

Reviewer #2:

Remarks to the Author:

The manuscript by Modic et al. demonstrates an intriguing mechanism by which LIN28A and PABP facilitate the degradation of naive pluripotency gene mRNAs. This mechanism is important for the termination of the naive pluripotent program during PSC differentiation. Although the functions of LIN28A in PSCs and differentiation have been extensively studied, this study provides novel mechanistic insights into how LIN28A and PABP regulate mRNA polyA length and stability. This study is conceptually and technically novel as it applies machine learning and bioinformatics tools to discover novel factors that play a role in the regulation of mRNA decay.

Overall, this study is interesting and the findings are critical to understanding the molecular regulation of RNA-binding proteins and mRNA metabolism in the transition between different cellular fates. The following main and minor points are for the authors to consider in further improving the manuscript.

Major Points:

1. The expression of the LIN28A protein (and LIN28B) in PSCs and during the primed transition should be provided. Functional redundancy in the pluripotent state transition was previously observed (PMID 27320042). Therefore, the authors should explain why LIN28A, but not Lin28B, was not identified in the pipeline applied in this study and discuss the potential implications in cell fate transitions.
2. The expression of the PABPC1 protein should be provided as it is the second hit in Fig. 1e and a major focus of the study.
3. In Fig. 1, the authors used sophisticated deep learning algorithms to discover the important features that relate to the rapid degradation of the naive mRNA regulon. Since

only the top hit, LIN28A, was chosen for further study, the authors should at least discuss the relevance of other candidates in Fig. 1e to highlight the robustness of such algorithms. 4. The authors quantified protein expression by immunostaining and presented the distribution of intensities using boxplots with p-values (Fig. S1b, Fig. S2g). While staining data can be arbitrary, WB data are more straightforward and should be provided. This also applies to the AP staining data (Figs 2i and S2i), where only boxplots were presented without actual images.

5. PABP has been reported to interact with the LIN28A CSD. The authors could further examine whether WT LIN28A, pLIN28A, or LIN28A-S200A have different binding affinities to PABP. This result would help us to understand the model presented in Fig. 4f.

6. LIN28A iCLIP of LIN28A was performed under conditions LIN28A-WT (-MEK), (+MEK), and LIN28A-S200A (+MEK) conditions, whereas iCLIP was performed under conditions LIN28A-KO (-MEK) and LIN28A-WT (+MEK). Is there a reason for such an experimental plan? It would be better to perform the iCLIP of LIN28A and PABP under the same conditions to enable a direct comparison and, therefore, to better understand the model. LIN28A-KO can be resistant to differentiation. Furthermore, LIN28A-S200A (+MEK) could be a better control than LIN28A-KO.

Minor points:

1. In Fig 4f, top, the middle "LIN28A" was labelled "LIN", is there a reason for that?
2. In certain places, "LIN28a" was used, which should be 'LIN28A' to be consistent.
3. In the Results, it was stated "MEK/ERK phosphorylation of LIN28A is also sufficient for commitment towards the primed state as indicated by the 3-fold increase in the amount of SSEA4-positive cells (Extended Data Fig 2g-h)". But such data were not presented in Fig. S2g-h.

Reviewer #3:

Remarks to the Author:

In this study, Modic and colleagues provide an elegant analysis on how protein-protein interactions of two RNA Binding Proteins drastically degrades the naïve PSCs regulon to allow the transition towards a primed state. The manuscript present very solid data accompanied by accurate computational analysis. The manuscript is very well written, easy to follow and the conclusions are robust. I suggest the following improvements and highly recommend it for publishing:

1- It is widely accepted that the naïve pluripotency transcriptional program is highly dynamic in PSCs firstly due to fluctuations over time in the expression levels of Nanog and other pluripotency regulators; and secondly, that different subpopulations of PSCs have different capacity to differentiate and self-renew. So it has been postulated that PSCs fluctuate in multiple interconvertible states and this metastability and plasticity might be fundamental for their pluripotent identity. Single-cell RNA-seq analysis has revealed that PSCs are highly heterogeneous in their gene expression when grown in standard SL medium conditions. Particularly, two classes of transcriptional heterogeneity have been described according to the gene expression distribution amongst PSCs: bimodal expression, with transcripts present in some cells and not in others; and sporadic expression, with transcripts in a reduced number of cells but at high levels. This transcriptional heterogeneity in PSCs has been proposed to arise from stochastic fluctuations in gene expression, cell cycle phase asynchrony, transcriptional bursting, differential epigenetic regulation, partial differentiation and differences in colony size and

morphology, and aid in symmetry breaking and cell-fate decision making as it may create competence for subpopulations of cells to respond to signalling cues. In line with this, single-cell sequencing of SL PSCs showed that cells exist in three distinct states within the whole population. The most numerous subpopulation exhibits high levels of pluripotency factors, another exhibits a differentiation permissive profile while the smallest subpopulation is already in the differentiation path. Another study found two subpopulations instead: one with a transcriptional signature similar to E2.5-3.5 embryos and another, similar to the E5.5 epiblast. They proposed that the transcriptional similarity to the E4.5 epiblast was to be interpreted as the combination of these two subpopulations (see Kolodziejczyk AA, et al., *Cell stem Cell*, 2015 and Papatsenko D, et al., *Stem Cell Reports*, 2015). Do the authors think that single cell analysis of LIN28A KO ESCs can help finetuning the exact role this protein in the different subpopulations?

2- The naïve-to-primed transition also comprises a metabolic switch. PSCs have a bivalent metabolism consisting of both oxidative phosphorylation and glycolysis, whereas RSCs have low rates of oxygen consumption and full dependence on glycolysis (Zou W, et al., *EMBO J* 2012). Do the genes that are regulated by LIN28A belong to these pathways?

3- RSCs express OCT4, SOX2 and NANOG, the core pluripotency transcription factors, also expressed in PSCs. However, the gene expression of Oct4 changes from being regulated mainly by its distal enhancer element in naïve pluripotency to being mainly regulated through its proximal enhancer element in primed pluripotency (Factor DC, et al., *Cell Stem Cell*, 2014). Did the authors check the promoter usage of the regulated transcripts by LIN28A?

4- Other molecular regulatory players also switch in the naïve-to-primed transition and their effects help shape the two identities. In fact, in comparison with nPSCs, RSCs show a more closed chromatin structure, use different enhancer elements and possess a different set of epigenetic regulators and miRNAs. Additionally, even if the regulatory players are in the same levels in both naïve and primed states, sometimes, their distribution changes. This is the case of H3K4me1, an active enhancer mark whose distribution changes in the naïve-to-primed transition. This change in H3K4me1 distribution plays a key role in establishing the primed identity as perturbing it results in the resetting of RSCs towards a more naïve pluripotent state (Zhang H, et al., *cell stem cell*, 2016). Can the authors elaborate on LIN28A expression in nPSCs and RSCs?

5- One study found that the splicing factor and RNA processing HTATSF1 played a role in naïve-to-primed pluripotency transition. More specifically, HTATSF1 mediates intron removal from around 45 ribosomal protein transcripts and thus mediates ribosomal abundance and protein-synthesis. Furthermore, they reported that downregulation of HTATSF1 is crucial in the establishment of primed pluripotency as they were not able to derive RSCs from Htatsf1-overexpressing ESCs. Conversely, they showed that overexpressing Htatsf1 facilitated the conversion of EpiSCs into reverted ESCs (Corsini et al., *Cell Stem Cell*, 2018). Can the authors check the overlap between the targets of LIN28A and HTATSF1?

Minor comments:

- Mislabelling in figure 1a and supp fig 1a (the median 4.5 fold median decrease is observed at 24 h) same for protein.
- Compare the expression levels and sub cellular localization of LIN28A and LIN28B in naïve and primed ESCs
- What is the control set that is used for the training network? Can the authors use different sets of 1000 transcripts randomly selected and compare their performance to the top 1000 that they used in terms of CLIP features?

- Extended figure 1 I expression levels of let7-f and let7-g should be compared to WT.
- Authors can use the previously published data by Zhang Jin and colleagues where they show how Lin28 (A and B) regulate transition from naïve to primed pluripotency and check the overlap of the predicted targets.
- In fig 3a what about the UP regulated genes category?
- In figure 3c can the authors show the IP of Lin28A-S200A in MEK- and + conditions?
- In extended figure 4e-f, if authors consider the region around PAS in the transcriptome, not only the lin28A targets, how would the AUU trimers enrichment look like compared to the lin28A down targets?

Author Rebuttal to Initial comments

We thank all three reviewers for the positive feedback on the manuscript. Reviewers 2 and 3 raised some important questions that required clarifications, discussion in the context of the literature, and some minor additional analysis. We added a new Extended Data Figure 3 and updated Figure 1, Extended Data Figures 1, 2, and 4 with new analysis and data. We have addressed all of the specific points made in response to the initial submission and we list them below. We hope to have addressed all concerns and that the reviewers and editor now consider this paper ready for publication.

Reviewer #1:

Remarks to the Author:

This is a fantastic stories the bring great new knowledge onto how phosphorylation of ERK leads to decay of naive pluripotency and priming towards formative and primed pluripotency. the authors provide very clean results on mouse naive ESCs, how upon priming ERK activity leads to LIN28 phosphorylation which then targets several naive mRNA transcripts for decay via repositioning pLIN28A to the PABP-RNA hubs at 3'UTR termini. The authors do very convincing work on motif analysis and RNA-decay to prove the results and selective decay hypothesis.

I did not find any major points to further improve the manuscript and did not find any flaws. It is super solid work and clearly written manuscript and with proper background and citations to the issues being addressed. I think it is a noteworthy contribution, linking signaling, cell state change and RBA in depth mechanism analysis to show selective RNA decay and changes.

Response:

We thank the reviewer for the review and for all the encouraging words in regards to our work.

Reviewer #2:

Remarks to the Author:

The manuscript by Modic et al. demonstrates an intriguing mechanism by which LIN28A and PABP facilitate the degradation of naive pluripotency gene mRNAs. This mechanism is important for the termination of the naive pluripotent program during PSC differentiation. Although the functions of LIN28A in PSCs and differentiation have been extensively studied, this study provides novel mechanistic insights into how LIN28A and PABP regulate mRNA polyA length and stability. This study is conceptually and technically novel as it applies machine learning and bioinformatics tools to discover novel factors that play a role in the regulation of mRNA decay.

Overall, this study is interesting and the findings are critical to understanding the molecular regulation of RNA-binding proteins and mRNA metabolism in the

transition between different cellular fates. The following main and minor points are for the authors to consider in further improving the manuscript.

We thank the reviewer for their positive appraisal of our work and for highlighting the relevance of the study.

Major Points:

1. The expression of the LIN28A protein (and LIN28B) in PSCs and during the primed transition should be provided. Functional redundancy in the pluripotent state transition was previously observed (PMID 27320042). Therefore, the authors should explain why LIN28A, but not Lin28B, was not identified in the pipeline applied in this study and discuss the potential implications in cell fate transitions.

We thank the reviewer for highlighting an important point that LIN28A/LIN28B can have complementary functions in some cases, such as when reprogramming MEFs to iPSCs, and both were reported to contribute to the conversion to a primed state after 5 days in primed media with fibroblast growth factor 2 (FGF2)/activin ((Shyh-Chang and Daley, 2013); (Zhang et al., 2016).

We profiled mRNA and protein expression of LIN28A/B in PSCs to observe that Lin28b expression is almost 100-fold lower compared to Lin28a, with RPKM values around 4 in our PSC priming condition (GEO accession GSE169555) (Fig. R1A). To validate the exclusive role of LIN28A in destabilising the naïve pluripotency regulon, we next analyzed RNA-seq data from LIN28A KO and rescued LIN28A-WT cells with active MEK and WNT signaling, as in Figures 2 and 4. Under these experimental conditions, *Lin28b* TPM values ranged from 0.3 to 1.1 (ENA PRJEB60519). Taking into consideration the threshold for noise being set at 5 RPKM, Lin28b expression was thus close to undetectable levels in our data. Moreover, we failed to observe any LIN28B expression with WB or any signal in the available proteome datasets in PSCs (deposited to the ProteomeXchange Consortium with the dataset identifier PXD024357) (Fig. R1B). Expression of LIN28A is further described in the next response along with PABPC1. Thus, we believe the reason our pipeline didn't detect LIN28A as a candidate regulator is because we couldn't detect any *Lin28b* expression signal prior to primed pluripotency. However, LIN28b does become expressed in the primed state, and given the previous findings of (Zhang et al., 2016), it remains possible that LIN28B plays a role in the maintenance of the primed state. Since we didn't evaluate this directly, we prefer not to speculate on this in the manuscript, unless the reviewer would wish that we do?

We add that LIN28A is predominantly described as a cytoplasmic protein and has been detected in association with ribosomes, P-bodies, and stress granules (Balzer and Moss, 2007; Piskounova et al., 2011; Shyh-Chang and Daley, 2013), whereas LIN28B was found to be exclusively localized to nucleoli across cancer cell lines, where it colocalises with the nucleolar marker Fibrillarin (Piskounova et al., 2011). We further validated this observation with analysis of the publicly available ProteinAtlas immunostaining of LIN28B, and with inducible overexpression

of LIN28B-FLAG in PSCs, which led to its localisation to nucleoli, without detectable expression in the cytoplasm beyond the background signal (Fig. R1B).

To clarify these points, we now further expanded the discussion section to highlight LIN28A non-redundant role: We demonstrate the biological potency of pLIN28A in directly regulating mRNA decay, which complements the well-studied capacity of LIN28 paralogs to indirectly affect mRNA stability via *let7* repression (Heo et al., 2009). The strong mRNA stability defects upon LIN28A loss indicates lack of redundancy with LIN28B, which is predominantly a nucleolar protein (Piskounova et al., 2011), with undetectable expression prior to priming in our cells. Beyond tumorigenesis (Lin and Gregory, 2015; Zou et al., 2022), pLIN28A-mediated direct regulation of mRNA decay might also account for the developmental timing of *C.elegans* larvae (Vadla et al., 2012) and the functions of LIN28A in glucose metabolism (Zhang et al., 2016), and could be relevant for a range of diseases where LIN28A has been implicated, including mouse tissue repair (Shyh-Chang et al., 2013), human β cell differentiation (Zhou et al., 2020) and Parkinson's disease pathogenesis (Chang et al., 2019).

Fig. R1. A.) Western Blot analysis of LIN28A and LIN28B in extracts prepared from 4 different PSC lines along with RNAseq quantifications for the respective *Lin28a* and *Lin28b* transcripts. B.) Representative immunofluorescence photomicrograph of (top) PSCs stably expressing LIN28B-FLAG from a dox-inducible PiggyBac construct and (below) publicly available ProteinAtlas immunostaining of LIN28B in HepG2 cells.

2. The expression of the PABPC1 protein should be provided as it is the second hit in Fig. 1e and a major focus of the study.

The reviewer is highlighting an important point and we now include the quantification of expression of LIN28A and PABPC1 during the transition from nPSCs towards primed PSCs, which is now included in modified Expanded Figure 1. We analysed the mass spectrometry results from the published quantitative proteomics study (Yang et al., 2019) and provide differential expression analysis of proteins during naïve-to-primed transition, comparing cells maintained in 2iLIF stage to 12, 24, 36, and 48hours upon transferring them to priming medium supplemented with *Fgf2* and *ActivinA*, thereby activating MEK and Nodal signalling, respectively. We observe a gradual increase of LIN28A during embryonic priming, while in contrast the expression of PABPC1 remains constant during naïve-to-primed transition (New Extended Data Fig. 1b).

New Extended Data Fig. 1b: Temporal dynamics of relative protein levels during naïve-to-primed transition (compared to 2iLIF - 0h) of naïve pluripotency gene (KLF4) and two top-ranking RBP to predict the transcripts with decreased stability, LIN28A and PABPC1. Proteome quantifications are analysed from (Yang et al., 2019).

3. In Fig. 1, the authors used sophisticated deep learning algorithms to discover the important features that relate to the rapid degradation of the naïve mRNA regulon. Since only the top hit, LIN28A, was chosen for further study, the authors should at least discuss the relevance of other candidates in Fig. 1e to highlight the robustness of such algorithms.

As suggested by the reviewer, we discuss the relevance of additional two top predicted RBPs involved in selective mRNA decay that are both also verified with a permutation test (Extended Data Fig. 1d). The following text has been added into the first paragraph:

“Apart from LIN28A, top predicted RBPs include also PABPC1, which was found capable of inducing mRNA decay under specific scenarios through its interaction with Ccr4-Not (Webster et al., 2018), and a helicase SKIV2L (Tuck et al., 2020), which can initiate 3’–5’ decay by binding to 3’UTRs of a subset of transcripts.”

The value of the ML model is additionally highlighted with additional top hits, such as XRN1 that is believed to act redundantly with SKI/SKIV2L complex in bulk RNA decay, based on synthetic lethality in yeast (Anderson and Parker, 1998) and Isocitrate dehydrogenases IDH1 with to-date unknown function in mRNA decay but intriguing localisation to P-bodies, hubs of mRNA decay (Chen and Shyu, 2013; Decker and Parker, 2012; Zheng et al., 2008).

4. The authors quantified protein expression by immunostaining and presented the distribution of intensities using boxplots with p-values (Fig. S1b, Fig. S2g). While

staining data can be arbitrary, WB data are more straightforward and should be provided. This also applies to the AP staining data (Figs 2i and S2i), where only boxplots were presented without actual images.

We thank the Reviewer for this great suggestion, which indeed strengthens the manuscript. Per their advice, we analysed the mass spectrometry data and provide differential expression analysis of key naïve pluripotency transcription factors at the same timepoints of Fgf2/Activin-triggered embryonic priming to supplement the immunostaining-based quantification of protein amount and compare the protein amount with the respective mRNA levels. In line with our immunostaining-based quantification of protein expression, mass spectrometry results confirm a gradual decrease of SOX2/KLF2/TBX3 (following the trend observed with IF quantification) but only upon the onset of developmental mRNA decay, which is first detected 12hours after the medium change. This additional protein quantification further supports the claim that mRNA depletion precedes protein downregulation, hence we modify the manuscript text to change “accompanied to” by “followed by” an over 2-fold decrease in protein abundance of naïve TFs.

New Extended Data Fig. 1b) Relative mRNA levels and proteome levels (Kalkan et al., 2017; Yang et al., 2019) (compared to 2iLIF at 0hrs) indicated by line plots and fluorescence arbitrary units of immunostained naïve markers SOX2, KLF2, TBX3, NR5A2 ($n = >200$) indicated by boxplots during hours of PSC naïve-to-primed differentiation. Below temporal dynamics of relative protein levels during naïve-to-primed transition (compared to 2iLIF - 0h) of two top-ranking RBP to predict the transcripts with decreased stability, LIN28A and PABPC1.

Per reviewer's request we also added an additional Extended Data Figure 3b with representative images of AP staining quantifications used in Fig 2i.

New Extended Data Fig. 3b) Representative images of alkaline phosphatase stainings as quantified in Fig. 2i

5. PABP has been reported to interact with the LIN28A CSD. The authors could further examine whether WT LIN28A, pLIN28A, or LIN28A-S200A have different binding affinities to PABP. This result would help us to understand the model presented in Fig. 4f.

Again, thanks for this suggestion, which we tried to disentangle in depth. PolyA binding proteins (PABPC1, PABPC4) have been previously found to interact with the CSD of LIN28A (Balzer and Moss, 2007) however in an RNA-dependent manner, which is in line with our observations (Fig. R2). This solidifies the notion that the "PABP bound to A/U sequences appears to be poised for convergence with regulatory RBPs such as pLIN28A that bind at nearby 3' UTR regions". It will be important to examine whether the observed convergent RNA binding of phosphorylated LIN28A with PABP is supported by protein-protein interactions between LIN28A and PABP, which could potentially involve also other RBPs bound to the multivalent AUU-rich motifs. Thus, while the purpose of our study was to identify and characterise the RNA and IDR elements that determine the selectivity and timing of mRNA decay, and how this coordinates a complex morphogenetic event, it will be important to further disentangle the protein-protein interactions that contribute to the signal-induced dynamics of the PABP-RNA hubs.

Our results:

Published (Balzer and Moss, 2007):

Fig R2) Left: LIN28A co-immunoprecipitation (co-IP) analysis of PABPC1 and UPF1 as a control. iLIN28A-WT PSCs supplemented with FGF2 were grown in active MEK signalling and the LIN28A-FLAG transgene was induced for 12 hours. Co-immunoprecipitation (co-IP) assay was performed using anti-PABPC1 and anti-UPF1 antibody and membranes were screened by western blot analysis for co-immunoprecipitating proteins in presence or absence of benzonase. Proteins from lysates without benzonase treatment confirmed the RNA-mediated interaction between PABPC1 and LIN28A, however upon RNA digestion (+benzonase) we fail to detect PABPC1-LIN28A interaction. This confirms RNA-mediated PABPC1-LIN28A interaction. **Right:** Published (Balzer and Moss, 2007) Co-IP immunoblots of TAP-purified complexes using anti-PABP and anti-TAP antisera. P19 cells were transfected with GFP:TAP (GFP), Lin28:TAP (Lin28), CCHC mutant fused to GFP:TAP (CCHC mut:GFP), or CSD mutant fused to GFP:TAP (CSD mut:GFP).

Nevertheless, since one recent study claimed that PABP-LIN28A interact independently of RNA (Yu et al., 2021), a possibility remains of direct interaction being regulated by S200 phosphorylation. Nevertheless, given the complexity of the current manuscript and wealth of existing methodologies, we hope the reviewer agrees that in vitro studies of the binding affinities

of WT LIN28A, pLIN28A, or LIN28A-S200A to PABP will be better described in a separate future manuscript.

6. LIN28A iCLIP of LIN28A was performed under conditions LIN28A-WT (-MEK), (+MEK), and LIN28A-S200A (+MEK) conditions, whereas iCLIP was performed under conditions LIN28A-KO (-MEK) and LIN28A-WT (+MEK). Is there a reason for such an experimental plan? It would be better to perform the iCLIP of LIN28A and PABP under the same conditions to enable a direct comparison and, therefore, to better understand the model. LIN28A-KO can be resistant to differentiation. Furthermore, LIN28A-S200A (+MEK) could be a better control than LIN28A-KO.

We thank the Reviewer for allowing us to explain the experimental design. The reviewer rightly points out that LIN28A iCLIP was performed 6 hours after MEK/ERK activation in the following conditions: (i) the expression of transgene LIN28A-WT was induced in the presence of FGF2 leading to active MEK signalling, i.e., + MEK (ii) the expression of transgene LIN28A-WT was induced with simultaneous inhibition of MEK signalling i.e. MEK-, (iii) the expression of transgene LIN28A-S200A was induced with active MEK signalling. However for PABP iCLIP experiments we wish to highlight that PABP was indeed performed under the same signalling conditions to enable direct effect of LIN28A transgene expression in +MEK condition (list below).

First, PABPC1 iCLIP was produced in 2iLIF conditions (-MEK) to observe an enriched PABP binding at 3'UTR termini already in naïve PSCs (Extended Data Fig. 5). We postulated that this binding “in a poised state to the mRNA termini of mRNAs in naïve pluripotent cells enables rapid response to MEK/ERK signalling”. To understand how the convergence with pLIN28A affects the quantity of PABP binding into the terminal mRNA hubs, we compared the PABPC1 and PABPC4 binding in +MEK conditions when LIN28A is already bound to mRNA termini. To provide better insight into the experimental design we added the following sentence to the manuscript “We therefore proceeded with producing iCLIP data for PABPC4 and PABPC1 in LIN28A knockout cells with or without induction of LIN28A-WT in the presence of FGF2”.

This demonstrates that we produced PABP iCLIP comparisons under active MEK signalling to observe “convergence of pLIN28A into the poised PABP-RNA hubs selectively enhances the PABP binding to these hubs in the DOWN mRNAs, and this rapid pLIN28A-PABP co-assembly at mRNA termini determines the selectivity of mRNA decay”.

List of PABP iCLIP experiments along with all files generated by running the analysis pipeline are now available on our updated Flow easy-to-use software: <https://app.flow.bio/projects/882635250203/%20> and <https://app.flow.bio/projects/340215254997/>

PABPC1	+MEK; iLIN28A-WT	1
		2
PABPC1	+MEK; KO	1
		2

PABPC4	+MEK; iLIN28A-WT	1
		2
PABPC4	+MEK; KO	1
		2
PABPC1	-MEK; endogenous LIN28A expression	1
		2

To decipher the cooperative relationship between LIN28A and PABPCs in mediating transcript stability, we developed a tightly controlled dox-inducible expression system (Extended Data Fig. 2c,d), which has proven essential for studying the impact of RBPs on gene expression modulation in a precise and controlled manner. Hence we believe that the comparison of +/- dox in the same cell line is the most appropriate approach for studying the dynamic effects of LIN28A on RNP reassembly around the PABP-RNA hubs also because this enables only a brief 6hr induction of a transgene, which ensures that both experimental conditions are done within the same cell state, circumventing the limitations (as the reviewer correctly points out and as previously published (Zhang et al., 2016)) of impaired differentiation potential of LIN28A KO cells. Furthermore, I hope that the reviewer agrees with our observation that brief 6hr induction of S200A transgene phenocopies LIN28A KO cells as, markedly, over a thousand genes were significantly downregulated upon induction of LIN28A-WT, but only 66 upon induction of LIN28A-S200A (Extended Data. Fig. 2e-g, both in +MEK).

Minor points:

1. In Fig 4f, top, the middle “LIN28A” was labelled “LIN”, is there a reason for that?
2. In certain places, “LIN28a” was used, which should be 'LIN28A' to be consistent.

Thank you for pointing that out. The reviewer is right; it should be always LIN28A. We fixed both typos.

3. In the Results, it was stated “MEK/ERK phosphorylation of LIN28A is also sufficient for commitment towards the primed state as indicated by the 3-fold increase in the amount of SSEA4-positive cells (Extended Data Fig 2g-h)”. But such data were not presented in Fig. S2g-h.

Thank you for noticing this discrepancy - the data mentioned is now included in Extended Data Fig.3a, c.

New Extended Data Fig. 3

*(a) Quantifications of alkaline phosphatase positive wild-type, iLIN28A-WT and iLIN28A-S200A PSC colonies after 24 hours in 2iLIF, LFW (Lif/Fgf2/Wnt), and LFI (Lif/FGF2/IWP2) with and without dox induction. Dots represent replicates from three independent experiments, Two-sided t-test; ***P = <0.0001.*

*(c) Quantification of flow cytometry analyses of PSCs that were treated with doxycycline (iLIN28A-S200A or iLIN28A-WT), or left untreated (LIN28A KO), and immunostained for naïve pluripotency marker CD17 and primed pluripotency marker SSEA-4 grown in indicated media for 24h. Two-sided t test, ***p < 0.001, **<0.01.*

Reviewer #3:

Remarks to the Author:

In this study, Modic and colleagues provide an elegant analysis on how protein-protein interactions of two RNA Binding Proteins drastically degrades the naïve PSCs regulon to allow the transition towards a primed state. The manuscript present very solid data accompanied by accurate computational analysis. The manuscript is very well written, easy to follow and the conclusions are robust. I suggest the following improvements and highly recommend it for publishing:

We thank the referee for acknowledging this work's importance and for their positive comments.

1- It is widely accepted that the naïve pluripotency transcriptional program is highly dynamic in PSCs firstly due to fluctuations over time in the expression levels of Nanog and other pluripotency regulators; and secondly, that different subpopulations of PSCs have different capacity to differentiate and self-renew. So it has been postulated that PSCs fluctuate in multiple interconvertible states and this metastability and plasticity might be fundamental for their pluripotent identity. Single-cell RNA-seq analysis has revealed that PSCs are highly heterogeneous in their gene expression when grown in standard SL medium conditions. Particularly, two classes of transcriptional heterogeneity have been described according to the gene expression distribution amongst PSCs: bimodal expression, with transcripts present in some cells and not in others; and sporadic expression, with transcripts in a reduced number of cells but at high levels. This transcriptional heterogeneity in PSCs has been proposed to arise from stochastic fluctuations in gene expression, cell cycle phase asynchrony, transcriptional bursting, differential epigenetic regulation, partial differentiation and differences in colony size and morphology, and aid in symmetry breaking and cell-fate decision making as it may create competence for subpopulations of cells to respond to signalling cues. In line with this, single-cell sequencing of SL PSCs showed that cells exist in three distinct states within the whole population. The most numerous subpopulation exhibits high levels of pluripotency factors, another exhibits a differentiation permissive profile while the smallest subpopulation is already in the differentiation path. Another study found two subpopulations instead: one with a transcriptional signature similar to E2.5-3.5 embryos and another, similar to the E5.5 epiblast. They proposed that the transcriptional similarity to the E4.5 epiblast was to be interpreted as the combination of these two subpopulations (see Kolodziejczyk AA, et al., Cell stem Cell, 2015 and Papatsenko D, et al., Stem Cell Reports, 2015). Do the authors think that single cell analysis of LIN28A KO ESCs can help finetuning the exact role this protein in the different subpopulations?

The reviewer rightly highlights that expression profiles are heterogeneous in PSCs, and this heterogeneity can only be captured by single cell analyses. At present we can't answer whether LIN28A-mediated mRNA decay plays any role in such heterogeneity, and

we agree that single cell analysis of LIN28A KO ESCs would be very valuable, and would address this question. We unfortunately were not able to produce such data yet, and we believe the analysis would go beyond the scope of the present manuscript. We hope that researchers working in this area will find the LIN28A KO ESCs of value and we will be happy to share all our cellular models and reagents for such purposes.

2- The naïve-to-primed transition also comprises a metabolic switch. PSCs have a bivalent metabolism consisting of both oxidative phosphorylation and glycolysis, whereas RSCs have low rates of oxygen consumption and full dependence on glycolysis (Zou W, et al., EMBO J 2012). Do the genes that are regulated by LIN28A belong to these pathways?

We thank the reviewer for this excellent suggestion. We have conducted the required GO-term comparisons and included the results in the Expanded Data Figure 2g. As the reviewer correctly pointed out, since the primed pluripotent state almost exclusively relies on glycolysis (Nichols and Smith, 2009; Tesar et al., 2007), we also observe an increased GO-terms associated with glycolytic processes in UP genes, hence those genes that are upregulated upon induction of pLIN28A are more closely linked to the primed cell state.

New Extended Data Fig. 2g: Gene Ontology annotation of DOWN and UP genes. Bar graph representing fold change of GO annotation of biological processes.

3- RSCs express OCT4, SOX2 and NANOG, the core pluripotency transcription factors, also expressed in PSCs. However, the gene expression of Oct4 changes from being regulated mainly by its distal enhancer element in naïve pluripotency to being mainly regulated through its proximal enhancer element in primed pluripotency (Factor DC, et al., Cell Stem Cell, 2014). Did the authors check the promoter usage of the regulated transcripts by LIN28A?

The reviewer is raising an important point and highlighting that the precise expression level of OCT4 determines the fate of PSCs, and Oct4 expression can be regulated by the switch in distal-to-proximal enhancer usage. OCT4 is highly expressed in PSCs and becomes silenced upon differentiation, however OCT4 expression is not changed between naïve, rosette, formative/primed cell state. Therefore OCT4 is considered a general pluripotency factor. To reference some observations from Smith (Kalkan et al., 2017), Jothi (Yang et al., 2019) and our lab (Neagu et al., 2020), OCT4 protein expression

does not change substantially nor significantly upon transition from naïve to primed/formative state within 48 hours - timepoint corresponding to transition to primed PSCs . We next investigated whether dox-inducible LIN28A overexpression impacts OCT4 destabilisation, and we did observe a significant decrease in the OCT4 expression upon 6hrs of dox-induction. These results suggest that LIN28A does not regulate the expression of OCT4 and is therefore unlikely that LIN28A would directly promote the switch of Oct4 distal promoter usage. This further supports that cytoplasmic LIN28A-mediated mRNA decay is specific for naïve mRNA clearance during the transition to primed pluripotency and does not impact general pluripotency factors such as OCT4.

We provide quantification for OCT4 in Fig. R5.

4- Other molecular regulatory players also switch in the naïve-to-primed transition and their effects help shape the two identities. In fact, in comparison with nPSCs, RSCs show a more closed chromatin structure, use different enhancer elements and possess a different set of epigenetic regulators and miRNAs. Additionally, even if the regulatory players are in the same levels in both naïve and primed states, sometimes, their distribution changes. This is the case of H3K4me1, an active enhancer mark whose distribution changes in the naïve-to-primed transition. This change in H3K4me1 distribution plays a key role in establishing the primed identity as perturbing it results in the resetting of RSCs towards a more naïve pluripotent state (Zhang H, et al., cell stem cell, 2016). Can the authors elaborate on LIN28A expression in nPSCs and RSCs?

The reviewer is highlighting an important point and we now include the quantification of LIN28A protein expression using quantitative proteomics to observe that total LIN28A protein abundance was increased during naïve to primed progression in comparison to nPSCs (new Extended Data Fig. 1b), anticorrelating the expression of naïve pluripotency TFs, while in contrast the expression of LIN28A-effector PABPC1 is not changed during the naïve pluripotency exit.

New Extended Data Fig. 1b: Temporal dynamics of relative protein levels during naïve-to-primed transition (compared to 2iLIF - 0h) of naïve pluripotency gene (KLF4) and two top-ranking RBP to predict the transcripts with decreased stability, LIN28A and PABPC1. Proteome quantifications are analysed from (Yang et al., 2019).

5- One study found that the splicing factor and RNA processing HTATSF1 played a role in naïve-to-primed pluripotency transition. More specifically, HTATSF1 mediates intron removal from around 45 ribosomal protein transcripts and thus mediates ribosomal abundance and protein-synthesis. Furthermore, they reported that downregulation of HTATSF1 is crucial in the establishment of primed pluripotency as they were not able to derive RSCs from Htatsf1-overexpressing ESCs. Conversely, they showed that overexpressing Htatsf1 facilitated the conversion of EpiSCs into reverted ESCs (Corsini et al., Cell Stem Cell, 2018). Can the authors check the overlap between the targets of LIN28A and HTATSF1?

We thank the reviewer for suggesting that we look at orthogonal datasets, comparing different studies that identified mechanisms regulation naïve-to-primed cell fate conversion. The reviewer rightly identified an intriguing study of HTATSF1-mediated regulation of intron retention of predominantly ribosomal protein genes, which in turn steers the rRNA biogenesis and to facilitate cell fate transition upon HTASF1 overexpression. We compared the genes identified between the two studies and found minimal overlap (Fig. R3) HTASF1 is a nuclear splicing regulator that specifically contributes gene expression changes via intron retention of <50 genes involved in rRNA processing (Corsini et al., 2018), while in contrast phosphorylated LIN28A targets a large cohort (>1100) of genes that are developmentally downregulated during naïve-to-primed conversion. Since these mechanisms are so different, and there is limited overlap between the regulated genes, it appears that they drive the key, but independent molecular transformations needed for early embryogenesis. We are happy to include the comparison and highlight the independence of these mechanisms of pluripotency control in the discussion if the reviewer finds this appropriate.

Fig. R3: UpSet plot shows the intersection size of the LIN28A-downregulated genes (>1100) with the gene expression changes upon HTASF1 overexpression (Corsini et al., 2018)

Minor comments:

- Mislabelling in figure 1a and supp fig 1a (the median 4.5 fold median decrease is observed at 24 h) same for protein.

Thank you for pointing this out, indeed it should be 24h. We fixed the typo.

- Compare the expression levels and sub cellular localization of LIN28A and LIN28B in naïve and primed ESCs

We thank the reviewer for highlighting an important point to provide expression and localisation information for LIN28A/B paralogs in order to disentangle their potential redundant functions, as for example observed for their function in the regulation of pre-miR let7 processing (Shyh-Chang and Daley, 2013).

We profiled mRNA and protein expression of LIN28A/B in PSCs to observe that Lin28b expression is almost 100-fold lower compared to Lin28a, with RPKM values around 4 in our PSC priming condition (GEO accession GSE169555) (Fig. R4A). To validate the exclusive role of LIN28A in destabilising the naïve pluripotency regulon, we next analyzed RNA-seq data from LIN28A KO and rescued LIN28A-WT cells with active MEK and WNT signaling, as in Figures 2 and 4. Under these experimental conditions, Lin28b TPM values ranged from 0.3 to 1.1 (ENA PRJEB60519). Taking into consideration the threshold for noise being set at 5 RPKM, Lin28b expression was thus close to undetectable levels in our data. Moreover, we failed to observe any LIN28B expression with WB or any signal in the available proteome datasets in PSCs (deposited to the ProteomeXchange Consortium with the dataset identifier PXD024357) (Fig. R1B). Expression of LIN28A is further described in the next response along with PABPC1. Thus, we believe the reason our pipeline didn't detect LIN28A as a candidate regulator is because we couldn't detect any Lin28b expression signal prior to primed pluripotency. However, LIN28b does become expressed in the primed state, and given the previous findings of (Zhang et al., 2016), it remains possible that Lin28b plays a role in the maintenance of the primed state. Since we didn't evaluate this directly, we prefer not to speculate on this in the manuscript, unless the reviewer would wish that we do?

We add that LIN28A is predominantly described as a cytoplasmic protein and has been detected in association with ribosomes, P-bodies, and stress granules (Balzer and Moss, 2007; Piskounova et al., 2011; Shyh-Chang and Daley, 2013), whereas LIN28B was found to be exclusively localized to nucleoli across cancer cell lines, where it colocalises with the nucleolar marker Fibrillarin (Piskounova et al., 2011). We further validated this observation with analysis of the publicly available ProteinAtlas immunostaining of LIN28B, and with inducible overexpression of LIN28B-FLAG in PSCs, which led to its localisation to nucleoli, without detectable expression in the cytoplasm beyond the background signal (Fig. R4B).

To clarify these points, we now further expanded the discussion section to highlight LIN28A non-redundant role: We demonstrate the biological potency of pLIN28A in directly regulating mRNA decay, which complements the well-studied capacity of LIN28 paralogs to indirectly affect mRNA stability via let7 repression (Heo et al., 2009). The strong mRNA stability defects upon LIN28A loss indicates lack of redundancy with LIN28B, which is predominantly a nucleolar protein (Piskounova et al.,

2011), with undetectable expression prior to priming in our cells. Beyond tumorigenesis (Lin and Gregory, 2015; Zou et al., 2022), pLIN28A-mediated direct regulation of mRNA decay might also account for the developmental timing of *C.elegans* larvae (Vadla et al., 2012) and the functions of LIN28A in glucose metabolism (Zhang et al., 2016), and could be relevant for a range of diseases where LIN28A has been implicated, including mouse tissue repair (Shyh-Chang et al., 2013), human β cell differentiation (Zhou et al., 2020) and Parkinson's disease pathogenesis (Chang et al., 2019).

Fig. R4. A.) Western Blot analysis of LIN28A and LIN28B in extracts prepared from 4 different PSC lines along with RNAseq quantifications for the respective Lin28a and Lin28b transcripts. B.) Representative immunofluorescence photomicrograph of (top) PSCs stably expressing LIN28B-FLAG from a dox-inducible PiggyBac construct and (below) publicly available ProteinAtlas immunostaining of LIN28B in HepG2 cells.

- What is the control set that is used for the training network? Can the authors use different sets of 1000 transcripts randomly selected and compare their performance to the top 1000 that they used in terms of CLIP features?

We thank the reviewer for the question and allowing us to explain how the machine learning prediction is controlled. For this we are using the permutation test that works by randomly shuffling the target variable values and then retraining the model on the shuffled data. By this we are obtaining a different random set of transcripts as a control dataset and the model's performance on the shuffled data is then compared to its performance on the original data. If the model's performance on the shuffled data is significantly worse than its performance on the original data, then this suggests that the model is overfitting to the training data. This was not the case in Extended Figure 1g-h, which confirms that the permutation tests successfully controlled the outcome of machine learning classifiers based on gradient-boosted trees.

- Extended figure 1 | expression levels of let7-f and let7-g should be compared to WT.

We acknowledge the reviewer's suggestion. However, we believe that our current approach of comparing let-7 expression levels between uninduced LIN28A KO cells and brief LIN28A-overexpressing cells (6hrs of dox) is more appropriate for the following reasons: i) LIN28A represses let-7 miRNAs, which are however expressed in LIN28A KOs (Shyh-Chang and Daley, 2013). We provided expression levels of let-7 molecules to demonstrate that the function of LIN28A in direct mRNAs destabilisation is let7-independent as brief 6hr LIN28A induction is not sufficient to repress let-7 expression. ii) All subsequent RNAseq and iCLIP experiments are carried out following the same experimental setup to compare compare the

impact of brief LIN28A overexpression to the uninduced LIN28A KO background cells. Therefore the experiment in Extended Fig. 1 serves to demonstrate that LIN28A direct destabilisation of its targets precedes let-7 repression and we therefore do not have a confounding effect of differential let-7 expression between +dox/-dox experiments used in subsequent figures. We hope that the reviewer agrees with the results section and with our conclusion “that mRNA destabilisation preceded expression of the let-7 miRNA family (Extended Data Fig. 1), thus it could not be mediated by the repressive effect of LIN-28 on let-7 biogenesis”. Again, thanks for this suggestion, which we tried to address. The Zhang et al, 2016, publication from Daley lab deposited Microarray data of Lin28a or Lin28b single knockout or double knockout MEF cells reprogrammed to iPS cells (GSE67568), however they do not disclose what media conditions were used (+MEK/-MEK). Therefore to address the overlap between our study and (Zhang et al., 2016) we relied on their qRT-PCR quantification of pluripotent marker genes of iPS cells cultured in regular ES media (LIF/serum) without MEK inhibition. The authors observed a marked increased expression of naïve pluripotency factors Nr5a2, Sall4, Esrrb, Gdf3, Nanog and Tbx3 when comparing WT to LIN28A PSCs. Our RNAseq results confirm their observations, showing that upon induction of LIN28A in MEK+ conditions, naïve pluripotency factors are markedly downregulated (A).

- Authors can use the previously published data by Zhang Jin and colleagues where they show how Lin28 (A and B) regulate transition from naïve to primed pluripotency and check the overlap of the predicted targets

Again, thanks for this suggestion, which we tried to address. The Zhang et al, 2016, publication from Daley lab deposited Microarray data of Lin28a or Lin28b single knockout or double knockout MEF cells reprogrammed to iPS cells (GSE67568), however we could not find sufficient information in the manuscript on experimental design, such as what media conditions (+MEK/-MEK) were used. Therefore to address the overlap between our study and (Zhang et al., 2016) we relied on their qRT-PCR quantification of pluripotent marker genes of iPS cells cultured in regular ES media (LIF/serum) without MEK inhibition. The authors observed a marked increased expression of naïve pluripotency factors Nr5a2, Sall4, Esrrb, Gdf3, Nanog and Tbx3 when comparing WT to LIN28A PSCs. Our RNAseq results confirm their observations, showing that upon induction of LIN28A in MEK+ conditions, naïve pluripotency factors are markedly downregulated (A). Our RNAseq results are also in line with their observations that the expression level of general pluripotency factors (Oct4/Pou5f1, Sox2, Utf1) (B) and the only discrepancy is observed for Zfp42 (Rex1), a naïve pluripotency factor that is as such significantly destabilised upon LIN28A overexpression, which is however not in agreement with results from (Zhang et al., 2016) that argue partial loss of REX1 in LIN28A KO PSCs.

A LIN28A-regulated Naïve pluripotency factors listed in Zhang et al, 2016

B OCT4/SOX2/REX1

C published qRT-PCR quantification of pluripotent marker genes (Zhang et al., 2016)

Fig. R5. Comparison of naïve pluripotency factors (A) and general pluripotency factors (B) in MEK+ LIN28A KO vs briefly induced LIN28A-WT (6hrs dox) RNA-seq data with published qRT-PCR quantification of pluripotent marker genes of iPS cells cultured in regular ES media (LIF/serum) (Zhang et al., 2016). Naïve pluripotency factors are labelled red (corresponding to A) and general pluripotency factors are labelled blue (corresponding to B). Expected anti-correlation between LIN28A KO vs iLIN28A (our data) and WT vs LIN28A KO PSCs (published, (C)) is observed for every pluripotency factor apart from de facto naïve pluripotency factor REX1 (ZFP42) that is depleted upon induction of iLIN28A according to our RNAseq, however in Zhang et al they observe depletion of REX1 in LIN28A KO cell.

- In fig 3a what about the UP regulated genes category?

We thank the reviewer for suggesting that we apply our hybrid deep learning model combining 1D convolutional and recurrent neural networks also to predict 3'UTR nucleotide sequence of UP genes as the sole training feature. As anticipated from the robustness of this model in classifying the transcripts based on sequence input alone, this re-analysis provided validation that AU-rich trimer importance scores are negative (as they are highly predictive for the opposing group of transcripts), while G-rich and GC-rich trimers show positive predictive scores for the classification of 3'UTR sequences into UP group.

Fig. R6. Line plots represent summarised trimer importance scores for classification of 3'UTR sequences underpinning predictive impact of trimers on the selection into UP group. Nucleotide-level importance scores were obtained with SHAP and summarised for each trimer along the evaluated groups of 3'UTRs. Trimers with the highest predictive impact are highlighted.

- In figure 3c can the authors show the IP of Lin28A-S200A in MEK- and + conditions?

We thank the Reviewer for this great suggestion, which indeed strengthens the manuscript. Per their advice, we included the IP of LIN28A-S200A with active and repressed MEK signalling to show that MEK-induced increase in efficiency of RNA crosslinking is specific for LIN28A-WT.

New Extended Data Fig 4f: Representative Li-Cor imaging of the iCLIP nitrocellulose membrane, representing the protein-RNA complexes (above) and the amount of IPed phosphomutant LIN28A-S200A protein in the experiment (below).

- In extended figure 4e-f, if authors consider the region around PAS in the transcriptome, not only the lin28A targets, how would the AUU trimers enrichment look like compared to the lin28A down targets?

Here we would like to clarify that the Ctr subsets of genes correspond to all other non-LIN28A-regulated genes that are expressed above the threshold, so they do represent the transcriptome quite well. We quantified the enriched incidence of AUU-motif group trimers, which are in the region around PAS by a factor of 1.6 or 2.3 higher in LIN28A-bound genes when compared to Control or UP transcripts, respectively (Extended Data Fig. 5e-f). This shows that AUU-multivalent regions at the termini of 3' UTRs are about ~2fold less enriched in the remaining genes when compared to those that are downregulated by LIN28A.

References used in responses to reviewers:

- Anderson, J.S., Parker, R.P., 1998. The 3' to 5' degradation of yeast mRNAs is a general mechanism for mRNA turnover that requires the SKI2 DEVH box protein and 3' to 5' exonucleases of the exosome complex. *EMBO J.* 17, 1497–1506.
- Balzer, E., Moss, E.G., 2007. Localization of the developmental timing regulator Lin28 to mRNP complexes, P-bodies and stress granules. *RNA Biol.* 4, 16–25.
- Chang, M.-Y., Oh, B., Choi, J.-E., Sulistio, Y.A., Woo, H.-J., Jo, A., Kim, J., Kim, E.-H., Kim, S.W., Hwang, J., Park, J., Song, J.-J., Kwon, O.-C., Henry Kim, H., Kim, Y.-H., Ko, J.Y., Heo, J.Y., Lee, M.J., Lee, M., Choi, M., Chung, S.J., Lee, H.-S., Lee, S.-H., 2019. LIN28A loss of function is associated with Parkinson's disease pathogenesis. *EMBO J.* 38, e101196.
- Chen, C.-Y.A., Shyu, A.-B., 2013. Deadenylation and P-bodies. *Adv. Exp. Med. Biol.* 768, 183–195.
- Decker, C.J., Parker, R., 2012. P-bodies and stress granules: possible roles in the control of translation and mRNA degradation. *Cold Spring Harb. Perspect. Biol.* 4, a012286.
- Heo, I., Joo, C., Kim, Y.-K., Ha, M., Yoon, M.-J., Cho, J., Yeom, K.-H., Han, J., Kim, V.N., 2009. TUT4 in concert with Lin28 suppresses microRNA biogenesis through pre-microRNA uridylation. *Cell* 138, 696–708.
- Kalkan, T., Olova, N., Roode, M., Mulas, C., Lee, H.J., Nett, I., Marks, H., Walker, R., Stunnenberg, H.G., Lilley, K.S., Nichols, J., Reik, W., Bertone, P., Smith, A., 2017. Tracking the embryonic stem cell transition from ground state pluripotency. *Development* 144, 1221–1234.
- Lin, S., Gregory, R.I., 2015. MicroRNA biogenesis pathways in cancer. *Nature Reviews Cancer*. <https://doi.org/10.1038/nrc3932>
- Neagu, A., van Genderen, E., Escudero, I., Verwegen, L., Kurek, D., Lehmann, J., Stel, J., Dirks, R.A.M., van Mierlo, G., Maas, A., Eleveld, C., Ge, Y., den Dekker, A.T., Brouwer, R.W.W., van IJcken, W.F.J., Modic, M., Drukker, M., Jansen, J.H., Rivron, N.C., Baart, E.B., Marks, H., Ten Berge, D., 2020. In vitro capture and characterization of embryonic rosette-stage pluripotency between naive and primed states. *Nat. Cell Biol.* 22, 534–545.
- Nichols, J., Smith, A., 2009. Naive and primed pluripotent states. *Cell Stem Cell* 4, 487–492.
- Piskounova, E., Polytarchou, C., Thornton, J.E., LaPierre, R.J., Pothoulakis, C., Hagan, J.P., Iliopoulos, D., Gregory, R.I., 2011. Lin28A and Lin28B inhibit let-7 microRNA biogenesis by distinct mechanisms. *Cell* 147, 1066–1079.
- Shyh-Chang, N., Daley, G.Q., 2013. Lin28: primal regulator of growth and metabolism in stem cells. *Cell Stem Cell* 12, 395–406.
- Shyh-Chang, N., Zhu, H., Yvanka de Soysa, T., Shinoda, G., Seligson, M.T., Tsanov, K.M., Nguyen, L., Asara, J.M., Cantley, L.C., Daley, G.Q., 2013. Lin28 enhances tissue repair by reprogramming cellular metabolism. *Cell* 155, 778–792.
- Tesar, P.J., Chenoweth, J.G., Brook, F.A., Davies, T.J., Evans, E.P., Mack, D.L., Gardner, R.L., McKay, R.D.G., 2007. New cell lines from mouse epiblast share defining features with human embryonic stem cells. *Nature* 448, 196–199.
- Tuck, A.C., Rankova, A., Arpat, A.B., Liechti, L.A., Hess, D., Iesmantavicius, V., Castelo-Szekely, V., Gatfield, D., Bühler, M., 2020. Mammalian RNA Decay Pathways Are Highly Specialized and Widely Linked to Translation.

Mol. Cell 77, 1222–1236.e13.

- Vadla, B., Kemper, K., Alaimo, J., Heine, C., Moss, E.G., 2012. lin-28 controls the succession of cell fate choices via two distinct activities. *PLoS Genet.* 8, e1002588.
- Webster, M.W., Chen, Y.-H., Stowell, J.A.W., Alhusaini, N., Sweet, T., Graveley, B.R., Collier, J., Passmore, L.A., 2018. mRNA Deadenylation Is Coupled to Translation Rates by the Differential Activities of Ccr4-Not Nucleases. *Mol. Cell* 70, 1089–1100.e8.
- Yang, P., Humphrey, S.J., Cinghu, S., Pathania, R., Oldfield, A.J., Kumar, D., Perera, D., Yang, J.Y.H., James, D.E., Mann, M., Jothi, R., 2019. Multi-omic Profiling Reveals Dynamics of the Phased Progression of Pluripotency. *Cell Systems*. <https://doi.org/10.1016/j.cels.2019.03.012>
- Yu, N.-K., McClatchy, D.B., Diedrich, J.K., Romero, S., Choi, J.-H., Martínez-Bartolomé, S., Delahunty, C.M., Muotri, A.R., Yates, J.R., 3rd, 2021. Interactome analysis illustrates diverse gene regulatory processes associated with LIN28A in human iPS cell-derived neural progenitor cells. *iScience* 24, 103321.
- Zhang, J., Ratanasirinrawoot, S., Chandrasekaran, S., Wu, Z., Ficarro, S.B., Yu, C., Ross, C.A., Cacchiarelli, D., Xia, Q., Seligson, M., Shinoda, G., Xie, W., Cahan, P., Wang, L., Ng, S.-C., Tintara, S., Trapnell, C., Onder, T., Loh, Y.-H., Mikkelsen, T., Sliz, P., Teitell, M.A., Asara, J.M., Marto, J.A., Li, H., Collins, J.J., Daley, G.Q., 2016. LIN28 Regulates Stem Cell Metabolism and Conversion to Primed Pluripotency. *Cell Stem Cell* 19, 66–80.
- Zheng, D., Ezzeddine, N., Chen, C.-Y.A., Zhu, W., He, X., Shyu, A.-B., 2008. Deadenylation is prerequisite for P-body formation and mRNA decay in mammalian cells. *J. Cell Biol.* 182, 89–101.
- Zhou, X., Nair, G.G., Russ, H.A., Belair, C.D., Li, M.-L., Shveygert, M., Hebrok, M., Blelloch, R., 2020. LIN28B Impairs the Transition of hESC-Derived β Cells from the Juvenile to Adult State. *Stem Cell Reports* 14, 9–20.
- Zou, H., Luo, J., Guo, Y., Liu, Y., Wang, Y., Deng, L., Li, P., 2022. RNA-binding protein complex LIN28/MSI2 enhances cancer stem cell-like properties by modulating Hippo-YAP1 signaling and independently of Let-7. *Oncogene* 41, 1657–1672.

Decision Letter, first revision:

Message: Our ref: NSMB-A48038A

13th Dec 2023

Dear Dr. Ule,

Thank you for submitting your revised manuscript "Poised PABP-RNA hubs implement signal-dependent mRNA decay in development" (NSMB-A48038A). It has now been seen by the original referees and their comments are below. The reviewers find that the paper has improved in revision, and therefore we'll be happy in principle to publish it in Nature Structural & Molecular Biology, pending minor revisions to comply with our editorial and formatting guidelines.

To facilitate our work at this stage, it is important that we have a copy of the main text as a word file. If you could please send along a word version of this file as soon as possible, we would greatly appreciate it; please make sure to copy the NSMB account (cc'ed above).

Sincerely,

Carolina Perdigoto, PhD
Chief Editor
Nature Structural & Molecular Biology
orcid.org/0000-0002-5783-7106

Reviewer #2 (Remarks to the Author):

Overall, I am satisfied with the revision that addresses all my previous critiques except for the lack of simple WB data on LIN28A (point #1), PABPC1 (point #2), and SOX2/TBX3/LF4 (point#4) in the pluripotent transitions. The MS quantitation data provided from a published study somewhat alleviates my concerns but is not the most ideal. So I will leave this to the editors to make the final decision.

Reviewer #3 (Remarks to the Author):

I am satisfied with the changes introduced in the new version of the draft. The authors have done a great job. GGT

Final Decision Letter:

Message: 28th Jun 2024

Dear Dr. Ule,

We are now happy to accept your revised paper "Poised PABP-RNA hubs implement signal-dependent mRNA decay in development" for publication as an Article in Nature Structural & Molecular Biology.

Your paper will be published online soon after we receive proof corrections and will appear in print in the next available issue. You can find out your date of online publication by contacting the production team shortly after sending your proof corrections.

Please note that *Nature Structural & Molecular Biology* is a Transformative Journal (TJ). Authors may publish their research with us through the traditional subscription access route or make their paper immediately open access through payment of an article-processing charge (APC). Authors will not be required to make a final decision about access to their article until it has been accepted. Find out more about Transformative Journals

Authors may need to take specific actions to achieve compliance with funder and institutional open access mandates. If your research is supported by a funder that requires immediate open access (e.g. according to Plan S principles) then you should select the gold OA route, and we will direct you to the compliant route where possible. For authors selecting the subscription publication route, the journal's standard licensing terms

will need to be accepted, including self-archiving policies. Those licensing terms will supersede any other terms that the author or any third party may assert apply to any version of the manuscript.

Sincerely,
Sara

Sara Osman, Ph.D.
Senior Editor
Nature Structural & Molecular Biology